# Vav2 catalysis-dependent pathways contribute to skeletal muscle growth and metabolic homeostasis

Sonia Rodríguez-Fdez [1,2,3], L. Francisco Lorenzo-Martín [1,2,3], Isabel Fernández-Pisonero [1,2,3], Begoña Porteiro[4,5], Christelle Veyrat-Durebex[6], Daniel Beiroa[4,5], Omar Al-Massadi[4,5,7], Antonio Abad[1,2,3], Carlos Diéguez[4,5,7], Roberto Coppari[6,8], Rubén Nogueiras [4,5,7] & Xosé R. Bustelo [1,2,3 ✉]

Skeletal muscle promotes metabolic balance by regulating glucose uptake and the stimulation of multiple interorgan crosstalk. We show here that the catalytic activity of Vav2, a Rho GTPase activator, modulates the signaling output of the IGF1- and insulin-stimulated phosphatidylinositol 3-kinase pathway in that tissue. Consistent with this, mice bearing a Vav2 protein with decreased catalytic activity exhibit reduced muscle mass, lack of proper insulin responsiveness and, at much later times, a metabolic syndrome-like condition. Conversely, mice expressing a catalytically hyperactive Vav2 develop muscle hypertrophy and increased insulin responsiveness. Of note, while hypoactive Vav2 predisposes to, hyperactive Vav2 protects against high fat diet-induced metabolic imbalance. These data unveil a regulatory layer affecting the signaling output of insulin family factors in muscle.

[1] Centro de Investigación del Cáncer, CSIC-University of Salamanca, 37007 Salamanca, Spain. [2] Instituto de Biología Molecular y Celular del Cáncer, CSIC-University of Salamanca, 37007 Salamanca, Spain. [3] Centro de Investigación Biomédica en Red de Cáncer (CIBERONC), CSIC-University of Salamanca, 37007 Salamanca, Spain. [4] Departamento de Fisioloxía, University of Santiago de Compostela, 15782 Santiago de Compostela, Spain. [5] Centro de Investigación en Medicina Molecular e Enfermidades Crónicas, University of Santiago de Compostela, 15782 Santiago de Compostela, Spain. [6] Department of Cell Physiology and Metabolism, Faculty of Medicine, University of Geneva, 1211 Geneva, Switzerland. [7] Centro de Investigación Biomédica en Red de Fisiopatología de la Obesidad y Nutrición (CIBEROBN), University of Santiago de Compostela, 15782 Santiago de Compostela, Spain. [8] Diabetes Center of the Faculty of Medicine, University of Geneva, 1211 Geneva, Switzerland. ✉email: xbustelo@usal.es

In addition to its intrinsic mechanical actions, the skeletal muscle is responsible for ≈80% of the insulin-stimulated whole-body glucose uptake and clearance under normal physiological conditions[1,2]. It also influences the metabolic status of other tissues such as the white adipose tissue (WAT) and the brown adipose tissue (BAT) via the secretion of a large number of growth factors and hormones[1,3,4]. As a result, the improper function of this tissue can contribute to type 2 diabetes and metabolic syndrome[1,2]. The development and growth of the skeletal muscle mass is under the regulation of insulin growth factor-1 (IGF1) and, to a lesser extent, insulin. Its metabolic functions are mainly regulated by the latter factor, although there is significant redundancy displayed by IGF1[5–9]. These extracellular factors trigger the stimulation of phosphatidylinositol 3-kinase (PI3K), leading to the production of phosphatidylinositol (3,4,5)-triphosphate (PIP$_3$) from phosphatidylinositol-4,5-bisphosphate. PIP$_3$ in turn leads to the plasma membrane recruitment and the subsequent activation of Akt, a serine/threonine kinase that modulates cell growth and metabolism by inhibiting via phosphorylation the activity of tuberin (TSC2), the forkhead (FoxO) family of transcriptional factors, and glycogen synthase kinase 3 (GSK3). The inhibition of TSC2 unleashes mammalian target of rapamycin activity, which, in turn, promotes protein synthesis, metabolic programs, and cell growth. The inhibition of FoxO favors the expression of genes involved in cell metabolism, cell survival, and cell cycle progression. In addition, it silences genes encoding E3 ubiquitin ligases that contribute to the loss of muscle mass. The inhibition of GSK3 leads to the stimulation of glycogen synthase activity, thus favoring the transient storage of glucose as glycogen. Akt also contributes to glucose uptake by promoting the translocation to the plasma membrane of the Glut4 glucose transporter[8,10]. Skeletal muscle mass is controlled by additional mechanisms, such as the tumor growth factor-β family member myostatin and adult stem cells that can be activated in response to regenerative demands[5,11]. Extensive genetic data indicate that tampering with many of the above signaling elements cause muscle atrophy, insulin resistance, type 2 diabetes, and metabolic syndrome in the case of loss-of-function events[9,12,13]. Conversely, gain-of-function changes in IGF1 pathway elements, including IGF1 itself, lead to muscle hypertrophy and protection against metabolic pathologies induced by chronic feeding on hypercaloric diets[14–16]. The same metabolic phenotype is observed when the muscle hypertrophy is triggered via inhibition of myostatin[17,18]. These signaling pathways can also become dysfunctional in the context of hypercaloric feeding, leading to insulin resistance and lipotoxicity in this tissue[5,8].

Rho GTPases are signaling switchers that cycle between an inactive (GDP-bound) and an active (GTP-bound) state in cells. This cycle is mediated by guanosine nucleotide exchange factors (GEFs), which catalyze the activation step, and by GTPase-activating proteins, which promote the inactivation state[19,20]. Previous studies have shown that the main members of this family, Rac1, RhoA, and Cdc42, play key and stepwise roles in skeletal myogenesis[21]. Rac1 and one of its effectors, Pak1, are also important for the translocation of Glut4 in the skeletal muscle[22–25]. RhoA and one of its downstream elements, the serine/threonine kinase Rock1, favor glucose uptake responsiveness in this tissue[26–28]. Most of these studies have focused on short-term responses, so the long-term impact of the deregulation of these GTPases in muscle function and overall metabolic homeostasis remains unknown. Likewise, little information is available regarding the Rho GEFs that are in charge of stimulating those GTPases in the skeletal muscle and other tissues. This is not an easy task to address, given that in mammalian species the Rho GEF family is composed of >70 members[19,20].

In this work, we have focused our attention on Vav2, a GEF that preferentially targets Rac1 and, to a lesser extent, RhoA in vivo[29]. As with the two other subfamily members, Vav1 and Vav3, the catalytic activity of this GEF is regulated by tyrosine phosphorylation by upstream protein tyrosine kinases[29–33]. This regulation entails the phosphorylation-mediated disruption of an autoinhibited state that is maintained in the nonphosphorylated state by interactions of the most N- and C-terminal domains with a central catalytic cassette composed of the catalytic Dbl homology (DH), a pleckstrin homology (PH), and a C1-like zinc-finger domain[29,31–36]. Disruption of this autoinhibitory loop by either truncation or point mutations generates Vav proteins with constitutive, phosphorylation-independent catalytic activity[29,33]. To assess the role of the catalytic activity of Vav2 in metabolic homeostasis, we utilized two complementary catalytic hypomorphic (Vav2[L332A]) and gain-of-function (Vav2[Onc]) strains of Vav2 knock-in mice. Our results indicate that the catalysis-dependent pathways of this GEF directly impact on IGF1 and insulin signaling in the skeletal muscle, as well as in the overall metabolic balance of mice.

## Results

### Vav2 catalytic output affects muscle weight and myocyte size.

We have utilized two mirror-image mouse models to analyze the contribution of the Vav2 catalysis-dependent pathways to organismal physiology. On the one hand, we used a recently described mouse knock-in strain (Vav2[L332A]) that expresses from the germline a Vav2 protein with a point mutation (L332A) in the catalytic site of the DH domain (Supplementary Fig. 1a)[37]. This mutant protein exhibits a ≈70% and 100% reduction in the enzymatic activity towards Rac1 and RhoA when compared to the wild-type (WT) counterpart, respectively[37]. We have shown before that Vav2[L332A/L332A] mice show impaired skin tumorigenesis[37]. By contrast, they do not exhibit the buphthalmia and elevation in blood pressure typically seen in the case of Vav2[−/−] mice[38–40], thus suggesting that Vav2[L332A] behaves as a hypomorphic mutant allele[37]. On the other hand, we utilized another previously described catalytic gain-of-function knock-in model (Vav2[Onc]) that expresses from the germline a mutant and HA-tagged version of Vav2 (Δ1–184, referred to hereafter as Vav2[Onc]) that exhibits constitutive GEF activity towards Rac1 and RhoA due to the removal of the N-terminal inhibitory regions (Supplementary Fig. 1a)[35,41]. We have shown before that this strain exhibits hypotension[41], the mirror-image phenotype of hypertension found in Vav2[−/−] mice[39,40]. Importantly, the Vav2[L332A] and Vav2[Onc] mutant proteins are expressed from the endogenous locus. Due to this, they exhibit patterns and levels of expression in tissues similar to those found in the case of the endogenous WT protein (Supplementary Table 1)[37,41]. No changes in expression were observed in Rac1 or other Rho GEFs in the tissues surveyed of those mice (Supplementary Table 1), thus indicating the absence of compensatory mechanisms in response to changes in Vav2 activity.

When subjected to nuclear magnetic resonance analyses, we found that the Vav2[L332A/L332A] mice have reduced levels of total lean mass in all the age time-points analyzed when compared to controls (Fig. 1a). Despite this, they have a body weight similar to controls between the fourth and eleventh month of age (Fig. 1a), probably because the lower lean mass is compensated by a parallel increase of total body fat depots in the animals (Fig. 1a). This increase in body fat eventually favors higher gains of weight in 12- and 13-month-old Vav2[L332A/L332A] animals when compared to age-matched controls (Fig. 1a). By contrast, we observed that 3- to 10-month-old Vav2[Onc/Onc] mice contain more lean mass than the same age-matched controls (Fig. 1b). Despite this, with

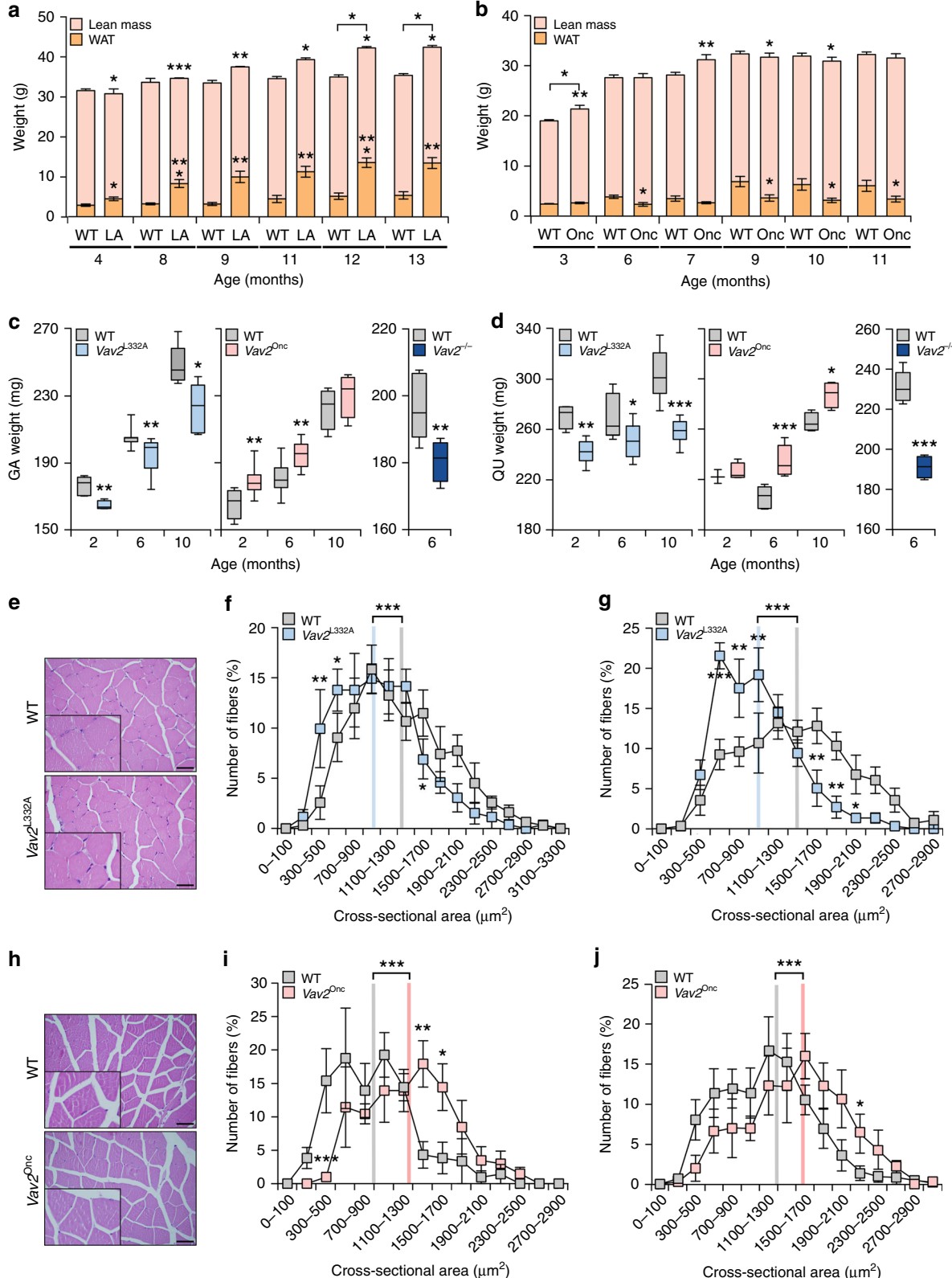

the exception of 3-month-old animals, these mice do not have a higher body weight than controls (Fig. 1b). This is probably due to parallel reduction in the content of total body fat found in these catalytic gain-of-function mice (Fig. 1b). In line with these observations, we observed that the gastrocnemius (Fig. 1c, left panel) and the quadriceps (Fig. 1d, left panel) are consistently lighter in $Vav2^{L332A/L332A}$ mice than in controls. This difference

is already detected in the earliest time-point analyzed (2-month-old mice) and maintained in subsequent age time-points (Fig. 1c, d; left panels). These changes in weight are more aggravated in the case of $Vav2^{-/-}$ knockout mice that totally lack Vav2 expression (Fig. 1c, d; right panels). Histological analyses indicated that the fibers present in the gastrocnemius muscle of $Vav2^{L332A/L332A}$ mice show a normal structure at all ages

**Fig. 1 Vav2 catalytic output affects muscle mass and fiber size. a, b** Body composition of animals of the indicated age and genotype (bottom) measured by nuclear magnetic resonance. LA, $Vav2^{L332A/L332A}$ mice; Onc, $Vav2^{Onc/Onc}$ mice. Data represent the mean ± SEM. Asterisks indicate differences in white adipose tissue (WAT) or lean mass with respect to their age-matched WT mice. The differences in weight between control and the appropriate knock-in mice are indicated using a horizontal bracket. *, $P = 0.0101$ (WAT, 4-month-old mice, **a**), $P = 0.0343$ (lean mass, 4-month-old mice, **a**), $P = 0.0102$ (lean mass, 11-month-old mice, **a**), $P = 0.0197$ (lean mass, 12-month-old mice, **a**), $P = 0.0337$ (lean mass, 13-month-old mice, **a**), $P = 0.0103$ (total, 12-month-old mice, **a**), $P = 0.0354$ (total, 13-month-old mice, a), $P = 0.0171$ (WAT, 6-month-old mice, **b**), $P = 0.0273$ (WAT, 9-month-old mice, **b**), $P = 0.0327$ (WAT, 10-month-old mice, **b**), $P = 0.0286$ (lean mass, 9-month-old mice, **b**), and $P = 0.0499$ (lean mass, 10-month-old mice, **b**); **, $P = 0.0029$ (WAT, 9-month-old mice, **a**), $P = 0.0035$ (WAT, 11-month-old mice, **a**), $P = 0.0012$ (WAT, 13-month-old mice, **a**), $P = 0.0061$ (lean mass, 9-month-old mice, **a**), $P = 0.0076$ (lean mass, 3-month-old mice, **b**) and $P = 0.0093$ (lean mass, 7-month-old mice, **b**); ***, $P = 0.0008$ (WAT and lean mass, 8-month-old mice, **a**) and $P = 0.0003$ (WAT, 12-month-old mice, **a**) using two-tailed Student's $t$ tests. $n = 5$ (9-, 11-, 12-, and 13-month-old WT controls for $Vav2^{L332A/L332A}$ mice; 6-, 7-, 9-, 10-, and 11-month-old $Vav2^{Onc/Onc}$ mice and their matched controls), 6 (4- and 5-month-old WT controls for $Vav2^{L332A/L332A}$ mice and 4-month-old $Vav2^{L332A/L332A}$ mice), 7 (8-, 9-, 11-, 12- and 13-month-old $Vav2^{L332A/L332A}$ mice), 10 (3-month-old $Vav2^{Onc/Onc}$ mice), and 12 (3-month-old WT animals). **c, d** Weight of the gastrocnemius (GA) (**c**) or quadriceps (QU) (**d**) from mice of the indicated ages (bottom) and genotypes (inset). Boxes, lines inside boxes, and bars represent the 25th to 75th percentiles, the median, and the minimum and maximum values, respectively. Asterisks refer to the $P$ value between age-matched animals. *, $P = 0.0271$ (6-month-old WT vs. $Vav2^{L332A/L332A}$ mice, QU), $P = 0.0116$ (10-month-old WT vs. $Vav2^{L332A/L332A}$ mice, GA), and $P = 0.0129$ (10-month-old WT vs. $Vav2^{Onc/Onc}$ mice, QU); **, $P = 0.0063$ (2-month-old WT vs. $Vav2^{L332A/L332A}$ mice, GA), $P = 0.0073$ (6-month-old WT vs. $Vav2^{L332A/L332A}$ mice, GA), $P = 0.0066$ (2-month-old WT vs. $Vav2^{Onc/Onc}$ mice, GA), $P = 0.0053$ (6-month-old WT vs. $Vav2^{Onc/Onc}$ mice, GA), $P = 0.0079$ (6-month-old WT vs. $Vav2^{-/-}$ mice, GA), and $P = 0.0020$ (2-month-old WT vs. $Vav2^{L332A/L332A}$ mice, QU); ***, $P = 0.0007$ (10-month-old WT vs. $Vav2^{L332A/L332A}$ mice, QU), $P = 0.00007$ (6-month-old WT vs. $Vav2^{Onc/Onc}$ mice, QU), and $P = 0.00006$ (6-month-old WT vs. $Vav2^{-/-}$ mice, QU) using two-tailed Student's $t$ tests. $n = 5$ (WT controls for 2- and 10-month-old $Vav2^{L332A/L332A}$, GA and $Vav2^{-/-}$ mice), 11 (WT controls for 6-month-old $Vav2^{L332A/L332A}$ mice), 7 (6-month-old $Vav2^{Onc/Onc}$ mice, GA and WT controls for 2-month-old $Vav2^{Onc/Onc}$ mice, GA), 9 (6-month-old $Vav2^{L332A/L332A}$ mice and 2-month-old $Vav2^{Onc/Onc}$ mice, GA), 10 (WT controls for 6-month-old $Vav2^{Onc/Onc}$ mice, GA), 6 (10-month-old $Vav2^{L332A/L332A}$ mice, their controls and 2-month-old $Vav2^{L332A/L332A}$ mice, QU), 3 (2-month-old $Vav2^{Onc/Onc}$ mice, QU), 8 (WT controls and 6-month-old $Vav2^{Onc/Onc}$ mice, QU), and 4 (rest of conditions). **e** Representative images of hematoxylin–eosin-stained sections of gastrocnemius muscles of 6-month-old animals of the indicated genotypes (left). $n = 5$ animals per group. Scale bar, 50 μm. **f, g** Distribution of the cross-sectional area of the fibers in the gastrocnemius muscle of 2- (**f**) and 6-month-old (**g**) animals of the indicated genotypes (inset). Values are presented as mean ± SEM. Gray and blue lines indicate the mean cross-sectional fiber area of WT and $Vav2^{L332A/L332A}$ animals, respectively. *, $P = 0.0472$ (500–700 range, **f**), $P = 0.0412$ (1500–1700 range, **f**), and $P = 0.0347$ (**g**); **, $P = 0.0032$ (**f**), $P = 0.0027$ (1500–1700 range, **g**), $P = 0.0032$ (1700–1900 range, **g**), $P = 0.0023$ (700–900 range, **g**), and $P = 0.0010$ (900–1100, **g**); ***, $P = 0.00004$ using two-tailed Student's $t$ tests (in the case of the mean) and two-way ANOVA followed by Fisher's LSD tests. $n = 4$ (2-month-old $Vav2^{L332A/L332A}$ animals) and 5 (rest of cases). **h** Representative images of hematoxylin–eosin-stained sections of gastrocnemius muscles of 6-month-old animals of the indicated genotypes (left). $n = 5$ WT and 6 $Vav2^{Onc/Onc}$ mice. Scale bar, 50 μm. **i, j** Distribution of the cross-sectional fiber area in the gastrocnemius of 2- (**i**) and 6-month-old (**j**) mice of the indicated genotypes (inset). Data are represented as mean ± SEM. Gray and light red lines indicate the mean fiber area of WT and $Vav2^{Onc/Onc}$ animals, respectively. *, $P = 0.0121$ (**i**), $P = 0.0425$ (**j**); **, $P = 0.0014$; ***, $P = 0.0008$ using two-tailed Student's $t$ tests (in the case of the mean) and two-way ANOVA followed by Fisher's LSD tests. $n = 4$ (**i**), 6 (WT mice, **j**), and 5 (6-month-old $Vav2^{Onc/Onc}$ mice). Source data for this figure are provided as a Source data file.

analyzed (Fig. 1e). However, we observed that their cross-sectional areas become progressively smaller than those of controls as the $Vav2^{L332A/L332A}$ animals age (Fig. 1f, g and Supplementary 1b). By contrast, we have not observed a statistically significant change in the total number of cells as determined by the number of nuclei/fiber in histological sections (Supplementary Fig. 1c, left panel). These data indicate that the reduction in muscle mass is primarily due to the reduction in fiber size. These defects do not seem to impair normal muscle function, as determined by the similar distance (Supplementary Fig. 1d) and time (Supplementary Fig. 1e) ran by 3-month-old $Vav2^{L332A/L332A}$ and WT mice under a treadmill challenge. This is comparable to observations made in mice with impaired PI3K signaling in the skeletal muscle[12,42]. The analysis of $Vav2^{Onc/Onc}$ mice revealed a reverse phenotype, with the gastrocnemius (Fig. 1c, middle panel) and quadriceps (Fig. 1d, middle panel) exhibiting more weight than controls in most age time-points analyzed. This phenotype is associated with a parallel increase in the cross-sectional area of the fibers of these muscles when compared to controls (Fig. 1h–j and Supplementary Fig. 1f). However, the number of cells does not change (Supplementary Fig. 1c, right panel). We did not find any alteration either in the exercise capacity of 3-month-old $Vav2^{Onc/Onc}$ mice when using the treadmill test (Supplementary Fig. 1g, h). Collectively, these results indicate that the catalytic activity of Vav2 regulates muscle mass and fiber size.

These two knock-in strains do not show statistically significant changes in the total number of the Sca1$^-$;CD45$^-$;CD34$^+$;integrin α7$^+$ population of muscle stem (satellite) cells when compared to WT mice (Supplementary Fig. 2a, b). However, when tested in culture, the $Vav2^{L332A/L332A}$ (Supplementary Fig. 2c, d) and $Vav2^{Onc/Onc}$ (Supplementary Fig. 2e, f) satellite cells display lower and higher proliferation rates than controls, respectively. The differentiating $Vav2^{Onc/Onc}$ satellite cells also show higher expression of the terminal differentiation marker myosin heavy-chain II (Supplementary Fig. 2g, h). Unlike the rest of muscle-associated alterations found in these mice, we found no differentiation defects in the case of primary $Vav2^{L332A/L332A}$ muscle stem cells (S.R.-F. and X.R.B., unpublished data). These results indicate that changes in Vav2 catalytic output influence the proliferation, but not the overall number of muscle satellite cells.

**Vav2 catalytic output affects insulin and IGF1 signaling.** Muscle metabolism and growth are under the control of the insulin-PI3K-Akt and IGF1-PI3K-Akt axes, respectively[5,8–10]. We observed that the phosphorylation of Akt in the Thr$^{308}$ residue was reduced in the gastrocnemius of both 2-month- and 4-month-old $Vav2^{L332A/L332A}$ mice that were infused with optimal concentrations of insulin (Fig. 2a, b). We also found reductions in the phosphorylation of the Akt Ser$^{473}$ phosphosite and GSK3, although such defects are only statistically significant in the case of 4-month-old $Vav2^{L332A/L332A}$ mice (Fig. 2a, c, d). We did not see any increase in Akt and GSK3 phosphorylation levels in the gastrocnemius of $Vav2^{Onc/Onc}$ mice when treated with optimal concentrations of insulin compared to controls (Supplementary

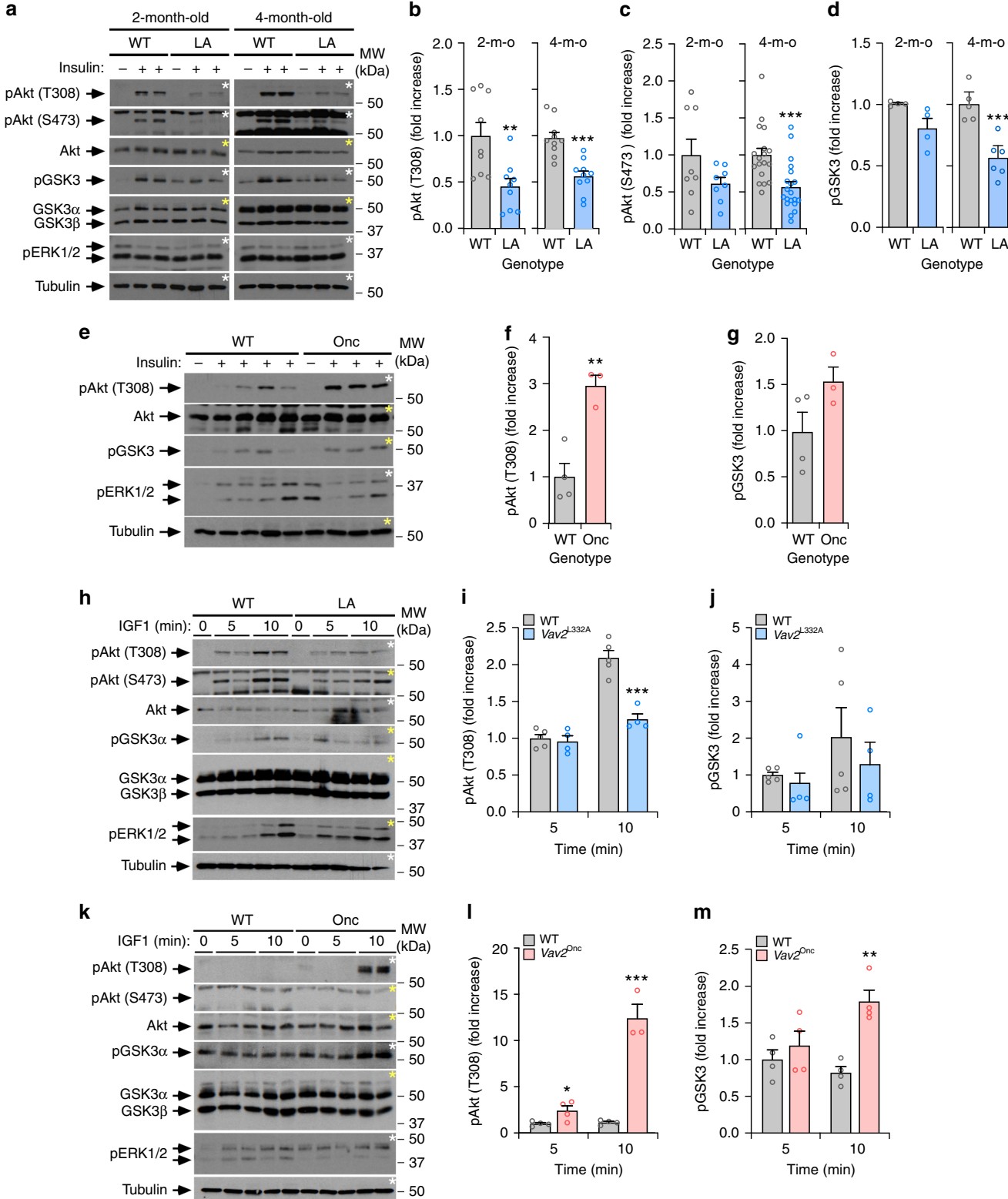

Fig. 3a–c). However, the phosphorylation of the downstream S6K does show an increase in this muscle in insulin-treated 4-month-old $Vav2^{Onc/Onc}$ mice (Supplementary Fig. 3a–d). We did find, however, elevated levels of phosphorylation of Akt when 3-month-old $Vav2^{Onc/Onc}$ mice were infused with suboptimal amounts of insulin (Fig. 2e–g). The implication of Vav2 catalysis-dependent pathways in insulin signaling is muscle-specific, since we could not find any statistically significant alteration in insulin signaling in the liver (Supplementary Fig. 3e, f) and WAT (Supplementary Fig. 3g, h) in $Vav2^{L332A/L332A}$ mice. We did not detect either any signaling alteration in the liver of $Vav2^{Onc/Onc}$ mice (Supplementary Fig. 3e, f). However, we did find higher levels of insulin-induced Akt phosphorylation in WAT from 4-month-old $Vav2^{Onc/Onc}$ mice (Supplementary Fig. 3g, h).

Reduced phosphorylation of Akt in Thr308 is also observed in 4-month-old $Vav2^{L332A/L332A}$ mice infused with IGF1 (Fig. 2h, i).

**Fig. 2 Vav2 catalytic output affects insulin and IGF1 signaling. a–d** Representative immunoblots (**a**) and quantification of the phosphorylation levels of indicated proteins and phosphosites (**b–d**, left) in gastrocnemius muscles from WT and $Vav2^{L332A/L332A}$ (LA) mice of the indicated ages (top) that were either untreated (−) or treated (+) with insulin for 5 min. In **a**, aliquots from the same lysates were analyzed in separate blots (each identified with same color asterisks). The same notation has been used in the rest of figures. MW, molecular weight; p, phospho; m-o, month old. In **b–d**, data are presented as mean ± SEM. **, $P = 0.005$; ***, $P = 0.0001$ (**b**), $P = 0.0007$ (**c**), $P = 0.0009$ (**d**) using two-tailed Student's $t$ tests. $n = 9$ (**b** and 4-month-old WT in **c**), 8 (2-month-old mice of both genotypes, **c**), 10 (4-month-old $Vav2^{L332A/L332A}$ animals, **b**), 18 (4-month-old WT mice, **c**), 20 (4-month-old $Vav2^{L332A/L332A}$ animals, **c**), 4 (2-month-old animals of both genotypes, **d**), 5 (4-month-old WT mice, **d**), or 6 (4-month-old $Vav2^{L332A/L332A}$ mice, **d**). **e–g** Immunoblots (**e**) and quantification of the phosphorylation levels of the indicated proteins and phosphosites (**f**, **g**) in the muscle from 3-month-old WT and $Vav2^{Onc/Onc}$ (Onc) mice infused with either placebo (−) or low doses of insulin (0.1 U kg$^{-1}$; +) for 5 min. In **f**, **g**, data of insulin-treated mice are presented as mean ± SEM. **, $P = 0.0041$ using two-tailed Student's $t$ tests. $n = 4$ (WT) and 3 ($Vav2^{Onc/Onc}$) mice. **h–m** Representative immunoblots (**h**, **k**) and quantification of phosphorylation levels of specified proteins and phosphosites (**i**, **j**, **l**, **m**; left) in gastrocnemius muscles from 4-month-old of indicated genotypes (top) upon being exposed to intravenously injected IGF1 for the indicated period of time (top). In **j**, **k**, **m**, **n**, data are presented as mean ± SEM. *, $P = 0.0377$ (**l**); **, $P = 0.0014$ (**m**); ***, $P = 0.0004$ (**i**) and 0.0003 (**l**) using two-tailed Student's $t$ tests. $n = 5$ (WT mice in **i**, **j**) and 4 (**l**, **m**, $Vav2^{L332A/L332A}$ mice) animals per group and genotype. Source data for this figure are provided as a Source data file.

A tendency towards reduced phosphorylation of GSK3 is also detected, although such changes have not reached statistical significance at the time-points analyzed (Fig. 2h, j). Conversely, the infusion of $Vav2^{Onc/Onc}$ mice with suboptimal amounts of IGF1 leads to the marked upregulation of the phosphorylation of both Akt (Thr$^{308}$) and GSK3 in the gastrocnemius (Fig. 2k–m). Under these conditions, we could not detect any significant phosphorylation of the Akt Ser$^{473}$ residue in either WT or $Vav2^{Onc/Onc}$ animals (Fig. 2k). In addition to IGF1, myocyte size can be negatively affected by increased degradation of muscle proteins via the upregulation of the muscle E3 ubiquitin ligases Trim63 (also known as MURF1) and Fbxo32 (also known as either atrogin 1 or MAFbx) at the transcriptional level by a FoxO-dependent mechanism[5,8,43]. However, we did not detect any statistically significant upregulation of the transcripts encoding those enzymes in the skeletal muscle from $Vav2^{L332A/L332A}$ mice using real-time quantitative PCR (qRT-PCR) analyses (Supplementary Fig. 3i).

To further test whether these defects are intrinsic to muscle cells, we next analyzed the role of Vav2 in insulin signaling in the murine myoblast C2C12 cell line using both loss- and gain-of-function approaches. In the former case, we generated independent cell clones in which the endogenous $Vav2$ locus was inactivated using CRISPR-Cas9-based gene-editing techniques (Fig. 3a). As an alternative approach, we also generated independent clones of Vav2-depleted cells using short hairpin RNA (shRNA) interference (Fig. 3b, c). For the gain-of-function experiments, we ectopically expressed either the catalytically hyperactive or a catalytically dead ($Vav2^{Onc+E200A}$) versions of $Vav2^{Onc}$ using a lentiviral delivery approach in the parental C2C12 cells. These two Vav2 proteins were HA-tagged at the N terminus to facilitate detection in cell lysates (Fig. 3d). The CRISPR-Cas9-mediated elimination of endogenous Vav2 leads to impaired phosphorylation of Akt (Ser$^{473}$ and Thr$^{308}$ residues), GSK3, and S6K upon the stimulation of undifferentiated C2C12 cells with insulin (Fig. 3e, left panel and Supplementary Fig. 4a). Interestingly, the extent of those signaling defects becomes amplified as we move from the most upstream to the most downstream signaling elements of the pathway (Fig. 3e, left panels and Supplementary Fig. 4a). By contrast, we did not find any defect in the phosphorylation of IRS1 under these conditions, thus indicating normal function from the upstream insulin receptor (Fig. 3f). Similar defects in the phosphorylation of Akt were found in insulin-stimulated $Vav2$ knockdown C2C12 cells (Fig. 3g).

The expression of HA-Vav2$^{Onc}$ in the parental cells leads to statistically significant increases in the foregoing signaling elements of the PI3K–Akt axis upon the stimulation of undifferentiated C2C12 cells (Fig. 3e, right panels and

Supplementary Fig. 4b). Such upregulation is not observed in the case of the HA-Vav2$^{Onc+E200A}$ mutant (Fig. 3e, right panels and Supplementary Fig. 4b), indicating that Vav2 influences insulin signaling in a catalysis-dependent manner. The antagonistic effects elicited by the depletion and the chronic catalytic activation of Vav2 in insulin signaling output is maintained in differentiated C2C12 cells (Fig. 3h and Supplementary Fig. 4c, d). In agreement with reduced levels of activation of PI3K, we found that the elimination of endogenous Vav2 reduces the production of its main substrate, PIP$_3$, in undifferentiated C2C12 upon insulin stimulation (Fig. 3i, left panel). The ectopic expression of Vav2$^{Onc}$, but not of the catalytically deficient Vav2$^{Onc+E200A}$, promotes in turn high levels of PIP$_3$ production in nonstimulated C2C12 cells (Fig. 3i, right panel).

In line with the observations made in mice, these cell models revealed that the ectopic expression of HA-Vav2$^{Onc}$ protein prompts the expression of genes encoding proteins involved in late differentiation phases of myoblasts such as the transcriptional factor myogenin ($Myog$) and specific myosin heavy-chain subunits ($Myh1$, $Myh4$, and $Myh7$) (Supplementary Fig. 5a, b). The effect of Vav2$^{Onc}$ in the expression of some of these genes ($Myog$, $Myh7$) is further enhanced when cells are stimulated with insulin (Supplementary Fig. 5c). All these effects are catalysis-dependent, as they cannot be triggered when using the catalytically dead version of Vav2 (Supplementary Fig. 5b, c). Confocal immunofluorescence experiments confirmed that Vav2$^{Onc}$-expressing C2C12 cells contain higher levels of myosin heavy-chain II than controls when induced to differentiate in basal differentiation media, insulin, or IGF1 (Supplementary Fig. 5d, e). We could not detect, however, any statistically significant change in the number of myotubes formed by those cells upon differentiation (Supplementary Fig. 5f, g). Despite this, those myotubes are slightly thicker than those found in the case of differentiated WT and Vav2$^{Onc+E200A}$-expressing C2C12 cells (Supplementary Fig. 5h). Using $Vav2$ knockdown cells (Fig. 3b, c), we could only find defects in the expression of the $Myog$ gene under basal conditions (Supplementary Fig. 6a). The differentiation of these cells is also similar to that observed in control C2C12 cells (Supplementary Fig. 6b–d).

**The Vav2–Rac1 axis favors stimulation of the PI3Kα-Akt route.** As in the case of the in vivo experiments using $Vav2^{Onc/}$$^{Onc}$ mice (Fig. 2), we observed that Vav2$^{Onc}$ cannot activate the downstream elements of the insulin pathway in nonstimulated C2C12 cells (Fig. 3e, h and Supplementary Fig. 4b, d). Given that Vav2$^{Onc}$ displays intrinsic, phosphorylation-independent GEF activity[31], those results suggested to us that Vav2 had to be silenced in the absence of upstream signaling or, alternatively,

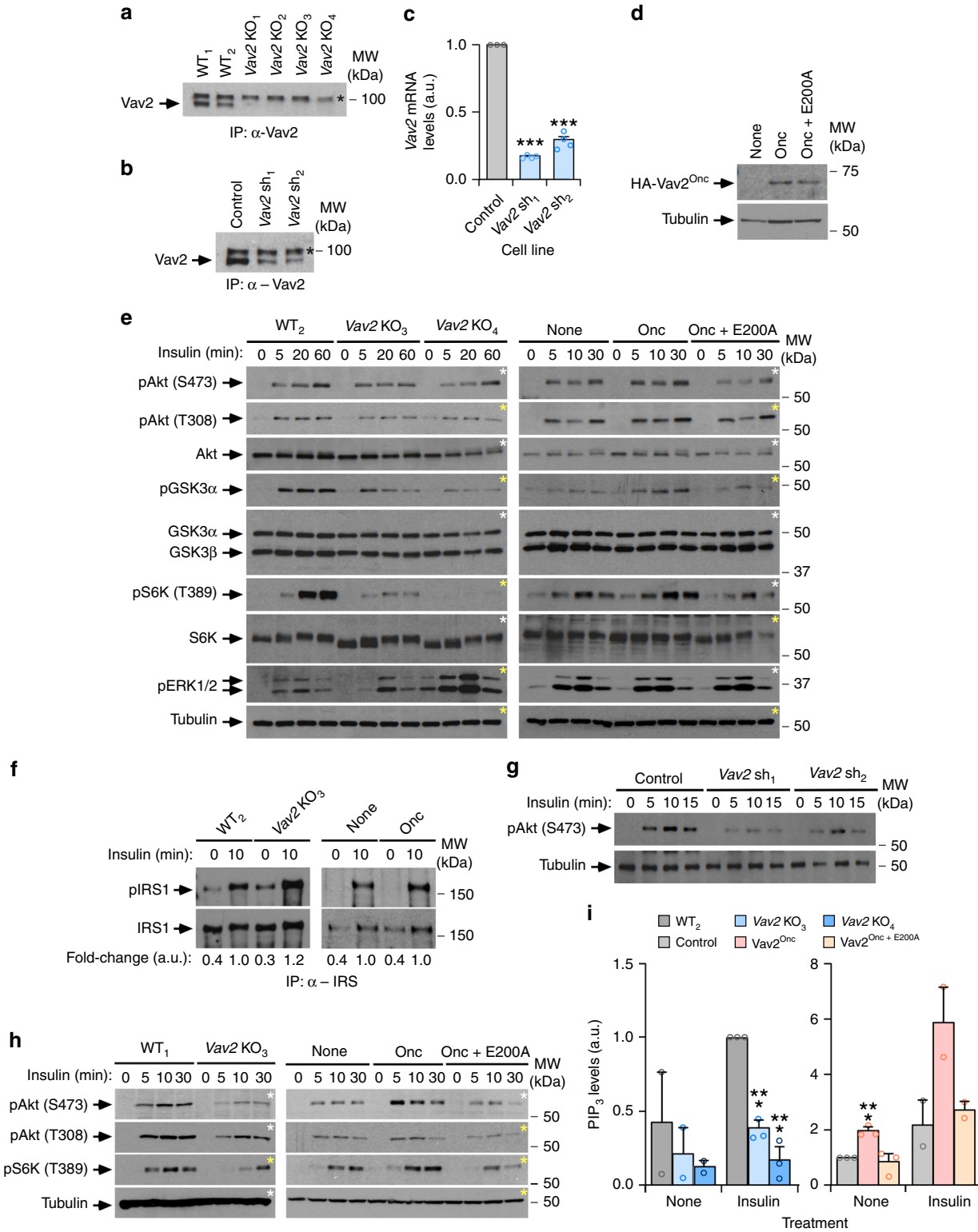

that its ability to stimulate the insulin pathway had to be very transient in the absence of ligand. This latter possibility was consistent with the detection of low, although statistically significant levels of increased $PIP_3$ production in Vav2$^{Onc}$-expressing C2C12 cells under basal conditions (Fig. 3i). To address this issue, we analyzed the activation status of the PI3K–Akt axis in cells using bioreporters containing the Akt PH domain fused to either the mCherry or the enhanced green fluorescent proteins (EGFPs). These bioreporters translocate from the cytosol to the plasma membrane in a $PIP_3$-dependent manner, thus allowing the indirect evaluation of the stimulation status of PI3K using epifluorescence microscopy techniques. As expected, we observed that the mCherry-Akt PH bioreporter moves from a cytosolic to a plasma membrane localization in an insulin-dependent manner in transiently transfected C2C12 cells (Fig. 4a). As a control, a nonchimeric EGFP does not undergo changes in its normal subcellular distribution under those experimental conditions (Fig. 4a). The coexpression of EGFP-Vav2$^{Onc}$, but not of

**Fig. 3 Vav2 catalytic output affects insulin responses in C2C12 cells. a** Expression of endogenous Vav2 in WT ($WT_1$, $WT_2$) and *Vav2* knockout ($KO_1$–$KO_4$) independent clones. A nonspecific band is indicated by an asterisk. IP immunoprecipitation. The WT clones are C2C12 cells subjected to the same protocol used for the generation of the *Vav2* KO clones, but that failed in being gene-edited ($n = 1$). **b, c** Vav2 protein (**b**) and *Vav2* mRNA (**c**) levels present in a control C2C12 cell line and two *Vav2* knockdown cell lines generated with different shRNAs ($sh_1$ and $sh_2$). a.u. arbitrary units. In **b**, data are shown as in **a** ($n = 1$). In **c**, data are shown as mean ± SEM. ***, $P < 0.000001$ ($sh_1$), $P = 0.000005$ ($sh_2$) using two-tailed Student's *t* tests ($n = 3$). **d** Expression of the HA-tagged Vav2 proteins in the C2C12 cell lines generated in this study ($n = 2$). **e** Levels of the specified phosphorylated sites and total proteins in the indicated cell lines (top) upon insulin stimulation. $n = 3$ (left) and $n = 4$ (right) independent experiments. **f** Tyrosine phosphorylation levels (top panel) and total abundance of IRS1 (bottom panel) immunoprecipitated from indicated cells and stimulation conditions (top). The quantification of immunoblots is shown below as the mean of 3 (left) and 4 (right) independent experiments. **g** Immunoblots showing the phosphorylation and total protein levels of the specified proteins in indicated cell lines (top) and insulin stimulation times (top). $n = 3$ independent experiments. **h** Phosphorylation levels of indicated phosphosites (left) in the insulin-stimulated cell lines (top) differentiated for 5 days prior to the stimulation step. $n = 3$ independent experiments. **i** $PIP_3$ levels in indicated cells (bottom) and experimental conditions. Values have been normalized to nonstimulated control cells and shown as mean ± SEM. ***, $P = 0.0008$ (*Vav2*[Onc]), $P = 0.0001$ (*Vav2* $KO_3$), and $P = 0.0004$ (*Vav2* $KO_4$) using two-tailed Student's *t* tests ($n = 3$ independent experiments). As comparative control, we included data from nonstimulated (left) and stimulated (right) cells ($n = 2$). Source data for this figure are provided as a Source data file. In panels **e** and **h**, aliquots from the same lysates were analyzed in separate blots (each identified with asterisks of the same color).

EGFP-Vav2[WT], leads to the translocation of the mCherry-Akt PH to the plasma membrane in the absence of insulin stimulation (Fig. 4a). This effect is catalysis-dependent, since EGFP-Vav2[Onc + E200A] cannot trigger the translocation of the bioreporter to the plasma membrane (Fig. 4a). Consistent with this, we observed that the ectopic expression of an EGFP fused to a constitutively active version of Rac1 (Q61L mutant) also promotes the translocation of the bioreporter to the plasma membrane (Fig. 4a). As in the case of Vav2[Onc], the Rac1[Q61L]-mediated translocation of the mCherry-Akt PH is further increased upon the stimulation of the transiently transfected cells with insulin (Fig. 4a). The behavior exhibited by Rac1[Q61L] in these experiments is similar to that found in both basal and insulin-stimulated L6 myoblasts[44]. These experiments also revealed that the EGFP-Vav2[Onc], EGFP-Vav2[Onc+E200A], and EGFP-Rac1[Q61L] proteins display a plasma membrane localization in cells (Fig. 4a). We obtained similar results when an independent EGFP-Akt PH bioreporter was transiently expressed in C2C12 cells stably expressing HA-Vav2[Onc] and HA-Vav2[Onc+E200A] (Fig. 4b, c).

The translocation of the bioreporter induced by both insulin and EGFP-Vav2[Onc] is abolished when C2C12 cells are treated with pan-specific (Wortmannin) and α-isoform-specific (PIK-75) PI3K inhibitors (Fig. 4b, c). By contrast, it is not affected by a PI3Kβ-specific inhibitor (TGX-221) (Fig. 4b, c). In agreement with the foregoing data, we found that Wortmannin also eliminates both the Akt and S6K phosphorylation in insulin-treated control and Vav2[Onc]-expressing C2C12 cells (Fig. 4d). These inhibitors do no block the membrane localization typically displayed by Vav2[Onc] in both nonstimulated and insulin-stimulated C2C12 cells (Fig. 4e, f), indicating that PI3Kα is not upstream of Vav2 in this pathway. Taken together, these results indicate that the active versions of both Vav2 and Rac1 can trigger the stimulation of PI3Kα and downstream targets per se. This basal stimulation, however, is not sufficient to induce the stable activation of the pathway in the absence of upstream stimulation by insulin.

**Vav2 regulates the PI3Kα-Akt pathway via Rac GTPases.** Given that Vav2 can stimulate both Rac subfamily GTPases and RhoA, we carried out further experiments to assess the implication of the most classical Rho GTPases in the Vav2-dependent responses in C2C12 cells. We found that, similarly to Rac1, the active version of the Rac-related RhoG protein can also promote the translocation of the mCherry-Akt PH bioreporter to the plasma membrane (Fig. 5a). Since RhoG cannot bind the downstream serine/threonine protein kinases of the Pak family[45], these results suggest that the activation of the PI3Kα-Akt pathway does not entail the recently described scaffolding function of those kinases[46,47].

This activity is Rac subfamily-specific, since the active versions of RhoA and Cdc42 cannot translocate the mCherry-Akt PH fusion protein to the plasma membrane in any of the experimental conditions tested (Fig. 5a). Consistent with these data, we found that the ectopic expression of EGFP-Rac1[Q61L] (Fig. 5b) and EGFP-RhoG[Q61L] (Fig. 5c) enhances the phosphorylation levels of Akt and S6K in insulin-stimulated C2C12 cells. These results are consistent with previous reports indicating that Rac1 and RhoG, but not RhoA and Cdc42, can bind to and promote the activation of PI3K in other cell types[48].

Further confirming the implication of Rac1 in this Vav2[Onc]-regulated pathway, we found that both the shRNA-mediated depletion of endogenous Rac1 (Fig. 5d and Supplementary Fig. 7a) and the chemical inhibition of the Vav2[Onc]-Rac1 interaction using the 1A-116 compound[49,50] (Fig. 5e) reduce the insulin-mediated stimulation of the PI3Kα-Akt pathway in control and Vav2[Onc]-expressing C2C12 cells. Interestingly, we found that the total depletion of endogenous Rac1 leads to the unexpected hyperstimulation of the insulin pathway in those cells (Supplementary Fig. 7b, c). This suggests the activation of compensatory mechanisms when the signaling from this GTPase is totally abrogated in cells. We found similar compensatory loops when PI3Kα is removed from cells (I.F.-P. and X.R.B., unpublished data). Consistent with the implication of Rac1 in Vav2 signaling, we also found reduced levels of activation of the endogenous GTPase in the case of insulin-stimulated skeletal muscle from *Vav2*[L332A/L332A] mice (Fig. 5f, g). This defect is not seen in the case of insulin-stimulated WAT (Supplementary Fig. 7d) and liver (Supplementary Fig. 7e) from those mice. Collectively, these results suggest that Vav2 and Rac1 are primarily involved in the stimulation of the PI3Kα–Akt axis in skeletal muscle cells. This idea is further reinforced by the observation that Vav2[Onc], but not the Vav2[Onc + E200A] mutant, can promote the translocation of the glucose Glut4 transporter both in nonstimulated and insulin-stimulated C2C12 cells (Supplementary Fig. 8a). It has been shown before that the translocation of this transporter is Rac1-dependent[22–25,51,52]. The lack of insulin dependency of Vav2[Onc] in this regulatory step is consistent with previous results using constitutively active versions of either Rac1 or other upstream GEFs such as Tiam1 and Plekhg4 (also known as FLJ00068 and puratrophin-1)[22,24,44,51,53,54]. We also found that the depletion of endogenous Vav2 leads to a delay in the transfer of Glut4 from the cytosolic reservoirs to the plasma membrane of C2C12 cells, further highlighting the connection of Vav2 with downstream Rac1 signaling in those cells (Supplementary Fig. 8c, d).

**Regulation of the PI3Kα–Akt axis by Vav2 is Pak-independent.** Previous evidence indicates that the Rac1-mediated activation of

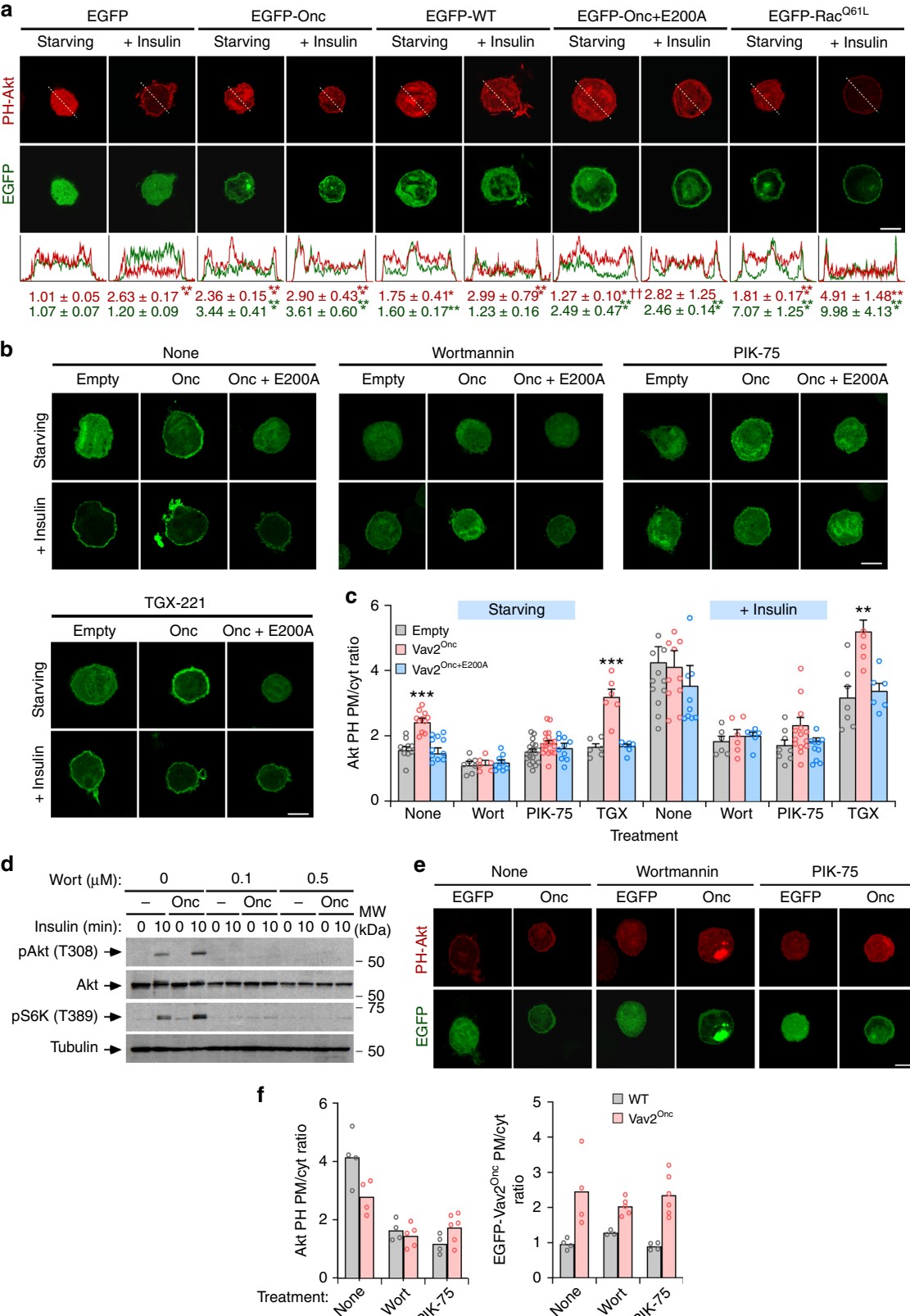

the PI3K-Akt pathway can involve either noncatalytic functions of Pak family kinases or the F-actin cytoskeleton in a number of cell types[46–48,55,56]. The results obtained with RhoG, a Rac1-like GTPase that cannot engage Pak kinases[45], suggest that the former mechanism is not involved in the regulation of this pathway in C2C12 cells (Fig. 5a, c). To assess the specific role of the

cytoskeleton, we treated control and Vav2$^{Onc}$-expressing C2C12 cells with the F-actin disrupting compounds latrunculin A and cytochalasin D. The former drug blocks F-actin formation due to its binding to free actin monomers[57,58]. The latter one binds to the growing ends of F-actin cables, thus preventing both the incorporation and release of G-actin molecules[59]. We observed

**Fig. 4 PI3Kα acts downstream of Vav2 during insulin signaling. a** Subcellular localization of mCherry-Akt PH (top panels, red color) and indicated EGFPs (bottom panels, green color) in parental C2C12 cells under indicated culture conditions. The fluorescence profile for each picture is shown at the bottom. The quantification of the ratio between plasma membrane and cytoplasm for each ectopically expressed protein is expressed as mean ± SEM (bottom). *, $P = 0.0125$ (WT − insulin, PH-Akt) and $P = 0.0118$ (Onc + E200A − insulin, PH-Akt); **, $P = 0.0019$; ***, $P = 0.0002$ (Onc − insulin, EGFP and Onc + insulin, PH-Akt), $P = 0.0005$ (Onc + E200A − insulin and Rac1$^{Q61L}$ − insulin and Rac1$^{Q61L}$ + insulin, EGFP), $P = 0.00005$ (EGFP + insulin, PH-Akt), $P = 0.0007$ (Rac1$^{Q61L}$ − insulin and WT + insulin, PH-Akt), $P = 0.0001$ (Rac1$^{Q61L}$ + insulin, PH-Akt), $P = 0.000002$ (Onc − insulin, PH-Akt), and $P < 0.000001$ (Onc + E200A + insulin, EGFP) relative to starved cells transfected with the EGFP empty vector using two-tailed Student's $t$ tests. ††, $P = 0.0015$ relative to starved cells expressing the EGFP-Vav2$^{Onc}$ vector using two-tailed Student's $t$ tests ($n = 3$ independent experiments). Scale bar, 10 μm. **b** Subcellular distribution of EGFP-Akt PH in C2C12 cells stably expressing the indicated proteins and under the specified culture conditions (top). Scale bar, 10 μm ($n = 3$ independent experiments). **c** Percentage of EGFP-Akt PH present at the plasma membrane (PM) and cytosol (Cyt) in the indicated cells and experimental conditions according to data from **b**. Wort Wortmannin; TGX TGX-221. Data are shown as mean ± SEM. **, $P = 0.0014$; ***, $P = 0.00002$ (Vav2$^{Onc}$ vs. Empty, None) and $P < 0.000001$ (Vav2$^{Onc}$ vs. Empty, TGX) relative to the nonstimulated (left) and stimulated (right) control using two-way ANOVA and Holm–Sidak multiple comparison tests ($n = 3$ independent experiments). **d** Phosphorylation and total protein levels of specified proteins in indicated cell lines (top) upon insulin stimulation (top) in the presence of specified amounts of Wortmannin (top). $n = 2$ independent experiments. **e** Subcellular distribution of mCherry-PH-Akt when coexpressed with either EGFP (Empty) or EGFP-Vav2$^{Onc}$ (Onc) (top) in insulin-stimulated cells treated with the indicated inhibitors for an hour (top). $n = 2$ independent experiments. Scale bar, 10 μm. **f** Plasma membrane (PM)/cytosolic (Cyt) ratio of mCherry-Akt PH (left) and EGFP-Vav2$^{Onc}$ (right) in the indicated cells (inset) and experimental conditions (bottom) according to data from **f**. Data are shown as mean ($n = 2$ independent experiments). Source data for this figure are provided as a Source data file.

that any of those two treatments eliminates the Vav2$^{Onc}$-driven translocation of the mCherry-Akt PH reporter to the plasma membrane (Fig. 6a). However, these treatments also eliminate the localization of Vav2$^{Onc}$ at the plasma membrane. This suggests that the F-actin cytoskeleton works upstream of Vav2$^{Onc}$ in this pathway, probably by ensuring the stable association of the protein in membranes. Unlike the case of Vav2$^{Onc}$, the F-actin-disrupting agents do not abrogate the translocation of the mCherry-Akt PH bioreporter induced by constitutively-active Rac1 (Fig. 6b). To further follow-up this result, we resorted to the use of two mutant versions of active Rac1 that cannot stimulate specific downstream signaling branches (Fig. 6c)[20,60]. Rac1$^{Q61L + F37A}$ can bind Pak and stimulate c-Jun N-terminal kinase (JNK), but cannot trigger membrane ruffling or G1 cell cycle transitions (Fig. 6c). By contrast, Rac1$^{Q61L + Y40C}$ is Pak-binding-deficient, and cannot stimulate JNK, but it does promote F-actin polymerization (Fig. 6c). Those two mutants trigger levels of the mCherry-Akt PH fusion protein to the plasma membrane comparable to those found with Rac1$^{Q61L}$ (Fig. 6d). These results suggest that the Vav2–Rac1 axis promotes the stimulation of the PI3Kα-Akt pathway using both Pak family- and F-actin-independent mechanisms (Fig. 6e). They also indicate that actin plays a role in the stabilization of Vav2$^{Onc}$ at the plasma membrane (Fig. 6e).

**Vav2 affects the short-term glucose responses in mice.** Skeletal muscle is a major insulin-responsive organ and the main site of glucose disposal. As a result, alterations in either muscle mass and/or responsiveness to insulin action can lead to the development of type 2 diabetes both in mice and humans[1,2,61,62]. This led us to test glucose homeostasis in Vav2$^{L332A/L332A}$ and Vav2$^{Onc/Onc}$ mice maintained under both chow (CD) and high-fat (HFD) feeding conditions. Under both diets, we found that Vav2$^{L332A/L332A}$ mice react to the infusion of a bolus of glucose with both higher peaks and longer duration of hyperglycemia elevations of glucose in plasma than controls at all the ages tested (Fig. 7a, b), indicating that they are glucose intolerant. Vav2$^{L332A/L332A}$ mice also become more resistant to insulin than controls according to both insulin tolerance (Fig. 7c–e) and hyperglycemic–euglycemic clamp (Fig. 7f and Supplementary Fig. 9a) analyses. This defect is more pronounced under HFD than under CD feeding contexts (Fig. 7, compare panels c and e). Despite this insulin resistance, the CD-fed Vav2$^{L332A/L332A}$ mice only exhibit a mild hyperglycemia when they become older than 8 months (Fig. 7g). The plasma levels of insulin are also WT-like at

least until the last time-point analyzed (seventh month of age) (Supplementary Fig. 9b). These results indicate that despite the problems in glucose tolerance and insulin resistance that are already detected in young animals, the CD-fed Vav2$^{L332A/L332A}$ mice must have compensatory mechanisms that maintain physiological levels of plasma glucose until they are 8 months old. Conversely, we found that Vav2$^{Onc/Onc}$ mice show better glucose tolerance than controls under both CD and HFD feeding conditions (Fig. 7h, i). The animals fed with CD (Fig. 7j) and HFD (Fig. 7k) also show slightly better transient responses to insulin than controls. We did not find any defects in insulin production in Vav2$^{L332A/L332A}$ or Vav2$^{Onc/Onc}$ mice either (Supplementary Fig. 9c, d).

Further analyses of glucose metabolism indicated that 3.5-month-old Vav2$^{L332A/L332A}$ mice show normal basal endogenous glucose production (a parameter mainly accounted for by hepatic glucose production)[63] (Supplementary Fig. 9e). Yet, the ability of insulin to suppress endogenous glucose production by the liver was impaired in Vav2$^{L332A/L332A}$ mice (Supplementary Fig. 9e). This result is not in agreement with the lack of alteration in the insulin-mediated stimulation of the PI3Kα-Akt pathway found in that tissue in Vav2$^{L332A/L332A}$ animals (Supplementary Fig. 3e, f). However, it is worth noting that previous studies have shown that this biological readout is not a good indicator of the actual insulin sensitivity of this tissue[64,65]. In fact, similar defects in the suppression of hepatic glucose production were found in loss-of-function mouse models for signaling elements of the insulin and IGF1 pathways in both skeletal muscle (IGF1, Akt family) and WAT (PI3Kα)[16,66,67]. Our analyses also indicated that Vav2$^{L332A/L332A}$ animals exhibit lower rates of insulin-induced glucose clearance than controls (Supplementary Fig. 9f), suggesting a defect in peripheral glucose uptake. Despite this, all the tissues tested, including a number of different skeletal muscle types, display normal levels of insulin-induced uptake of circulating glucose (Supplementary Fig. 9g, h). These results suggest that the insulin signaling dysfunctions found in the skeletal muscle of Vav2$^{L332A/L332A}$ mice are compensated by other mechanisms intrinsic or extrinsic to myocytes[68]. Similar results have been found before using loss-of-function mouse models for a large variety of insulin and IGF1 signaling elements, including receptors, PI3K regulatory subunits, and PI3Kα itself[9,12,13,42]. Given the importance of the skeletal muscle in glucose homeostasis, both in terms of proportional mass and clearance capacity (10-fold higher than WAT according to our data in Supplementary Figure 9g, h), the glucose clearance defect found in Vav2$^{L332A/L332A}$ mice is

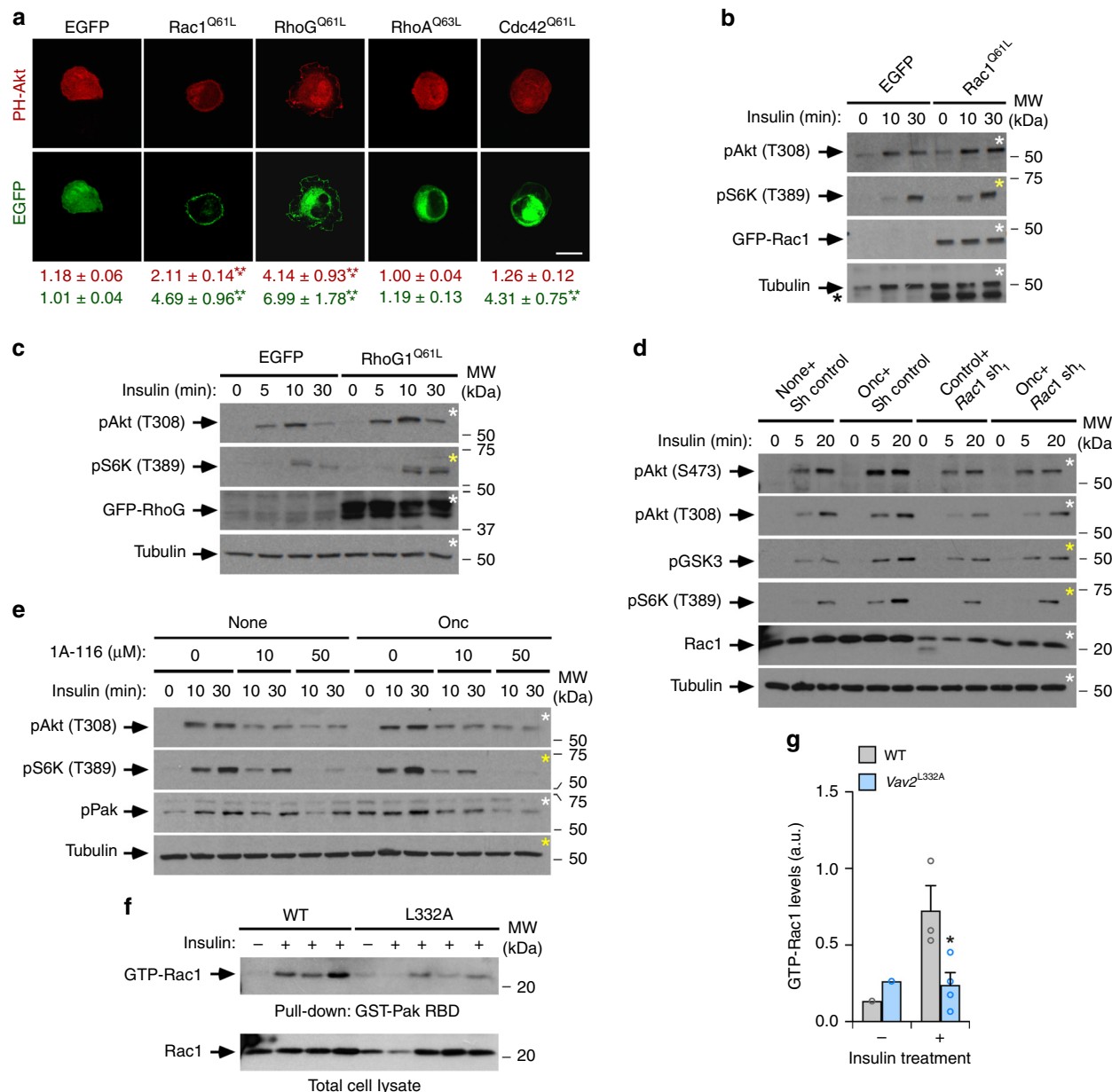

**Fig. 5 Vav2 regulates the PI3Kα-Akt pathway in a Rac-dependent manner. a** Representative image showing the subcellular localization of mCherry-Akt PH in starved, nonstimulated C2C12 cells transiently transfected with the indicated proteins (top). The quantification of the plasma membrane (PM) to cytoplasm (Cyt) ratio of the mCherry-Akt PH bioreporter (red) and the indicated EGFP-GTPase (green) is shown at the bottom. Values represent the mean ± SEM. ***, $P = 0.000006$ (Rac1$^{Q61L}$, EGFP), $P = 0.0008$ (Rac1$^{Q61L}$, PH-Akt), $P = 0.00007$ (RhoG$^{Q61L}$, EGFP), $P = 0.000005$ (RhoG$^{Q61L}$, PH-Akt), and $P = 0.000006$ (Cdc42$^{Q61L}$, EGFP) relative to the cells transfected with an empty vector using a Kruskal–Wallis test and two-sided Dunn's multiple comparison tests ($n = 3$ independent experiments). Scale bar, 10 μm. **b**, **c** Representative immunoblots of the phosphorylation levels of indicated proteins and phosphosites (left) in cells transiently expressing the specified EGFPs (top) upon insulin stimulation (top). The asterisks show the remaining EGFP-Rac1 signal that was carried over from the previous immunoblot analysis of the same filter. **d** Phosphorylation levels of the specified phosphosites and proteins (left) in the indicated insulin-stimulated C2C12 cells. Sh shRNA. **e** Representative immunoblot showing the levels of indicated phosphorylated sites and total proteins (left) in the specified cell lines that were stimulated with insulin for the indicated periods and either in the absence or presence of the indicated amounts of the 1A-116 inhibitor (top). $n = 3$ independent experiments. **f** GTP-bound levels of Rac1 in the skeletal muscle from 3-month-old mice of indicated genotypes (top) that were infused with insulin as indicated (top panel). As a control, aliquots of the same total cell lysates were analyzed in parallel (bottom panel). RBD Rac-binding domain. **g** Quantification of the fraction of GTP-bound Rac1 found in the experiments shown in **f**. Data are shown as mean ± SEM. *, $P = 0.0338$ relative to insulin-stimulated WT mice using two-tailed Student's $t$ tests. $n = 3$ (WT) and 4 ($Vav2^{L332A/L332A}$) insulin-stimulated animals. Source data for this figure are provided as a Source data file.

probably explained by the reduced muscle mass present in those animals (Fig. 1a).

**Vav2 influences other metabolic-related processes in mice.** The foregoing data, together with the increased fat content found in

$Vav2^{L332A/L332A}$ mice (Fig. 1a), led us to analyze the status of the WAT and liver in those animals under both chow and HFD feeding conditions. In the former case, we found that $Vav2^{L332A/L332A}$ mice have overall lower weight than controls between 2 and 4 months of age (Fig. 8a). However, this difference

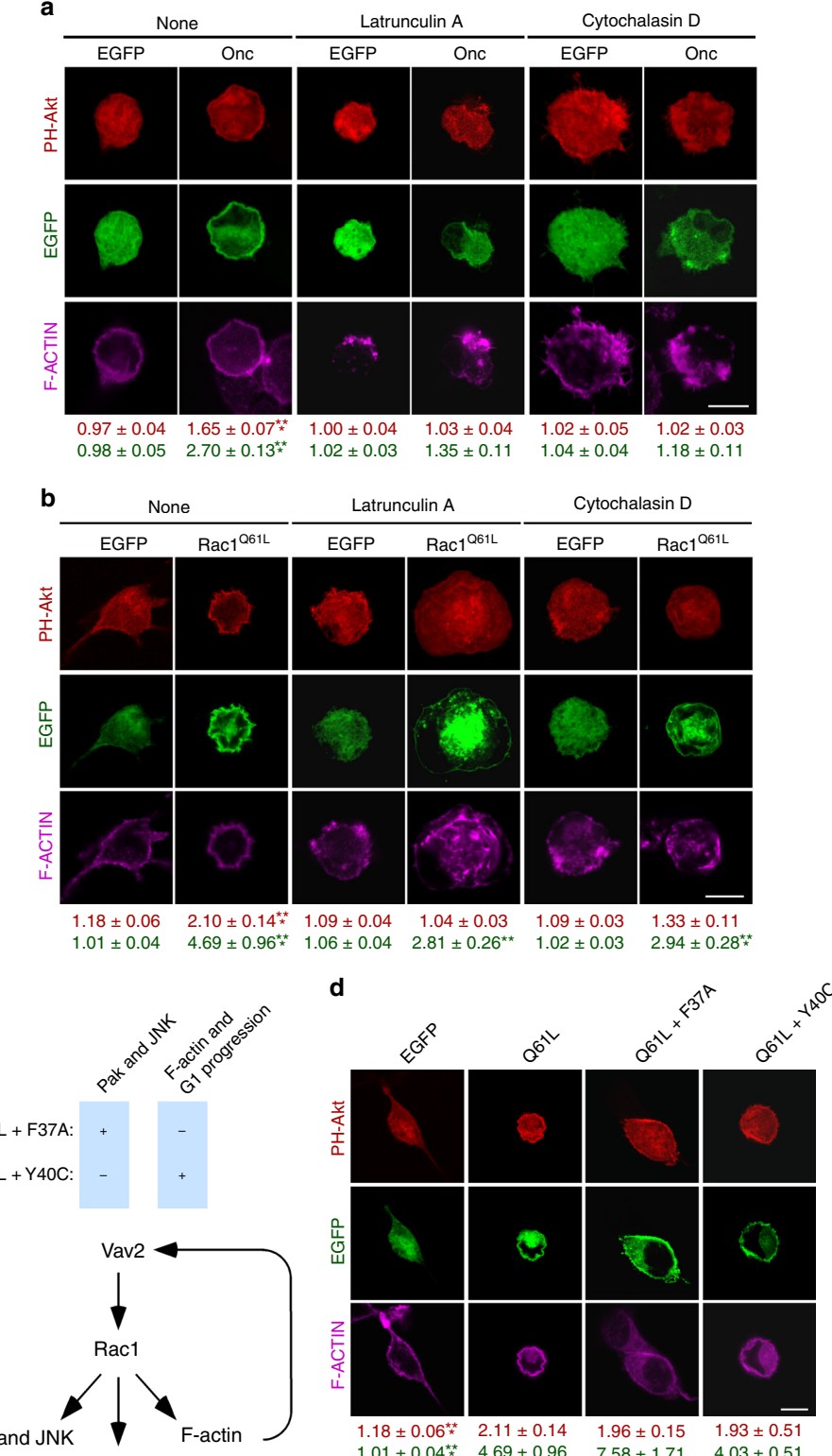

disappears later on and resurfaces again in older animals (Figs. 1a and 8a), probably as a consequence of the progressive accumulation of WAT both in the perigonadal and subcutaneous fat pads of those animals (Figs. 1a and 8b, c). This adiposity is associated with the progressive hypertrophy of white adipocytes in those tissues (Fig. 8d, e). However, we could not see any alterations in the levels of mRNAs encoding key metabolic enzymes in this tissue in the case of 4-month-old $Vav2^{L332A/L332A}$ mice

(Supplementary Fig. 10a). In the context of HFD feeding, we observed that $Vav2^{L332A/L332A}$ mice gain more weight than controls from the beginning of the diet change (Fig. 8a). This is linked to the accumulation of WAT (Fig. 8b, f) and a slight hypertrophy of the resident white adipocytes (Fig. 8g, h). Further metabolic determinations indicated that the increase of WAT in CD-fed 5-month-old $Vav2^{L332A/L332A}$ mice is not associated with changes in food intake (Supplementary Table 2), body

**Fig. 6 The activation of the PI3Kα-Akt pathway by Vav2 is F-actin-dependent and Pak family-independent. a**, **b** Representative images of the subcellular localization of the bioreporter mCherry-Akt PH in starved C2C12 cells expressing the indicated EGFP-tagged proteins (top) and treated with the actin-depolymerizing drugs shown on top. The quantification of membrane-translocated mCherry-Akt PH (red) and GFP-protein (green) are shown at the bottom. Values are represented as mean ± SEM. **, $P = 0.0015$; ***, $P = 0.0007$ (cytochalasin D-treated, $Rac1^{Q61L}$-transfected cells), and $P < 0.000001$ (rest of analyses) relative to untreated cells transfected with an empty vector using two-way ANOVA and Holm–Sidak's multiple comparison tests ($n = 3$ independent experiments). Scale bar, 10 μm. **c** Signaling properties of the indicated Rac1 switch mutant proteins. +, activation; −, lack of activation. **d** Representative image showing the subcellular localization of mCherry-Akt PH in starved, nonstimulated C2C12 cells transiently transfected with the indicated EGFPs (top). Data are shown as in **a**. ***, $P = 0.00004$ (EGFP) and $P = 0.000003$ (PH-Akt) relative to cells expressing $Rac1^{Q61L}$ using a Kruskal–Wallis test and two-sided Dunn's multiple comparison tests ($n = 3$ independent experiments). Scale bar, 10 μm. **e** Schematic representation of the Vav2-Rac1 signaling pathway in muscle cells according to the data obtained in Figs. 4–6. Source data for this figure are provided as a Source data file.

temperature (Supplementary Table 2), locomotor activity (Supplementary Table 2), or energy expenditure when corrected by their percentage of lean mass (Fig. 8i, j). However, we did observe a reduced respiratory quotient in these mice during the light exposure period relative to controls (Fig. 8k). This indicates that these animals predominantly use fat rather than carbohydrates as a fuel source during this period of the day. Most of those metabolic parameters are maintained within WT-like levels in 12-month-old $Vav2^{L332A/L332A}$ animals with the single exception of a statistically significant increase in overall food intake (Supplementary Table 2).

The overall weight of CD-fed $Vav2^{Onc/Onc}$ mice is similar to that displayed by control animals (Fig. 9a). These mice also contain similar content of gonadal WAT than controls (Fig. 9b, c). However, they do show smaller white adipocytes in average (Fig. 9d, e). These adipocytes show WT-like levels of transcripts for key metabolic enzymes when interrogated in 4-month-old mice (Supplementary Fig. 10b). Under HFD, $Vav2^{Onc/Onc}$ mice gain less weight (Fig. 9a) and develop less WAT (Fig. 9b, f) than controls. This is associated with a reduction in the overall size of the white adipocytes when compared to WT mice (Fig. 9g, h). Metabolic analyses indicated that CD-fed 3-month-old $Vav2^{Onc/Onc}$ mice show a slight increase in overall food intake (Supplementary Table 2) and energy expenditure (Fig. 9i). However, these parameters become WT-like when they are normalized according to the lean mass content present in them (Supplementary Table 2 and Fig. 9j). These animals also show an elevation of BAT temperature at this age (Supplementary Table 2). By contrast, they show normal rectal temperature and locomotor activity (Supplementary Table 2). All these parameters become WT-like in 10- and 12-month-old $Vav2^{Onc/Onc}$ mice (Supplementary Table 2). Opposite to the phenotype of $Vav2^{L332A/L332A}$ mice (Fig. 8k), we also found that the 3-month-old $Vav2^{Onc/Onc}$ mice display increased respiratory quotients during the light exposure period (Fig. 9k). This indicates that the preferred fuel in these mice comes from carbohydrate sources rather than from fat. Such a shift in respiratory quotients has been observed in other mouse models with skeletal muscle hypertrophy[15,17].

The liver of $Vav2^{L332A/L332A}$ animals under CD feeding conditions do not show any change in weight when compared to controls (Supplementary Fig. 11a). However, they show histological signs of incipient steatosis (Supplementary Fig. 11b). Moreover, we observed a sharp increase in the triglyceride content in the liver of 6-month-old $Vav2^{L332A/L332A}$ mice (Supplementary Fig. 11c). This fits the time in which these animals exhibit more gonadal fat (Fig. 8c). This phenotype is further aggravated in HFD-fed $Vav2^{L332A/L332A}$ mice (Supplementary Fig. 11d), leading to increases in liver weight (Supplementary Fig. 11e) and steatosis (Supplementary Fig. 11f) when compared to WT animals. We did not see any significant alteration either in the weight, histology, or triglyceride content of the liver of $Vav2^{Onc/Onc}$ mice under CD conditions (Supplementary Fig. 11g–i). Comparable liver parameters are also observed in

HFD-fed $Vav2^{Onc/Onc}$ mice, with the only exception of reduced liver weights (Supplementary Fig. 11j–l).

Despite the above-mentioned metabolic changes found in WAT and liver, we did not observe any statistically significant variation in the serum levels of cholesterol and triglycerides in $Vav2^{L332A/L332A}$ (Supplementary Fig. 12a, b) and $Vav2^{Onc/Onc}$ (Supplementary Fig. 12c, d) mice under CD conditions when compared to controls. However, under HFD, these animals exhibited higher and lower levels of cholesterol in serum, respectively (Supplementary Fig. 12a, c). We also found the progressive accumulation of triglycerides in the skeletal muscle of 10-month-old (Supplementary Fig. 12e) and HFD-fed (Supplementary Fig. 12f) $Vav2^{L332A/L332A}$ mice. Such alterations are not detected in younger animals under a CD regimen (Supplementary Fig. 12e). All those changes, therefore, take place at later times than the signaling and histological alterations found in the muscle of those two mouse strains. No variations in triglyceride content were observed in the skeletal muscle from $Vav2^{Onc/Onc}$ animals (Supplementary Fig. 12g, h).

In addition to normal insulin signaling in the liver, we found that other biological responses associated with this tissue are normal in both $Vav2^{L332A/L332A}$ and $Vav2^{Onc/Onc}$ mice. Thus, these animals show WT-like responses when using a diet lacking methionine and choline, a classical dietary regimen to induce nonalcoholic fatty liver disease (Supplementary Figs. 13 and 14). $Vav2^{L332A/L332A}$ mice also display WT-like responses upon the administration of tunicamycin (Supplementary Fig. 15), a yeast antibiotic that promotes endoplasmic reticulum stress by blocking N-glycosylation in hepatocytes. Finally, we found that metformin, a drug that primarily acts by reducing glucose production by the liver[69], does not ameliorate the phenotype of $Vav2^{L332A/L332A}$ mice (Supplementary Fig. 16). Taken together, these results suggest that the metabolic alterations found in adipose tissue and liver with deregulated catalytic activity of Vav2 are downstream effects of the initial alterations seen in the skeletal muscle.

**Vav2 catalytic output influences BAT status.** Given the impact of Vav2 catalytic output in WAT content and BAT temperature (in the case of $Vav2^{Onc/Onc}$ mice), we decided to investigate the status of BAT in both $Vav2^{L332A/L332A}$ and $Vav2^{Onc/Onc}$ mice under both CD and HFD conditions. We found that CD-fed $Vav2^{L332A/L332A}$ mice display alterations in the histology and function of this tissue, including reductions in the cell density of brown adipocytes (Fig. 10a, b) and the increase in the size of lipid droplets present in those cells (Fig. 10a, c). The increase in lipid content is the earliest dysfunction emerging in $Vav2^{L332A/L332A}$ mice (Fig. 10c). These histological alterations mimic, although in a milder fashion, the changes seen in the BAT of WT mice under HFD feeding conditions (Fig. 10d–g). Despite these changes, we could not see any statistically significant variation of Ucp1 protein in the BAT from 4-month-old $Vav2^{L332A/L332A}$ mice (Supplementary Fig. 17a) or in the basal oxygen consumption rate

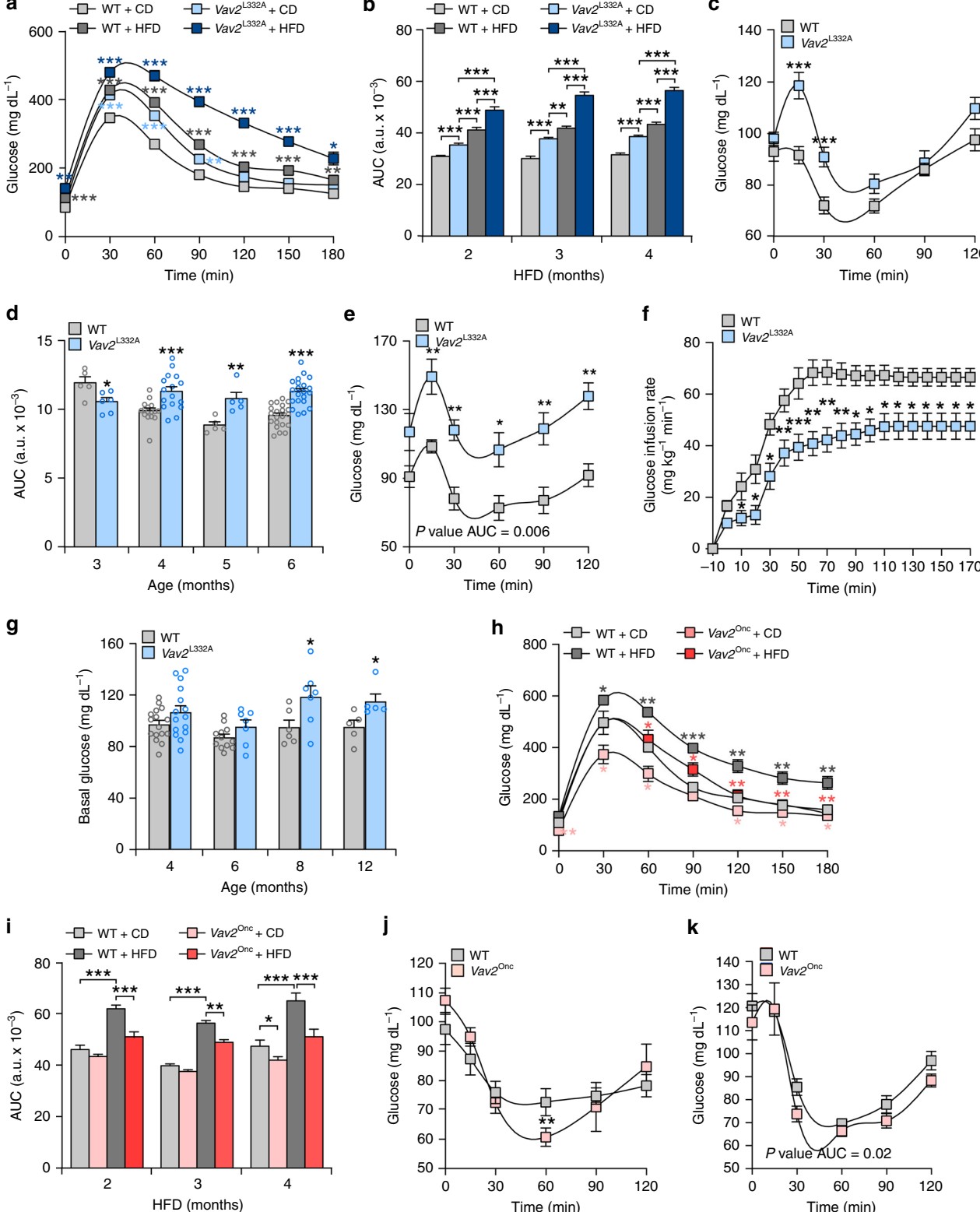

(Supplementary Fig. 17b) and the number of mitochondria (Supplementary Fig. 17c) in the brown adipocytes from 3.5-month-old $Vav2^{L332A/L332A}$ animals. These mice also respond with a progressive drop of body temperature similar to WT controls when subjected to low environmental temperature (Supplementary Fig. 17d). The BAT phenotype displayed by $Vav2^{L332A/L332A}$ mice is further exacerbated when subjected to an HFD (Fig. 10d–g).

$Vav2^{Onc/Onc}$ mice display the opposite phenotype in BAT under the two diet conditions examined (Fig. 10h–n). As in the former case, we could not observe any significant alteration in basal consumption rates (Supplementary Fig. 17b) and mitochondrial content (Supplementary Fig. 17c) in young animals. Their thermogenic response to low temperature exposure is also similar to that found in controls (Supplementary Fig. 17d). The kinetics of these alterations are also similar to those found in

**Fig. 7 Alterations in Vav2 catalytic activity alter short-term responses to glucose. a** Response of 6-month-old mice of the indicated genotypes and diet conditions (inset) to the infusion of a bolus of glucose. The HFD was maintained for 4 months. CD chow diet. Data represent the mean ± SEM. *, $P =$ 0.0291; **, $P =$ 0.0023 (HFD- vs. CD-fed $Vav2^{L332A/L332A}$ mice, 0 min) and $P =$ 0.0021 (CD-fed $Vav2^{L332A/L332A}$ mice, 90 min); ***, $P =$ 0.0003 (HFD-fed WT mice, 0 min), $P =$ 0.00009 (HFD-fed WT mice, 120 min), $P =$ 0.0004 (HFD-fed WT mice, 150 min), $P =$ 0.00001 (CD-fed $Vav2^{L332A/L332A}$ mice, 30 min), $P =$ 0.0005 (HFD-fed $Vav2^{L332A/L332A}$ mice, 30 min), and $P <$ 0.000001 (rest of analyses) relative to chow-fed WT animals (in the case of CD-fed $Vav2^{L332A/L332A}$ and HFD-fed WT mice) or HFD-fed WT animals (in the case of experiments with HFD-fed $Vav2^{L332A/L332A}$ animals) using two-way ANOVA and Holm–Sidak multiple comparison tests. $n =$ 14 (CD-fed WT) and 15 (others) mice per group from three independent experiments. **b** Areas under the curve (AUC) obtained in the glucose tolerance tests carried out in mice of the indicated genotypes (inset), diet conditions (inset), and ages (bottom). Data represent the mean ± SEM. **, $P =$ 0.0041; ***, $P =$ 0.00009 (CD-fed $Vav2^{L332A/L332A}$ vs. HFD-fed WT mice, 4 months), 0.0002 (CD-fed $Vav2^{L332A/L332A}$ vs. CD-fed WT mice, 2 months), 0.000004 (CD-fed $Vav2^{L332A/L332A}$ vs. HFD-fed WT mice, 2 months) and $P <$ 0.000001 (rest) relative to the control at each time-point using two-way ANOVA and Holm–Sidak multiple comparison tests ($n =$ 15 mice per group and genotype utilized in three independent experiments). **c** Response of CD-fed 4-month-old mice of the indicated genotypes to insulin. Data represent the mean ± SEM. ***, $P =$ 0.00007 (15 min) and 0.0009 (30 min) relative to the value control at each time-point using two-tailed Student's $t$ tests. $n =$ 16 (WT) and 15 ($Vav2^{L332A/L332A}$) animals used in three independent experiments. **d** AUC responses of CD-fed mice of indicated genotypes (inset) and ages (bottom) to the infusion of insulin. Data are presented as mean ± SEM. *, $P =$ 0.0271; **, $P =$ 0.006; ***, $P =$ 0.0009 (4-month-old mice) and $P <$ 0.000001 (6-month-old mice) relative to the values obtained in the appropriate control samples using two-tailed Student's $t$ tests. $n =$ 5 (3-month-old WT and 5-month-old), 6 (3-month-old $Vav2^{L332A/L332A}$), 16 (4-month-old) and 21 (6-month-old) mice. **e** Response of HFD-fed 6-month-old mice of the indicated genotypes maintained for 4 months in HFD to the infusion of insulin. Data are shown as mean ± SEM. *, $P =$ 0.0255; **, $P =$ 0.0058 (15 min), $P =$ 0.0022 (30 min), $P =$ 0.0091 (90 min), and $P =$ 0.0020 (120 min) relative to the value obtained in control samples in same time-point using two-tailed Student's $t$ tests ($n =$ 5 mice per experimental condition). **f** Glucose infusion rates found in 3-month-old animals of the indicated genotypes. Data are shown as mean ± SEM. *, $P =$ 0.0312 (10, 110, 120, 130, 140, 150, 160, and 170 min), $P =$ 0.0231 (20 and 30 min), $P =$ 0.0157 (90 min), and $P =$ 0.0236 (100 min); **, $P =$ 0.0028 (40 min), $P =$ 0.0013 (60 min), $P =$ 0.0047 (70 min), and $P =$ 0.0082 (80 min); ***, $P =$ 0.0006 relative to value obtained in control samples in same time-point using two-tailed Student's $t$ tests. $n =$ 6 (WT) and 8 ($Vav2^{L332A/L332A}$) mice. **g** Basal levels of glucose in CD-fed animals of the indicated genotypes (inset) and ages (bottom) that were fasted overnight. Data are shown as mean ± SEM. *, $P =$ 0.0465 (8-month-old mice) and 0.0352 (12-month-old mice) relative to control samples using two-tailed Student's $t$ tests. $n =$ 5 (12-month-old), 6 (8-month-old WT), 7 (6- and 8-month-old $Vav2^{L332A/L332A}$), 12 (6-month-old WT), 15 (4-month-old $Vav2^{L332A/L332A}$), and 16 (4-month-old WT) mice. **h** Representative response of 6-month-old mice of indicated genotypes and diet conditions (inset) to the infusion of glucose. Data are shown as mean ± SEM. *, $P =$ 0.0142 (HFD-fed WT mice, 30 min), $P =$ 0.0244 (CD-fed $Vav2^{Onc/Onc}$ mice, 30 min), $P =$ 0.05 (CD-fed $Vav2^{Onc/Onc}$ mice, 60 min), $P =$ 0.0211 (CD-fed $Vav2^{Onc/Onc}$ mice, 120 min), $P =$ 0.0497 (CD-fed $Vav2^{Onc/Onc}$ mice, 150 min), $P =$ 0.0136 (CD-fed $Vav2^{Onc/Onc}$ mice, 180 min), $P =$ 0.0244 (HFD-fed $Vav2^{Onc/Onc}$ mice, 60 min), and $P =$ 0.0254 (HFD-fed $Vav2^{Onc/Onc}$ mice, 90 min); **, $P =$ 0.0057 (HFD-fed WT mice, 60 min), $P =$ 0.0026 (HFD-fed WT mice, 120 min), $P =$ 0.0029 (HFD-fed WT mice, 150 min), $P =$ 0.0037 (HFD-fed WT mice, 180 min), $P =$ 0.0013 (CD-fed $Vav2^{Onc/Onc}$ mice, 0 min), $P =$ 0.0087 (HFD-fed $Vav2^{Onc/Onc}$ mice, 120 min), $P =$ 0.0068 (HFD-fed $Vav2^{Onc/Onc}$ mice, 150 min), and $P =$ 0.0029 (HFD-fed $Vav2^{Onc/Onc}$ mice, 180 min); ***, $P =$ 0.0007 relative to either CD-fed WT animals (in the case of CD-fed $Vav2^{Onc/Onc}$ and HFD-fed WT mice) or HFD-fed WT animals (in the case of HFD-fed $Vav2^{Onc/Onc}$ mice) using two-way ANOVA and Holm–Sidak multiple comparison tests ($n =$ 5 animals per experimental group). **i** AUC responses of mice of indicated genotypes (inset), ages (bottom), and diet conditions (inset) to the infusion of glucose. Data represent the mean ± SEM. *, $P =$ 0.0337; **, $P =$ 0.007; ***, $P =$ 0.0001 (HFD-fed $Vav2^{Onc/Onc}$ mice, 2 months) and $P <$ 0.000001 (rest of analyses) relative to the control at each time-point using two-way ANOVA and Holm–Sidak multiple comparison test. $n =$ 15 (HFD-fed $Vav2^{Onc/Onc}$, CD-fed WT controls for 2 and 4 months and CD-fed $Vav2^{Onc/Onc}$, 4 months), 16 (HFD-fed, 2 months), 17 (HFD-fed WT, 3 and 4 months), and 18 (CD-fed, 3 months) animals per group. **j** Representative response of CD-fed 4-month-old mice of indicated genotypes (inset) to the infusion of insulin. Data represent the mean ± SEM. **, $P =$ 0.0092 relative to values obtained in control samples in the same time-point and normalized to basal using two-tailed Student's $t$ tests. $n =$ 7 (WT) and 6 ($Vav2^{Onc/Onc}$) mice. **k** Response of 6-month-old mice of indicated genotypes (inset) and maintained under HFD for 4 months to the infusion of insulin. Data are shown as mean ± SEM. $n =$ 8 (WT) and 5 ($Vav2^{Onc/Onc}$) mice. Source data for this figure are provided as a Source data file.

$Vav2^{L332A/L332A}$ mice (Fig. 10b, c). However, further experiments indicated that molecular features associated with thermogenic BAT activity become upregulated in 6-month-old mice (Supplementary Fig. 17e–h). Those include transcripts encoding Ucp1 (Supplementary Fig. 17e), Hsl (Supplementary Fig. 17f), Pgc1α (peroxisome proliferator-activated receptor γ coactivator 1α) (Supplementary Fig. 17g) and Prdm16 (PRD1-BF1-RIZ1 homologous-domain-containing 16) (Supplementary Fig. 17h). This is specific, since other mRNAs for BAT thermogenic and developmental factors (Cidea, Pparγ, Cebpα) do not become deregulated in $Vav2^{Onc/Onc}$ mice (Supplementary Fig. 17i–k). This phenomenon can contribute to the changes in body fat content and body weight detected in animals older than 6 months.

## Discussion

In this work, we have reported the identification of a hitherto unknown Vav2-dependent regulatory layer that contributes to fine-tune the signaling output of the PI3Kα–Akt axis in the skeletal muscle upon exposure to IGF1 and insulin. This regulatory layer is highly dependent on the catalytic activity of this GEF. Thus, we have found that mice carrying a Vav2 mutant protein with reduced catalytic activity exhibit impaired IGF1- and insulin-mediated PI3Kα signaling, reduced muscle mass, and increased glucose intolerance (Supplementary Fig. 18a). Given that these mice still keep 30% of the enzyme activity typically exhibited by Vav2[37], we predict that the actual contribution of the Vav2 pathway to the PI3Kα signaling output must be even larger in the skeletal muscle. Conversely, $Vav2^{Onc/Onc}$ mice display the opposite changes in the foregoing physiological parameters (Supplementary Fig. 18a). These signaling alterations can be recapitulated ex vivo using both loss- and gain-of-function approaches in C2C12s cells, indicating that Vav2 specifically affects the overall signaling output of PI3Kα in insulin-stimulated myocytes. Further linking the connection of Vav2 with the skeletal muscle, we have found impaired and enhanced basal proliferation rates in the satellite cells from $Vav2^{L332A/L332A}$ and $Vav2^{Onc/Onc}$ mice, respectively (Supplementary Fig. 18a). The satellite cells from the latter animals also show higher expression levels of differentiation-associated genes (Supplementary Fig. 18a), a feature that has been reproduced when tested in C2C12 cells ectopically expressing Vav2^Onc. Unlike the rest of

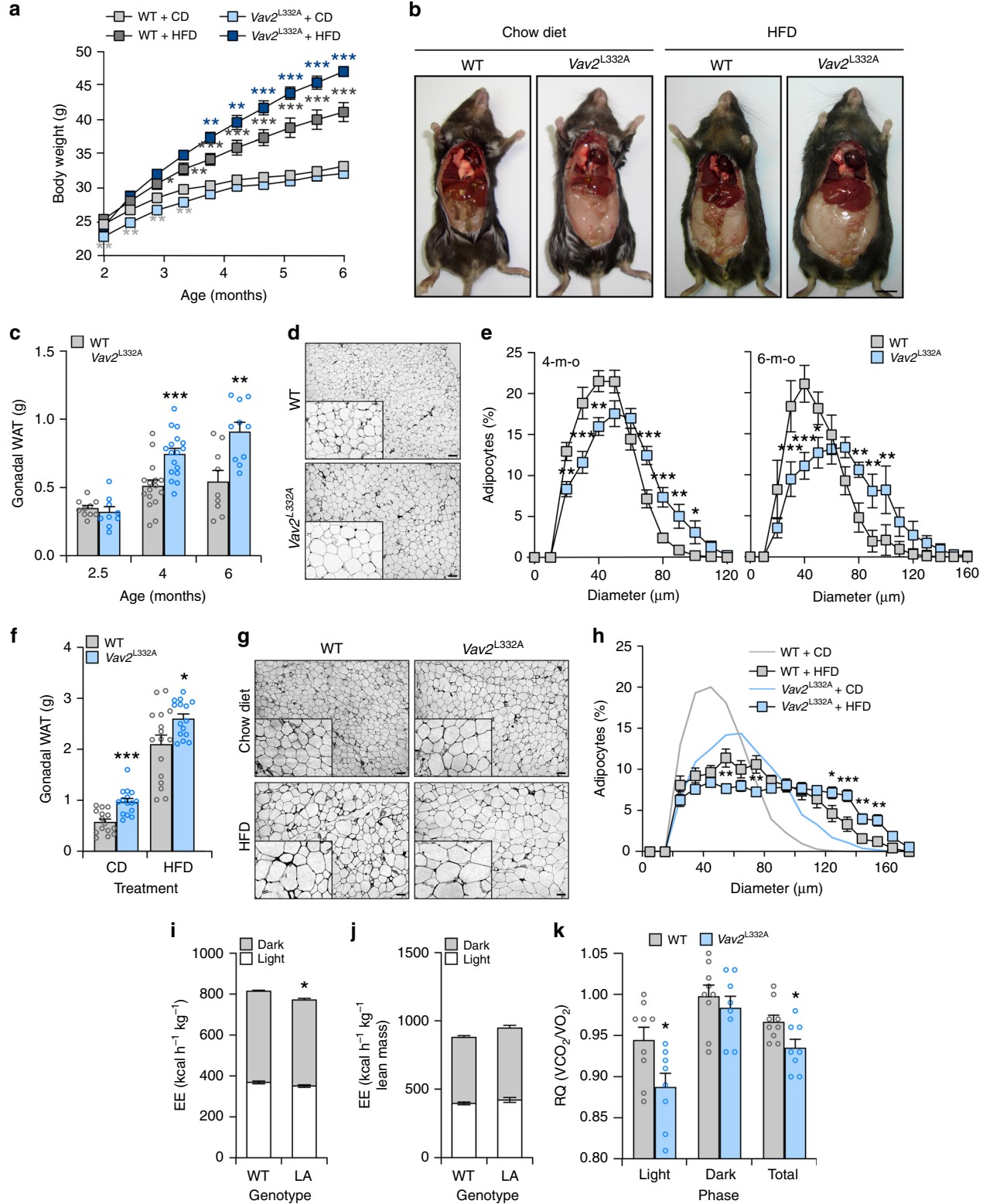

parameters discussed above, we could not find any overt defect in the differentiation kinetics of $Vav2^{L332A/L332A}$ satellite cells (Supplementary Fig. 18a). This suggests that other compensatory, Vav2-independent mechanisms might exist to maintain the basal proliferation of primary satellite cells in the absence of Vav2. Alternatively, as in the case of the Vav2-dependent pathways previously found in keratinocytes and smooth muscle cells[37], it is possible that the residual Vav2 GEF activity that must persist in

$Vav2^{L332A/L332A}$ mice could be sufficient to maintain this proliferative response.

Using reporters to monitor the activation status of PI3K and PI3K isoform-specific inhibitors, we found that $Vav2^{Onc}$ can promote the specific stimulation of PI3Kα in C2C12 cells in an insulin-independent but catalysis-dependent manner in transiently transfected C2C12 cells. These results are consistent with the idea that $Vav2^{Onc}$ behaves as a constitutively active,

**Fig. 8 Reduced Vav2 catalytic activity leads to increased fat content in white adipocytes. a** Evolution of the body weight of mice of indicated genotypes (inset) that were maintained under the indicated diet conditions (inset) from the 8th to the 26th week of age. Data represent the mean ± SEM. *, $P =$ 0.0198; **, $P = 0.0013$ (HFD-fed WT vs. CD-fed WT, 16 weeks), $P = 0.0075$ (HFD-fed $Vav2^{L332A/L332A}$ vs. WT mice, 16 weeks), $P = 0.0014$ (HFD-fed $Vav2^{L332A/L332A}$ vs. WT mice, 18 weeks), $P = 0.0064$ (CD-fed $Vav2^{L332A/L332A}$ vs. WT mice, 8 weeks), $P = 0.0029$ (CD-fed $Vav2^{L332A/L332A}$ vs. WT mice, 10 weeks), $P = 0.0050$ (CD-fed $Vav2^{L332A/L332A}$ vs. WT mice, 12 weeks), and $P = 0.0067$ (CD-fed $Vav2^{L332A/L332A}$ vs. WT mice, 14 weeks); ***, $P = 0.00006$ (HFD-fed vs. CD-fed WT mice, 18 weeks), $P = 0.0001$ (HFD-fed $Vav2^{L332A/L332A}$ vs. WT mice, 20 weeks), $P = 0.000007$ (HFD-fed $Vav2^{L332A/L332A}$ vs. WT mice, 22 weeks), $P = 0.000001$ (HFD-fed $Vav2^{L332A/L332A}$ vs. WT mice, 24 weeks), and $P < 0.000001$ (rest of analyses) relative to the value obtained with the respective control at the same time-point using two-way ANOVA and Holm–Sidak multiple comparison tests and two-tailed Student's $t$ tests in the case of the light gray asterisks. $n = 14$ (CD-fed WT) and 15 (rest) animals per group used in three independent experiments). **b** Representative image of mice at the end of the experiments shown in **a**. Scale bar, 1 cm. **c** Weight of the gonadal WAT mass from CD-fed animals of indicated genotypes (inset) and ages (bottom). Data are presented as mean ± SEM. **, $P = 0.0032$; ***, $P = 0.0004$ relative to control at each age point using two-tailed Student's $t$ tests. $n = 9$ (2-month-old $Vav2^{L332A/L332A}$ and 6-month-old WT mice), 10 (2-month-old WT and 6-month-old $Vav2^{L332A/L332A}$ mice), 16 (4-month-old WT mice), and 17 (4-month-old $Vav2^{L332A/L332A}$ mice). **d** Representative images of gonadal WAT sections from CD-fed 6-month-old mice of indicated genotypes (left). Scale bar, 100 µm. $n = 9$ (WT) and 10 ($Vav2^{L332A/L332A}$) mice. **e** Distribution of the mean diameter of gonadal white adipocytes from CD-fed 4- (left) and 6-month-old (right) mice of indicated genotypes (inset). Data are shown as mean ± SEM. *, $P =$ 0.0160 (50 µm, right panel) and $P = 0.0482$ (100 µm, left panel); **, $P = 0.0015$ (20 µm, left panel), $P = 0.0064$ (40 µm, left panel), $P = 0.0044$ (90 µm, left panel), $P = 0.0092$ (80 µm, right panel), and $P = 0.0064$ (90 and 100 µm, right panel); ***, $P = 0.0001$ (30 µm, left panel), $P = 0.0003$ (70 µm, left panel), $P = 0.0006$ (80 µm, left panel), $P = 0.00008$ (30 µm, right panel), and $P = 0.00001$ (40 µm, right panel) relative to the value obtained with the respective control at the same time-point using two-way ANOVA, followed by Fisher's LSD tests. $n = 16$ (4-month-old WT mice), 13 (4-month-old $Vav2^{L332A/L332A}$ mice), 9 (6-month-old WT mice), and 10 (6-month-old $Vav2^{L332A/L332A}$ mice). **f** Weight of the gonadal WAT mass from 6-month-old mice of indicated genotypes (inset) that were subjected to either CD or HFD (bottom) for 4 months. Data are presented as mean ± SEM. *, $P = 0.02$; ***, $P = 0.0001$ relative to the value obtained with the respective control using two-tailed Student's $t$ tests ($n = 15$ animals per experimental group). **g** Representative images of gonadal WAT sections from 6-month-old mice of indicated genotypes (top) that were subjected to either CD or HFD conditions (left). Scale bar, 100 µm. $n = 15$ animals per group. **h** Distribution of the mean diameter of gonadal white adipocytes from 6-month-old mice of the indicated genotypes that were maintained under CD (light gray and blue lines, quantified in **e**) and HFD (black lines with boxes) conditions. Data are presented as mean ± SEM. *, $P = 0.0471$; **, $P = 0.0019$ (140 µm), $P = 0.005$ (150 µm), $P = 0.006$ (50 µm), and $P = 0.009$ (70 µm); ***, $P = 0.0004$ relative to the value obtained with the respective control at the same time-point using two-way ANOVA, followed by Fisher's LSD tests ($n = 15$ animals per experimental group). **i**, **j** Energy expenditure (EE) corrected by total body weight (**i**) and lean mass (**j**) exhibited by CD-fed 5-month-old WT and $Vav2^{L332A/L332A}$ (LA) animals (bottom) during the indicated light cycle periods (inset). Data are shown as mean ± SEM. *, $P = 0.0206$ using two-way ANOVA and Holm–Sidak multiple comparison tests. $n = 12$ (WT) and 11 ($Vav2^{L332A/L332A}$) mice. **k** Respiratory quotient (RQ) of CD-fed 5-month-old animals of the indicated genotypes (inset) during the indicated light cycle periods (bottom). $VCO_2$ volume of $CO_2$, $VO_2$ volume of oxygen. Data represent the mean ± SEM. *, $P = 0.0253$ (light) and 0.0282 (total) using two-tailed Student's $t$ tests. $n = 9$ (WT) and 8 ($Vav2^{L332A/L332A}$) mice. Source data for this figure are provided as a Source data file.

phosphorylation-independent GEF due to the removal of its N-terminal autoinhibitory domains[31,33]. In agreement with this catalysis-dependent process, we have observed that the constitutively active Rac1$^{Q61L}$ and RhoG$^{Q61L}$ proteins can also trigger the translocation of the Akt PH-based bioreporters to the plasma membrane. This pathway seems to be Rac subfamily-dependent, since chronically activated versions of RhoA and Cdc42 cannot trigger the translocation of Akt to the plasma membrane under the same experimental conditions. Consistent with the foregoing observations, we have found that the depletion of endogenous Rac1 using shRNA interference methods impairs the stimulation of the PI3K-Akt pathway in insulin-stimulated parental and Vav2$^{Onc}$-expressing C2C12 cells. Same results were obtained when using a chemical inhibitor that specifically disrupts the Vav2–Rac1 but not the Vav2–RhoA or the Vav2–Cdc42 interaction. Despite these data, the single expression of Vav2$^{Onc}$ is not self-sufficient to promote the stimulation of the PI3K pathway in both skeletal muscle and stably transfected C2C12 cells. This suggests that the stimulation of PI3Kα and PIP$_3$ production by Vav2$^{Onc}$ is very short-lived in the absence of upstream stimulation. Alternatively, the activity of Vav2$^{Onc}$ could be counterbalanced by the long-term development of a negative feedback mechanism. It is important to note, however, that this issue is merely academic because the nonphosphorylated Vav2$^{WT}$ protein must be catalytically silent in nonstimulated cells and, therefore, should only become catalytically active upon cell stimulation. Unlike the case of the activation of the PI3Kα–Akt axis, we have observed that ectopically expressed Vav2$^{Onc}$ can promote the translocation of the Glut4 transporter to the plasma membrane in the absence of insulin stimulation. These

results mimic the observations previously made with chronically activated versions of both Rac1 and other upstream GEFs[22,24,44,51,53,54]. They are also consistent with previous findings indicating that Rac1 and the PI3Kα–Akt axis contribute to the translocation of Glut4 using mechanistically independent pathways[22,51,52,70]. We have also found that the depletion of endogenous Vav2 delays the translocation of Glut4 in C2C12 cells. Despite those data, the relevance of this regulatory step in vivo is unclear given that $Vav2^{L332A/L332A}$ and $Vav2^{Onc/Onc}$ mice do not exhibit overt alterations in glucose uptake. It is possible, however, that this function of Vav2 could be redundantly performed by other Rac1 GEFs present in the skeletal muscle. Based on the current data, the most likely candidate for this is Plekhg4[53].

Unlike our current data and results from other studies using active Rac1 mutants[44,47], the analysis of skeletal muscle-deficient Rac1 knock-in mice has not detected any defects in the phosphorylation levels of Akt during insulin signaling[23,24,70]. This discrepancy can be due to the fact that we have analyzed several stimulation time-points rather than a single one as in those studies. Alternatively, the Rac1-deficient mice could have developed compensatory mechanisms to overcome the total depletion of Rac1 activity. Consistent with this latter idea, we have observed that the total depletion of endogenous Rac1 leads to the hyperactivation of the PI3K-Akt pathway in insulin-stimulated C2C12 cells.

Recent results have shown that Rac1 can favor the stimulation of the PI3Kα–Akt axis via three, not mutually exclusive mechanisms: (i) The formation of multiprotein complexes nucleated by Pak kinases[46,47]. (ii) The polymerization of actin in

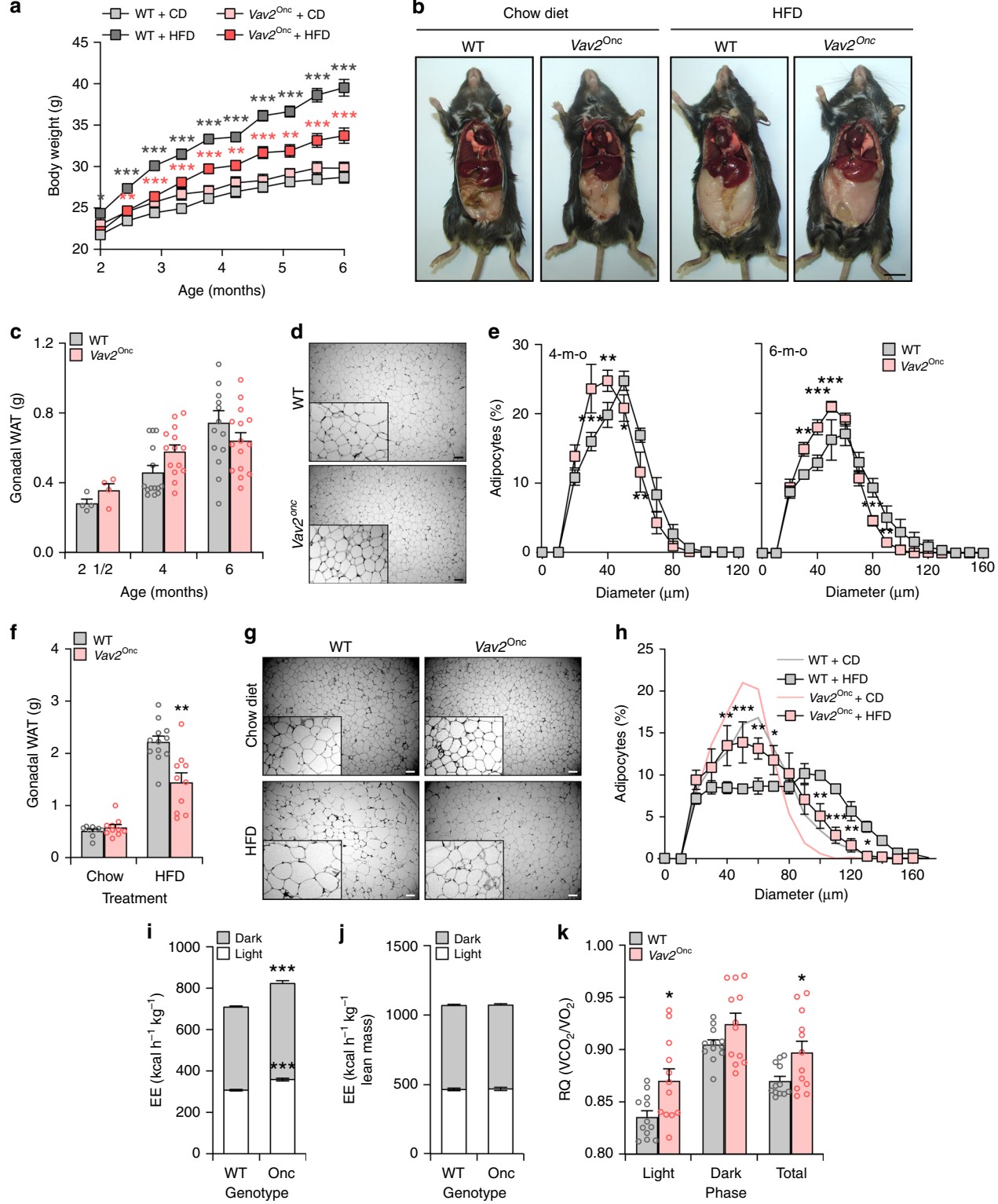

juxtamembrane areas of cells[48,55,56]. (iii) Direct interactions with PI3Kα signaling complexes[48,55]. The first of those models is not compatible with our data, given that RhoG[Q61L] and Rac1[Q61L+Y40C] can stimulate the PI3Kα-Akt pathway as efficiently as Rac1[Q61L] itself, despite lacking the ability to bind Pak family members[45,60]. Our observations also indicate that the cytoskeleton favors this signaling step, although from an upstream rather than a downstream position relative to Vav2. Based on the above, it is likely that Rac1, upon activation by

Vav2, will directly form complexes with PI3Kα as previously described in other cell types[48,55]. Whether such complexes entail direct physical contacts or the participation of bridge molecules such as the PI3K noncatalytic subunits remains to be determined[55]. Regardless of the mechanism involved, our protein depletion experiments support the idea that Rac1 is the main Vav2 catalytic substrate involved in this pathway. In line with this, it is worth noting that no skeletal muscle defects have been found so far in $Rhog^{-/-}$ mice[71–73].

**Fig. 9 Upregulated Vav2 catalytic activity reduces fat content in white adipocytes. a** Evolution of the body weight of mice of indicated genotypes (inset) that were maintained under the indicated diet conditions (inset) from the 8th to the 26th week of age. Data represent the mean ± SEM. *, $P = 0.0274$; **, $P = 0.0047$; ***, $P = 0.0001$ (WT HFD vs. WT CD mice, 10 weeks), $P = 0.00002$ ($Vav2^{Onc/Onc}$ HFD vs. WT HFD mice, 12 weeks), $P = 0.0001$ ($Vav2^{Onc/Onc}$ HFD vs. WT HFD mice, 14 and 18 weeks), $P = 0.00004$ ($Vav2^{Onc/Onc}$ HFD vs. WT HFD mice, 16 weeks), and $P < 0.000001$ (rest of analyses) relative to the value obtained with the respective control at the same time-point using two-way ANOVA and Holm–Sidak multiple comparison tests. $n = 10$ (CD) or 15 (HFD) animals per group used in two or three independent experiments. **b** Representative image of mice at the end of the experiments shown in **a**. Scale bar, 1 cm. **c** Weight of the gonadal WAT mass from CD-fed animals of indicated genotypes (inset) and ages (bottom). Data are presented as mean ± SEM. $n = 4$ (2.5-month-old mice), 19 (4-month-old mice, 14 (6-month-old WT animals), or 15 (6-month-old $Vav2^{Onc/Onc}$ mice). **d** Representative images of gonadal WAT sections from CD-fed 6-month-old mice of indicated genotypes (left). Scale bar, 100 μm. $n = 5$ (WT) and 6 ($Vav2^{Onc/Onc}$) mice. **e** Distribution of the mean diameter of gonadal white adipocytes from CD-fed 4- (left) and 6-month-old (right) mice of indicated genotypes (inset). Data are shown as mean ± SEM. *, $P = 0.0190$; **, $P = 0.0017$ (60 μm, left panel), $P = 0.0055$ (30 μm, right panel), $P = 0.0064$ (90 μm, right panel), and $P = 0.0035$ (40 μm, left panel); ***, $P = 0.000009$ (30 μm, left panel), $P = 0.0001$ (40 μm, right panel), and $P = 0.0003$ (50 and 80 μm, right panel) relative to the value obtained with the respective control at the same time-point using two-way ANOVA followed by Fisher's LSD tests. $n = 6$ (6-month-old $Vav2^{Onc/Onc}$) and 5 (other conditions) animals per experimental group. **f** Weight of the gonadal WAT mass from 6-month-old mice of indicated genotypes (inset) that were subjected to either CD or HFD (bottom) for 4 months. Data are presented as mean ± SEM. **, $P = 0.0013$ relative to the value obtained with the respective control using two-way ANOVA and Holm–Sidak multiple comparison tests. $n = 9$ (CD-fed WT), 10 (CD- and HFD-fed $Vav2^{Onc/Onc}$), and 12 (HFD-fed WT) mice. **g** Representative images of gonadal WAT sections from 6-month-old mice of the indicated genotypes (top) that were subjected to either CD or HFD conditions (left). Scale bar, 100 μm. $n = 4$ (CD) and 5 (HFD) animals per experimental group. **h** Distribution of the mean diameter of gonadal white adipocytes from 6-month-old mice of indicated genotypes that were maintained under CD (light gray and red lines, quantified in **e**) and HFD (black lines with boxes) conditions. Data are presented as mean ± SEM. *, $P = 0.0444$ (70 μm) and $P = 0.0239$ (130 μm); **, $P = 0.0014$ (40 μm), $P = 0.0043$ (60 μm), $P = 0.0019$ (100 μm), and $P = 0.0088$ (120 μm); ***, $P = 0.0004$ (110 μm) and $P = 0.0005$ (50 μm) relative to the value obtained with the respective control at the same time-point using two-way ANOVA, followed by Fisher's LSD tests. $n = 4$ (CD) or 5 (HFD) animals per experimental group. **i, j** Energy expenditure (EE) corrected by total body weight (**i**) and lean mass (**j**) exhibited by CD-fed 3-month-old WT and $Vav2^{Onc/Onc}$ (Onc) mice (bottom) during the indicated light cycle periods (inset). Data are shown as mean ± SEM. ***, $P = 0.000004$ (dark period) and 0.00007 (light period) using two-way ANOVA and Holm–Sidak multiple comparison tests ($n = 12$ animals per group). **k** RQ of CD-fed 3-month-old animals of the indicated genotypes (inset) during the indicated light cycle periods (bottom). Data are presented as mean ± SEM. *, $P = 0.0143$ (light period) and 0.0499 (total) using two-way ANOVA and Holm–Sidak multiple comparison tests ($n = 12$ animals per group). Source data for this figure are provided as a Source data file.

The Vav2-mediated regulation of the PI3Kα–Akt axis is skeletal muscle-specific, given that we could not detect any overt change in the levels of phosphorylation of Akt in other insulin-responsive tissues in the case of $Vav2^{L332A/L332A}$ mice. This is to some extent unexpected, given that Vav2 is expressed at high levels in the liver, WAT and BAT (Supplementary Table 1). It is likely that the role of Vav2 in these tissues could be redundant with other GEFs that are also expressed in those tissues such as P-Rex1 and P-Rex2 (liver), Vav3, Tiam1 and P-Rex family members (WAT), or Vav3 and P-Rex2 (BAT) (Supplementary Table 1). Consistent with this idea, it has been shown before that P-Rex2, a GEF for Rac1, is involved in the stimulation of the PI3K–Akt axis in white adipocytes[74]. However, given that $Vav2^{Onc/Onc}$ mice do enhance the insulin-mediated activation of Akt in WAT, we cannot exclude at present whether the remaining catalytic activity of $Vav2^{L332A}$ could suffice to ensure efficient insulin signaling in that tissue. It also remains to be explored whether Vav2 mediates the signaling output of the PI3Kα–Akt axis in other insulin-responsive tissues not tested in this work. In any case, the lack of hyperglycemia in young $Vav2^{L332A/L332A}$ mice argues against this possibility.

Despite the insulin and IGF1 signaling defects present in the skeletal muscle, the $Vav2^{L332A/L332A}$ mice can maintain an euglycemic state until they reach the eighth month of age. This feature has been previously seen in other loss-of-function models for insulin and IGF1 signaling elements, including PI3Kα itself[9,12,13,42]. It is possible that euglycemia could be maintained in those animals through the uptake of glucose in the highly expanded WAT tissue. Alternatively, the skeletal muscle of those animals could engage an as yet unidentified membrane receptor that could contribute to the regulation of glucose metabolism by forming complexes with either the insulin or IGF1 receptors. This idea is consistent, for example, with the observation that a dominant-negative mutant version of the IGF1 receptor, but not the concurrent elimination of both the insulin and IGF1

receptors, triggers hyperglycemia and hyperinsulinemia when expressed in the skeletal muscle[9]. Further work will be needed to unveil these compensatory mechanisms present in the skeletal muscle and, perhaps, other glucoregulatory tissues.

We have observed that $Vav2^{L332A/L332A}$ and $Vav2^{Onc/Onc}$ mice progressively develop mirror-image histological and functional alterations in the BAT, WAT, and liver (Supplementary Fig. 18a, b). These physiological alterations, which take place at older ages than the skeletal muscle dysfunctions, recapitulate well those extensively seen in other mouse models with either reduced (in the case of $Vav2^{L332A/L332A}$ mice) or increased (in the case of $Vav2^{Onc/Onc}$ mice) skeletal muscle mass[12,13,15,17,18,68,75]. Many dysfunctions found in $Vav2^{L332A/L332A}$ mice also resemble those found in HFD-fed WT mice (this work), further suggesting that they are caused by the progressive accumulation of body fat indirectly caused by the early alterations in muscle mass and function detected in those mice. According to our data and previous publications[12,13,15,17,18,68,75], we believe that the most plausible explanation for all those late dysfunctions is that they are side effects of the signaling alterations present in the skeletal muscle of both $Vav2^{Onc/Onc}$ and $Vav2^{L332A/L332A}$ mice. However, we caution the readers that we cannot formally exclude the possibility that they could be the consequence of the alteration of some intrinsic functions of Vav2 in other peripheral tissues given that we have not utilized skeletal muscle-specific knock-in animals in our work.

$Vav2^{L332A/L332A}$ and $Vav2^{Onc/Onc}$ mice exhibit lower and higher respiratory quotients during the light cycle than control animals, respectively (Supplementary Fig. 18a, b). The metabolic feature found in $Vav2^{Onc/Onc}$ mice has also been observed in mouse models with increased skeletal muscle[15,17], suggesting that it is a direct consequence of the alterations seen in the skeletal muscle. In line with this, the preferred use of carbohydrates over fat as a fuel source in the resting period of these animals can be a consequence of the elevated availability of glycogen due to the

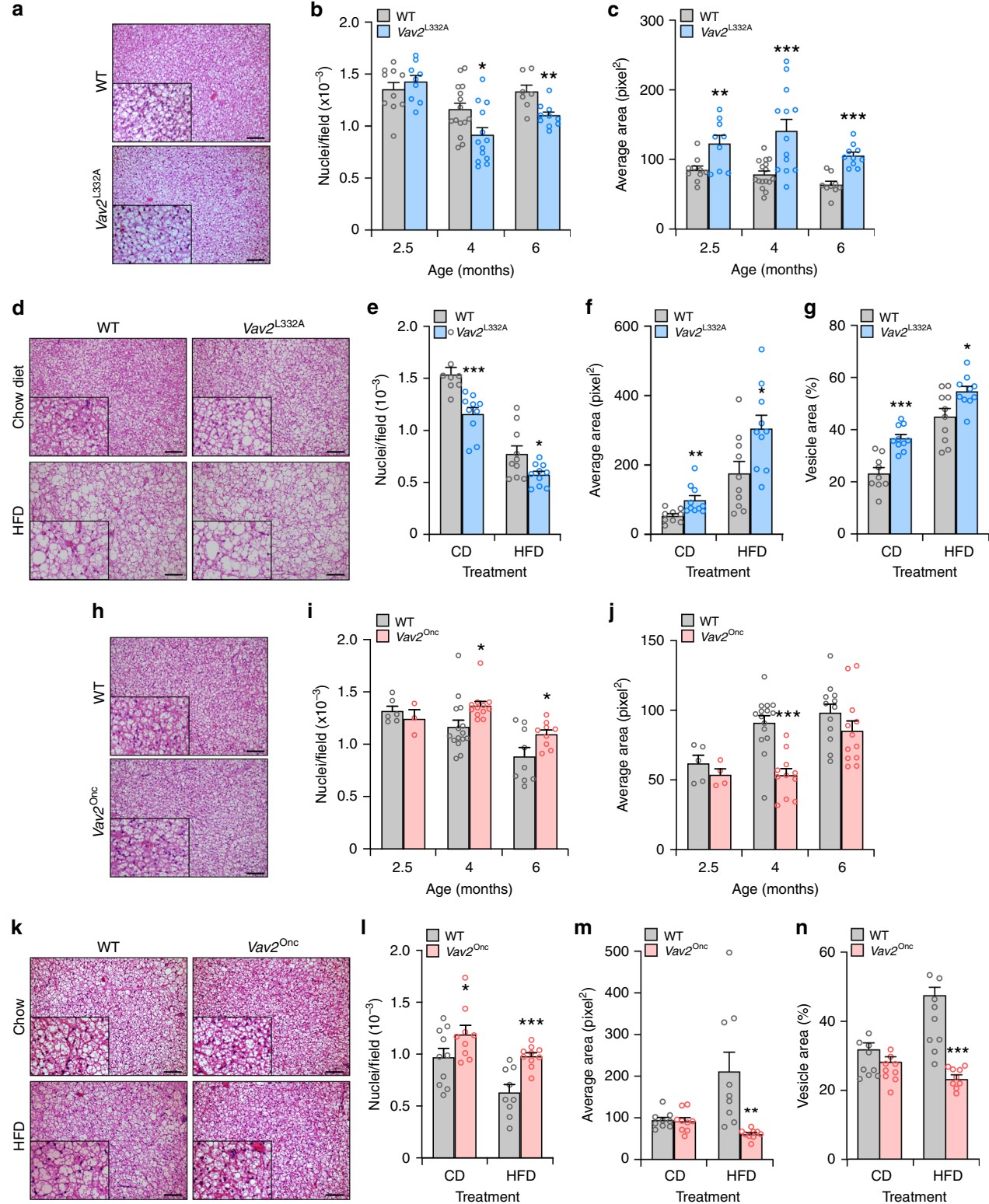

increased skeletal muscle mass and insulin responsiveness during the dark period of the day (the active period for these animals) as well as by the limited availability of fat due to the reduced WAT mass present in $Vav2^{Onc/Onc}$ mice. The reverse scenario can explain the preferred use of fat as fuel during the light period of the day in the case of $Vav2^{L332A/L332A}$ mice.

Finally, our work has also shed light into the issue of the Rho GEF catalytic thresholds that are required to maintain physiological homeostasis in Rho GTPase-dependent tissues. This issue is of paramount importance to determine the side effects that the therapeutic targeting of these Rho GEFs will cause in healthy tissues. In addition, it is important to predict the existence of therapeutic windows in which the positive effects of Rho GEF inhibitors could be dissociated from unwanted side effects. The $Vav2^{L332A}$ mouse strain has been developed to specifically address those issues, since the residual catalytic activity of

**Fig. 10 Vav2 catalytic output influences the fat content of brown adipocytes. a** Representative images of interscapular BAT sections from CD-fed 6-month-old mice of the indicated genotypes (inset). Scale bar, 100 μm. $n = 7$ WT and 10 $Vav2^{L332A/L332A}$ mice. **b** Number of interscapular BAT adipocytes in CD-fed animals of the indicated genotypes (inset) and ages (bottom). Data represent the mean ± SEM. *, $P = 0.0173$; **, $P = 0.0047$ relative to the value obtained with the respective control at the same time-point using two-tailed Student's t tests. $n = 10$ (2.5-month-old WT and 6-month-old $Vav2^{L332A/L332A}$ mice), 9 (2.5-month-old $Vav2^{L332A/L332A}$ mice), 7 (6-month-old WT mice), 15 (4-month-old WT mice), and 13 (4-month-old $Vav2^{L332A/L332A}$ mice). **c** Size of intracellular lipid droplets present in interscapular BAT adipocytes from CD-fed animals of the indicated genotypes (inset) and ages (bottom). Data represent the mean ± SEM. **, $P = 0.0081$; ***, $P = 0.0005$ (4-month-old mice) and $P = 0.00002$ (6-month-old animals) relative to the value obtained with the respective control at the same time-point using two-tailed Student's t tests. $n = 10$ (2.5-month-old WT and 6-month-old $Vav2^{L332A/L332A}$ mice), 9 (2.5-month-old $Vav2^{L332A/L332A}$ and 6-month-old WT mice), 16 (4-month-old WT mice), 13 (4-month-old $Vav2^{L332A/L332A}$ mice), and 13 (6-month-old $Vav2^{L332A/L332A}$ mice). **d** Representative images of interscapular BAT sections from CD- and HFD-fed 6-month-old animals of the indicated genotypes (top). Scale bar, 100 μm. $n = 9$ (CD-fed WT) and 10 (CD- and HFD-fed $Vav2^{L332A/L332A}$) mice. **e** Number of interscapular BAT adipocytes in 6-month-old animals of the indicated genotypes (inset) that were maintained under CD or HFD conditions (bottom) for 4 months. Data represent the mean ± SEM. *, $P = 0.0295$; ***, $P = 0.0008$ relative to the value obtained with the respective control using two-tailed Student's t tests. $n = 9$ (CD-fed WT) and 10 (CD- and HFD-fed $Vav2^{L332A/L332A}$) mice. **f** Size of intracellular lipid droplets present in interscapular BAT adipocytes from 6-month-old animals of the indicated genotypes (inset) that were maintained under CD or HFD conditions (bottom) for 4 months. Data represent the mean ± SEM. *, $P = 0.0216$; **, $P = 0.0077$ relative to the value obtained with the respective control using two-tailed Student's t tests. $n = 9$ (CD-fed WT) and 10 (CD- and HFD-fed $Vav2^{L332A/L332A}$) mice. **g** Percentage of BAT tissue occupied by lipids. *, $P = 0.0163$; ***, $P = 0.00008$ relative to the value obtained with the respective control using two-tailed Student's t tests. $n = 9$ (CD-fed WT) and 10 (CD- and HFD-fed $Vav2^{L332A/L332A}$) mice. **h** Representative images of interscapular BAT sections from CD-fed 6-month-old mice of the indicated genotypes (inset). Scale bar, 100 μm. $n = 9$ animals per group. **i** Number of interscapular BAT adipocytes in CD-fed animals of the indicated genotypes (inset) and ages (bottom). Data represent the mean ± SEM. *, $P = 0.0191$ (4-month-old mice) and 0.0406 (6-month-old mice); relative to the value obtained with the respective control at the same time-point using two-tailed Student's t tests. $n = 5$ (2.5-month-old WT mice), 4 (2.5-month-old $Vav2^{Onc/Onc}$ mice), 15 (4-month-old WT mice), 12 (4-month-old $Vav2^{Onc/Onc}$ mice), and 9 (6-month-old mice). **j** Size of their intracellular lipid droplets present in interscapular BAT adipocytes from CD-fed animals of the indicated genotypes (inset) and ages (bottom). Data represent the mean ± SEM. ***, $P = 0.00002$ relative to the value obtained with the respective control at the same time-point using two-tailed Student's t tests. $n = 5$ (2.5-month-old WT mice), 4 (2.5-month-old $Vav2^{Onc/Onc}$ mice), 15 (4-month-old WT mice), 11 (4-month-old $Vav2^{Onc/Onc}$ mice), or 12 (6-month-old mice). **k** Representative images of interscapular BAT sections from CD- and HFD-fed 6-month-old animals of the indicated genotypes (top). Scale bar, 100 μm. **l** Number of interscapular BAT adipocytes in 6-month-old animals of the indicated genotypes (inset) that were maintained under CD or HFD conditions (bottom) for 4 months. Data represent the mean ± SEM. *, $P = 0.0444$; ***, $P = 0.0008$ relative to the value obtained with the respective control using two-tailed Student's t tests ($n = 9$ animals per group). **m** Size of intracellular lipid droplets present in interscapular BAT adipocytes from 6-month-old animals of the indicated genotypes (inset) that were maintained under CD or HFD conditions (bottom) for 4 months. Data represent the mean ± SEM. **, $P = 0.0052$ relative to the value obtained with the respective control using two-tailed Student's t tests ($n = 9$ animals per group). **n** Percentage of BAT tissue occupied by lipids. ***, $P = 0.00008$ relative to the value obtained with the respective control using two-tailed Student's t tests ($n = 9$ mice per group). Source data for this figure are provided as a Source data file.

Vav2 still present in these mice mimics the effect of drugs that usually do not achieve the complete inhibition of their targets. In the case of Vav2, this genetic approach indicates that different tissues might have variable requirements for the minimal amount of GEF activity needed to maintain organ homeostasis. For example, $Vav2^{L332A/L332A}$ mice do not develop the hypertension and glaucoma problems that are detected in $Vav2^{-/-}$ mice[37–40], indicating that the cells associated with those pathologies can function well with 30% of the normal levels of Vav2 catalytic activity. By contrast, skeletal muscle requires higher levels of Vav2 catalytic output both to grow and to maintain proper insulin responsiveness (this work). The Vav2-dependent skin tumors also require catalytic thresholds >30% to develop[37]. Based on these data, we can predict that the use of inhibitors that will reduce 70% the catalytic activity of Vav2 could preserve normal blood and eye pressure homeostasis while being fully effective as anticancer compounds. However, our data also unveils that a potential Damocles' sword associated with the chronic administration of such compounds is the long-term development of skeletal muscle atrophy and type 2 diabetes. The combined use of $Vav2^{Onc/Onc}$ mice also allowed us to demonstrate that the catalytic activity of Vav2 has signaling autonomous roles in all the aforementioned biological programs and pathologies. Given the information obtained with these "mirror-image" mouse models, it would be interesting to apply similar genetic approaches to validate other disease-associated Rho GEFs as potential therapeutic targets.

## Methods

**Ethics statement**. All mouse experiments were performed according to protocols approved by the Bioethics Committees of the University of Salamanca and the

University of Santiago de Compostela. Care of mice and experiments done (euglycemic–hyperinsulinemic clamps) at the University of Geneva were within the procedures approved by the animal care and experimentation authorities of the Canton of Geneva, Switzerland. We have not utilized patients or patient-derived samples in this work.

**Animals**. The methods used for the generation of $Vav2^{Onc/Onc}$ knock-in (C57BL/6 background), $Vav2^{L332A/L332A}$ knock-in (mixed genetic background), and $Vav2^{-/-}$ knockout (C57BL/10 genetic background) mice have been described in previous publications[37,41,76]. These mice and genetic background-matched WT controls were housed with an artificial 12-h light /12-h dark cycle under controlled temperature (23 °C) and humidity (50%) conditions. They were maintained under ad libitum access to a standard chow global diet (Cat. #2018; Teklad global 18% protein) and tap water. When indicated, 8-week-old mice were shifted to either an HFD (45% fat, 4.73 kcal g$^{-1}$; Cat. #D12451; Research Diets) or a metformin regimen (2 mg mL$^{-1}$ in the drinking water; Cat. #PHR1084, Sigma-Aldrich) for 4 months before being euthanized. Endoplasmic reticulum stress was induced in the liver by injecting tunicamycin (2 μg kg$^{-1}$; Cat. #T7765, Sigma-Aldrich) intraperitoneally to 2-month-old mice. When indicated, 8-week-old mice were shifted to a methionine- and choline-deficient diet (Cat. #A02082002B, Research Diets) for the indicated period of time before being sacrificed. The experiments involving the in vivo infusion of insulin and IGF1 were performed indistinctly in age-matched male and female animals. The rest of the experiments were done using males.

**Cell lines**. C2C12 cells were obtained from P. Muñoz-Cánovas (University Pompeu-Fabra, Barcelona, Spain) and cultured in Dulbecco's modified Eagle's medium (DMEM) containing 10% fetal bovine serum, 1% L-glutamine, penicillin (10 μg mL$^{-1}$), and streptomycin (100 μg mL$^{-1}$), and maintained at 37 °C and a 5% $CO_2$ humidified atmosphere. All the reagents were obtained from Gibco. For cell stimulation studies, C2C12 cells were starved for 3 h and then stimulated with insulin (75 nM; CN #775502, Actrapid NovoNordisk) for the indicated times. When indicated, cells were treated for 1 h with the indicated concentration of Wortmannin (100 nM, Cat. #19545-26-7, Calbiochem), PIK-75 (200 nM, Cat. #372196-77-5, Cayman Chemical), TGX-221 (500 nM, Cat. #663619-89-4, Cayman Chemical), latrunculin A (200 nM, Cat. #L5163, Sigma-Aldrich) and cytochalasin

D (2 μM, Cat. #C8273, Sigma-Aldrich). When other concentrations of inhibitors were used, they are indicated in the appropriate figure. The 1A-116 inhibitor was kindly provided by P. Lorenzana Menna (CONICET and Quilmes University, Buenos Aires, Argentina)[49] and used at the indicated concentrations for 3 h. The Lenti-X 293T lentiviral packaging cell line was obtained from Clontech (Cat. #632180) and cultured as above.

**Mammalian expression vectors.** All the constructs used in this work encode the murine versions of the proteins and were DNA sequence-verified in our Genomics facility. Plasmids encoding EGFP-Vav2 (pAA7), EGFP-Rac1$^{Q61L}$ (pNM42), EGFP-RhoG$^{Q61L}$ (pVOS17), EGFP-RhoA$^{Q63L}$ (pNM041), EGFP-Cdc42$^{Q61L}$ (pNM040), EGFP-Rac1$^{Q61L+F37A}$ (pMJC6), and Rac1$^{Q61L+Y40C}$ (pMJC7) have already been described[77–79]. To generate the plasmid encoding EGFP-Vav2$^{Onc}$ (pNM115), the plasmid pKES19[31] was digested with BstXI, filled-in, and cloned into the SmaI-linearized pEGFP-C2 vector (Clontech, Cat. #632481). The plasmid encoding EGFP-Vav2$^{Onc+E200A}$ (pFLM07) was obtained by site-directed mutagenesis using the plasmid pNM115 as a template for the PCR, the high-fidelity Pfu Turbo DNA polymerase (Cat. #600250, Agilent), and the primers 5′-GAC AAG AGA AGC TGC TGC TTG TTA GCG ATT CAG GAG ACC GAG GCC AAG TAC-3′ and 5′-CAT GAA CCG GAG CCA GAG GAC TTA GCG ATT GTT CGT CGT CGA AGA GAA CAG-3′ (the altered nucleotides used to create the E200A mutation are underlined). The plasmids pMD-G and pNGVL-MLV-gag-pol were provided by Drs. R.C. Mulligan (Children's Hospital, Boston, MA, USA) and A. Bernard (CNIC, Madrid, Spain, EU), respectively. The pEGFP-C1-PH-Akt and pmCherry-C1-PH-Akt vectors were obtained from Dr. J.P. Simas (Lisbon University, Lisbon, Portugal)[80].

To generate the lentiviral vector encoding HA-tagged Vav2$^{Onc}$ (pCCM34), a cDNA fragment encoding HA-Vav2$^{Onc}$ was amplified by PCR using the plasmid pCMV-Vav2 HA as the template[80] and the primers 5′-AAT AAC TAG TGC CAC CAT GTA CCC ATA CGA CGT CCC AGA CTA CGC TAA AAT GGG AAT GAC TGA GGA CGA C-3′ and 5′-ATA GAC CGC GGC CGC TCA CTG GAT GCC CTC CTC TTC TAC GTA-3′ (restriction sites underlined). Upon purification and digestion with SpeI and NotI, the fragment was cloned into the SpeI-NotI-linearized pLVX-IRES-Hyg vector (Cat. #632185, Clontech). The pCCM34 vector was used as a template to generate the plasmid encoding HA-Vav2$^{Onc+E200A}$ (pFLM12) by site-directed mutagenesis using the primers 5′-GAC AAG AGA AGC TGC TGC TTG TTA GCG ATT CAG GAG ACC GAG GCC AAG TAC-3′ and 5′-CAT GAA CCG GAG CCA GAG GAC TTA GCG ATT GTT CGT CGT CGA AGA GAA CAG-3′ to introduce the mutation E200A (the altered nucleotides used to create the E200A mutation are underlined). Site-directed mutagenesis was performed using the high-fidelity NZYProof DNA polymerase (Cat. #14601, NZYTech).

**RNA isolation and quantitation.** Total RNA was isolated using NZYol (Cat. #MB18501, NZYtech) and analyzed by qRT-PCR using the Power SYBR Green RNA-to-CT™ 1-Step kit (Cat. #4389986, Applied Biosystems) and the StepOnePlus Real-Time PCR System (Cat. #4376600, Applied BioSystems) according to the supplier's instructions. Raw qRT-PCR data were processed with the StepOne software v2.1 (Applied Biosystems) using *Gapdh* as intersample normalization control. Primers used for transcript quantitation included 5′-AAG CCT GTG TTG ACC TTC CAG-3′ (forward for mouse *Vav2*), 5′-GTG TAA TCG ATC TCC CGG GAT-3′ (reverse for mouse *Vav2*); 5′-ATG GAG ACC GG TGG AAG CAG TG-3′ (forward for mouse *Vav3*), 5′-TCC GCC TTC ATC AAG TCT TC-3′ (reverse for mouse *Vav3*); 5′-AAA GCT CAG GGC GTT CTA CC-3′ (forward for mouse *Prex1*), 5′-TAG TAA GGG GCA GGA GGC AT-3′ (reverse for mouse *Prex1*); 5′-ACA CTG GTT GCC CTG TTT GA-3′ (forward for mouse *Prex2*), 5′-CAG CGA TGC GTT TGG ATC TG-3′ (reverse for mouse *Prex2*); 5′-TGG TTA CAG GAG AGA CTT GGG-3′ (forward for mouse *Tiam1*), 5′-GTC CTC CGG GTC TTG TGT G-3′ (reverse for mouse *Tiam1*); 5′-TAT GGG ACA CAG CTG GAC AA-3′ (forward for mouse *Rac1*), 5′-ACA GTG GTG TCG CAC TTC AG-3′ (reverse for mouse *Rac1*); 5′-AGT CCC AGG TCA ACA AGC TG-3′ (forward for all *Myh* cDNAs), 5′-TCT TTG GTC ACT TTC CTG CAC T-3′ (reverse for *Myh1*), 5′-GCA TGA CCA AAG GTT TCA CA-3′ (reverse for *Myh2*), 5′-TTT CTC CTG TCA CCT CTC AAC A-3′ (reverse for *Myh4*), 5′-TTC CAC CTA AAG GGC TGT TG-3′ (reverse for *Myh7*); 5′-TCA GGT GCT TTG AGA GAT CGA C-3′ (forward for *Myod*), 5′-CGA AAG GAC AGT TGG GAA GAG T-3′ (reverse for *Myod*), 5′-CAC TGG AGT TCG TCC CAA A-3′ (forward for *Myog*) and 5′-TGT GGG CGT CTG TAG GGT C-3′ (reverse for *Myog*); 5′-CCA CAG CTG CTG CAG AAC AC-3′ (forward for *Pck1*), 5′-GAA GGG TCG CAT GGC AAA-3′ (reverse for *Pck1*); 5′-GGA GGT GGT GAT AGC CGG TAT-3′ (forward for *Fasn*), 5′-TGG GTA ATC CAT AGA GCC CAG-3′ (reverse for *Fasn*); 5′-GGA CGA GAG AAC CTT CGG GG-3′ (forward for *Acaca*), 5′-CGG ACA AGG TAA GCC CCA AT-3′ (reverse for *Acaca*); 5′-TCC TCA GTC AGC TGC CCC GT-3′ (forward for *Ppara*), 5′-TCC CGC GAG TAT GAC CCG GG-3′ (reverse for *Ppara*); 5′-AAT GGC AGT GTG CAC GTC TA-3′ (forward for *Cebpa*), 5′-CCC CAG CCG TTA GTG AAG AG-3′ (reverse for *Cebpa*); 5′-TGG ACG TTG TGT TAC TGT GG −3′ (forward for *Mttp*), 5′-TCT TAG GTG TAC TTT TGC CC-3′ (reverse for *Mttp*); 5′-TGG CCT TAC TTG GGA TTG G-3′ (forward for *Cd36*), 5′-CCA GTG TAT ATG TAG GCT CAT CCA-3′ (reverse for *Cd36*); 5′-GTC GGT CCT TCC TTG GTG TA-3′ (forward for *Ucp1*), 5′-TGC ACC ACC AAC TGC TTA GC-3′ (reverse for *Ucp1*);

5′-GTG AAT GAG ATG GCG AGG GT-3′ (forward for *Hsl*), 5′-GAG CTC CGC CTT TAA TGG GT-3′ (reverse for *Hsl*); 5′-CGG TGT TGT GCG GTG TCT GTA GT-3′ (forward for *Ppargc1a*), 5′-CGA TCA CCA TAT TCC AGG TCA AG-3′ (reverse for *Ppargc1a*); 5′-TGC TCT TCT GTA TCG CCC AGT-3′ (forward for *Cidea*), 5′-GCC GTG TTA AGG AAT CTG CTG-3′ (reverse for *Cidea*); 5′-CGG TTT CAG AAG TGC CTT GCT-3′ (forward for *Pparg*), 5′-CAC CAA CTT CTC CTT CTC-3′ (reverse for *Pparg*); 5′-CCA CTA GCA GCG AGG ACT TCA C-3′ (forward for *Prdm16*), 5′-GGG GAC TCT CGT AGC TCG AA-3′ (reverse for *Prdm16*); 5′-GAG GGG CTA CCT TCC TCT CA-3′ (forward for *Trim63*), 5′-TTT ACC CTC TGT GGT CAC GC-3′ (reverse for *Trim63*); 5′-TCT GGG ACC TTA GGA GAG CC-3′ (forward for *Fbxo32*), 5′-CCC CCA CCC CAG GAA TTA AC-3′ (reverse for *Fbxo32*); 5′-TGC ACC ACC AAC TGC TTA GC-3′ (forward for *Gapdh*), 5′-TCT TCT GGG TGG CAG TGA TG-3′ (reverse for *Gapdh*).

**Nuclear magnetic resonance.** Body composition was measured by nuclear magnetic resonance (Whole Body Composition Analyzer; EchoMRI, Houston, TX) as indicated[81].

**Histology.** Tissues were extracted, fixed in 4% paraformaldehyde (Cat. #252931, PanReac), paraffin-embedded, cut in 2–3-μm- (in the case of muscle, liver, and BAT) or 5-μm- (in the case of the WAT) thick sections, and stained with hematoxylin–eosin. Images were captured using an Olympus BX51 microscope coupled to an Olympus DP70 digital camera.

**Histological quantifications.** The cross-sectional area of the skeletal fibers was measured with the ImageJ software (National Institutes of Health, Bethesda, MD) using the Region of Interest Manager tool and manually selecting the limits of at least 50 fibers per animal. Two independent regions of hematoxylin–eosin-stained gastrocnemius sections (×400 amplified pictures) were used for each quantification. Values obtained were transformed according to the scale used. The diameter of C2C12-derived myotubes was calculated using a similar approach.

The diameter of the adipocytes was measured using ×100 amplified images of the stained histology samples from either two or three representative regions per animal utilizing the macro "Adipocyte Tool" of the ImageJ software. To this end, the options in the "p" and "s" buttons of macro were configured for values between 80 and 20,000 using the Huang thresholding method. Errors in the automatic detection of the adipocytes were manually corrected. Values obtained were transformed according to the scale used.

The number of nuclei per field in BAT was quantified in hematoxylin–eosin-stained sections with ImageJ using at least two different ×200 amplified pictures per animal. After splitting the color channels, the threshold was adjusted in the red channel to select the cell nuclei. The number of nuclei was obtained using the tool "Analyze Particles" after the selection of the adequate size range. For the analysis of the BAT lipid vacuoles, the green channel was selected after the color separation and the colors inverted to visualize the vacuoles in black. The threshold was adjusted to select the lipid droplets. The size of the vacuoles was quantified using also the ImageJ "Analyze Particles" tool.

**Treadmill assays.** Three-month-old mice were trained for 2 days before conducting the experiment with an uphill inclination of 10° on a custom four-lane treadmill (TSE systems). On the first training day, mice ran for 5 min at 8 m min$^{-1}$. On the second training day, mice ran for 5 min at 8 m min$^{-1}$, followed by an extra 5 min run at 10 m min$^{-1}$. For the experimental run, mice ran for 40 min at 10 m min$^{-1}$ and then the speed was increased at a rate of 1 m min$^{-1}$ every 10 min for 30 min. Finally, mice were forced to run further by increasing the speed at a rate of 1 m min$^{-1}$ every 5 min until exhausted. Total running time and distance were calculated for each animal.

**Isolation of muscle satellite cells.** Muscles were minced with razors and digested twice in DMEM media containing collagenase D (0.08%, Cat. #11088866001; Roche) and trypsin (0.125%, Cat. #1590-046; Invitrogen) at 37 °C for 25 min in a shaking water bath. Samples were then centrifuged at low speed and the digestion repeated with the pelleted fragments. Cells were then filtered, centrifuged, and incubated in Lysis buffer 1× (Cat. #555899; BD) for 10 min on ice. After centrifugation, cells were resuspended in phosphate-buffered saline solution supplemented with 2.5% fetal bovine serum (Cat. #10270, Gibco) and stained for 20 min with allophycocyanin-conjugated anti-CD45 (Cat. #103112, BioLegend), Pacific blue-conjugated Sca1 (Cat. #108119, BioLegend), phycoerythrin-conjugated anti-α7-integrin (Cat. #53-0010-05, ABlab), and fluorescein isothiocyanate-conjugated anti-CD34 (Cat. #11-0341-82, eBioscience). Sca1$^-$;CD45$^-$;integrin α7$^+$;CD34$^+$ cells were sorted using a FACS Aria II (BD). The purified satellite cells were cultured in Ham's F10 medium (Cat. #11550-043, Gibco) supplemented with 30% fetal bovine serum (Cat. #10270, Gibco) and 0.025 μg mL$^{-1}$ human basic fibroblast growth factor (Cat. #hFGDBCF, Immunostep) on collagen type I-coated wells (Cat. #3447-020-01, Cultrex, R&D Systems).

**Activation and proliferation of satellite cells.** One thousand cells were plated on collagen I-treated 10-mm-diameter coverslips in 12-well plates with growth

medium. After 24 h (before the first division to assess activation) and 72 h (to assess proliferation) in culture, cells were labeled with EdU (Click-iT EdU Alexa Fluor 647 Imaging kit, Cat. #C10340; Invitrogen) according to the manufacturer's instructions and stained with 4′,6-diamidino-2-phenylindole dihydrochloride (1 μg mL⁻¹, Cat. #D1306; Invitrogen). The percentage of EdU-positive cells was quantified under a fluorescence microscope.

**Myogenic differentiation of satellite and C2C12 cells**. Ten thousand satellite cells were plated on collagen I-treated 12-well plates and maintained in Ham's F10 medium supplemented with 30% fetal bovine serum and 0.025 μg mL⁻¹ human basic fibroblast growth factor until a 70% confluence was reached. The fetal bovine serum in the medium was then decreased to 2% and replaced every other day until the end of the experiment. C2C12 cells were shifted into a differentiation medium (DMEM supplemented with 2% horse serum; Cat. #16050122, Gibco), 1% L-glutamine, penicillin (10 μg mL⁻¹), and streptomycin (100 μg mL⁻¹) when they reached 80% confluency. The medium was replaced every other day until the end of the experiment.

**Immunofluorescence**. Cells were fixed in 4% formaldehyde (Cat. #F8775, Sigma-Aldrich) in phosphate-buffered saline solution for 30 min, washed twice with phosphate-buffered saline solution, permeabilized with 0.5% Triton X-100 (Cat. #X100, Sigma-Aldrich) in TBS-T [25 mM Tris-HCl (pH 8.0), 150 mM NaCl, 0.1% Tween-20 (Cat. #P7949, Sigma-Aldrich)] for 15 min, washed three times with TBS-T, blocked in TBS-T supplemented with 2% bovine serum albumin (Cat. #A4503, Sigma-Aldrich) for 2 h, and then incubated with antibodies to MHCII (1:700, Cat. #M4276; Sigma-Aldrich) in blocking buffer at 4 °C overnight. Alexa Fluor 647-labeled goat anti-mouse IgG (1:500 in blocking buffer, Cat. #A28181; Thermo Fisher Scientific) was used as the secondary antibody. After 30 min, the cells were washed three times in TBS-T and finally stained with 1 μg mL⁻¹ of 4′,6-diamidino-2-phenylindole (Cat. #D1306, Invitrogen) for 5 min to visualize the nuclei. Images were captured in an EVOS FL Cell Imaging System microscope (Cat. #AMF4300, Thermo Fisher Scientific).

**Infusion of mice with insulin and IGF1**. Mice were fasted for 7 h, deeply anesthetized with a combination of ketamine (CN #571267.3; Merial) plus xylazine (Cat. #572126.2; Bayer), and injected via the inferior vena cava with either insulin (0.5 U kg⁻¹ for normal dosage experiments and 0.1 U kg⁻¹ in the case of suboptimal stimulation conditions when indicated; Cat. #775502; Actrapid NovoNordisk) or IGF1 (0.2 and 0.5 mg/kg in the case of $Vav2^{Onc/Onc}$ and $Vav2^{L332A/L332A}$ mice, respectively; Cat. #100-11, PeproTech). After 2 (liver), 5 (skeletal muscle), and 7 min (perigonadal WAT), tissues were collected and snap frozen.

**Western blotting**. In the case of mouse tissue extracts, the frozen samples were disrupted with the help of a Dispomix Drive homogenizer (Cat. #900020.00, Medic Tools AG) in gentleMACS M tubes (Cat. #130-096-335, Miltenyi Biotec) in ice-cold lysis buffer [50 mM Tris-HCl (pH 7.5), 1 mM ethylene glycol tetraacetic acid (Cat. #E3889, Sigma-Aldrich), 1 mM ethylenediaminetetraacetic acid (Cat. #ED-2SS, Sigma-Aldrich), 1% Triton X-100 (Cat. #X100, Sigma), 1 mM sodium orthovanadate (Cat. #S6508, Sigma-Aldrich), 50 mM sodium fluoride (Cat. #S7920, Sigma-Aldrich), 5 mM sodium pyrophosphate (Cat. #221368; Sigma-Aldrich), 0.27 M sucrose (Cat. #S9378, Sigma-Aldrich), 1 mM phenylmethanesulfonyl fluoride (Cat. #P7626, Sigma-Aldrich), and CØmplete Protease Inhibitor Cocktail (Cat. #11836145001, Roche)]. After short centrifugation to eliminate cellular debris, the samples were centrifuged once (in the case of muscle), twice (in the case of the liver), or thrice (in the case of WAT) at 4 °C and 8500 × g for 30 min to clarify the sample. In the case of extracts from cultured cells, samples were washed with phosphate-buffered saline solution and lysed in RIPA buffer [10 mM Tris-HCl (pH 7.5), 150 mM sodium chloride, 1% Triton X-100 (Cat. #X100, Sigma-Aldrich), 1 mM sodium orthovanadate (Cat. #S6508, Sigma-Aldrich), 1 mM sodium fluoride (Cat. #S7920; Sigma-Aldrich), and CØmplete]. Cellular extracts were kept for 5 min on ice and centrifuged at 16,500 × g for 10 min at 4 °C. Regardless of the source, protein extracts were separated electrophoretically in sodium dodecyl sulfate-polyacrylamide gel electrophoresis (SDS-PAGE) gels, transferred onto nitrocellulose membranes, and subjected to immunoblot analyses[82]. Antibodies used include: phospho-Akt (Thr³⁰⁸) (dilution 1:1000; Cat. #4056, Cell Signaling Technologies), phospho-Akt (Ser⁴⁷³) (dilution1:1000; Cat. #4051, Cell Signaling Technologies), Akt (dilution 1:1000; Cat. #2920, Cell Signaling Technologies), phospho-GSK3α/β (dilution 1:1000; Cat. #9327, Cell Signaling Technologies), GSK3α/β (dilution 1:1000; Cat. #5676, Cell Signaling Technologies), phospho-S6K (Thr³⁸⁹) (dilution 1:1000; Cat. #9215, Cell Signaling Technologies), phospho-ERK1/2 (Thr202/Tyr204) (dilution 1:1000; Cat. #4370, Cell Signaling Technologies), phospho-Pak (dilution 1:1000; Cat. # ab40795, Abcam), α-tubulin (dilution 1:2000; Cat. #CP06, Calbiochem), MHCII (dilution 1:2000; Cat. #M4276; Sigma), S6K (dilution 1:1000; Cat. #sc-230, Santa Cruz Biotechnologies), HA (dilution 1:1000; Cat. #3724, Cell Signaling Technologies), phospho-Tyr (for detecting phospho-IRS1; 1:1000 dilution, Cat. #sc-7020, Santa Cruz Biotechnologies), IRS1 (dilution 1:1000; Cat. #05-1085, Millipore), Rac1 (dilution 1:1000; Cat. #ARC03, Cytoskeleton), GFP (dilution 1:2000; Cat. #902601, BioLegend), and Ucp1 (dilution 1:500; Cat. #sc-293418, Santa Cruz Biotechnologies). The polyclonal rabbit antibody to Vav2 (dilution 1:1000) was homemade using as epitope the acidic region of this protein[83]. For quantification of the relative protein phosphorylation levels, the intensity of the bands was measured with the ImageJ software. Values were normalized taking into consideration the amount of total protein present in each sample.

**Generation of stable cell lines**. We used the CRISPR/Cas9 gene-editing strategy to knockout the *Vav2* locus in C2C12 cells. To this end, two different sgRNAs targeting the exon 1 of *Vav2* at the sequences 5′-CGC CCA ACC ACC GCG TCG TG-3′ (guide 1) and 5′-GCA GTG CCG GGA CCC TCA CC-3′ (guide 2) were annealed and ligated into the plasmid pSpCas9(BB)-2A-GFP (Cat. #48138, Addgene). The plasmids were introduced into C2C12 cells using Lipofectamine 2000 (Cat. #11668019; Invitrogen), and 36 h later, the GFP-positive cells were single-cell sorted using a FACS Aria II (BD) to obtain individual clones. Cell clones were grown in 96-well plates, screened by PCR, and further verified by both DNA-sequencing and immunoprecipitation. To this end, 2 mg of protein extracts obtained from C2C12 cells were incubated in 1 mL of RIPA buffer with 1.5 μL of antibody to Vav2 (see above) for 2 h at 4 °C. Immunocomplexes were collected with Gammabind G-Sepharose beads (Cat. #GE17-0885-01; GE Healthcare), washed three times in RIPA buffer, resuspended in SDS-PAGE buffer, boiled for 5 min, and subjected to immunoblot analysis using antibodies to Vav2 as indicated above. Independent clones (WT and *Vav2* null) were then used in the experiments.

Cell pools ectopically expressing the indicated proteins were generated using a lentiviral-delivery method. To generate the viral particles, 2.5 μg of pMD-G, 7.5 μg of pNGVL-MLV-gag-pol, and 10 μg of the appropriate lentiviral plasmid were transfected into HEK293T cells using Lipofectamine 2000 according to the manufacturer's instructions. Cell culture supernatants were collected 24, 48, and 72 h later, passed through 0.45-μm filters (Cat. #10462100, GE Healthcare), supplemented with Polybrene (8 μg mL⁻¹; Cat. #H9268, Sigma-Aldrich), and poured onto exponentially growing C2C12 cells. Twenty-four hours after the infection, the cells were washed and selected with hygromycin B (Cat. #21414000, Roche). Expression of proteins was verified using immunoblots with antibodies to the HA epitope.

*Vav2* and *Rac1* knockdown cells were generated by infection of C2C12 cells with lentiviral particles encoding different shRNAs (TRCN0000097094 for *Vav2* sh₁, TRCN0000097098 for *Vav2* sh₂, and TRCN0000310888 for *Rac1* sh₁ plus TRCN0000310901 for *Rac1* sh₂; Sigma-Aldrich) as above. Positive cells were selected with puromycin (Cat. #P8833; Sigma-Aldrich). Protein depletion in *Vav2* knockdown cells was checked by immunoprecipitation analyses as above. In the case of Rac1, cellular extracts were interrogated using immunoblot analyses with antibodies to Rac1. From these analyses, we selected a *Rac1* knockdown cells with either partial (*Rac1* sh₁) or total depletion of the endogenous protein (*Rac1* sh₁,₂).

To generate the Glut4 reporter cell line, C2C12 cells were infected with lentivirus produced in HEK293T cells that were transfected with the plasmid pLenti-myc-GLUT4-mCherry (Cat. #64049; Addgene). Upon transduction, mCherry-positive cells were isolated by flow cytometry.

**IRS1 phosphorylation levels**. C2C12 cells cultured in 10-cm plates were either nontreated or treated with insulin for 10 min, washed in phosphate-buffered saline solution, and lysed in RIPA buffer as above. Equal amounts of protein were incubated in 1 mL of RIPA buffer with 2 μL of antibody to IRS1 for 2 h at 4 °C. Immunocomplexes were collected with Gammabind G-Sepharose beads, washed three times in RIPA buffer, resuspended in SDS-PAGE buffer, boiled for 5 min, and subjected to sequential immunoblot analysis with antibodies to phospho-Tyr and IRS1.

**PIP₃ determination in cells**. Cells at 80% confluence cultured in 10-cm plates were either nontreated or treated with insulin for 5 min. After the treatment, plates were immediately placed on ice and the extraction and quantitation of phospholipids and PIP₃ was done using an ELISA kit as indicated by the manufacturer (K-2500s, Echelon Biosciences).

**Detection of PI3K activation using bioreporters**. A total of $1 \times 10^6$ exponentially growing C2C12 cells were electroporated using the Neon Transfection System (1650 V, 15 ms, 2 pulses; Cat. #MPK5000, Invitrogen) and 20 μg of the appropriate plasmids following the manufacturer's instructions. In all cases, cells were transfected with the bioreporter-encoding plasmid (pEGFP-C1-PH-Akt or pmCherry-C1-PH-Akt) and, whenever required, co-transfected with expression vectors expressing the indicated EGFPs. After 36 h, cells were trypsinized, resuspended in serum-free medium, and replated onto coverslips. After 3 h of starvation plus the indicated treatments, cells were either left unstimulated or were stimulated with 75 nM insulin for 10 min. Cells were then washed with phosphate-buffered saline solution, fixed in 4% formaldehyde (Cat. #F8775, Sigma-Aldrich) in phosphate-buffered saline solution for 15 min, permeabilized in 0.5% Triton X-100 in TBS-T for 10 min, washed in phosphate-buffered saline solution, and mounted with Mowiol (Cat. #475904, Calbiochem). Images were captured with LAS AF software (version 2.6.0.72266, Leica) in a Leica TCS SP5 confocal microscope. When appropriate, cells were treated with wortmannin (100 nM), PIk-75 (200 nM), TGX-221 (500 nM), latrunculin A (200 nM), and cytochalasin D (2 μM). When

indicated, samples were stained for 20 min with Alexa Fluor 647-labeled phalloidin (1:200; Cat. #A22287, Thermo Fisher Scientific) and washed before mounting.

**Determination of GTP-Rac1 levels**. Frozen tissue samples were homogenized using a mortar and a pestle and lysed in 1 mL of pulldown buffer (20 mM Tris-HCl [pH 7.5], 150 mM NaCl, 5 mM $MgCl_2$, 0.5% Triton X-100, 10 mM β-glycer-ophosphate, 1 mM dithiothreitol (DTT), and CØmplete). After an incubation of 10 min on ice and short centrifugation at 4 °C to eliminate cell debris, the supernatants were collected and protein concentrations quantitated (Bradford; Cat. #5000006, Bio-Rad). The lysates with an equal amount of total protein were incubated with 10 µg of a GST protein fused to the Rac1 binding domain of Pak1 (GST-Pak1 RBD) that was bound to glutathione-Sepharose beads (Cat. #GE17-0756-01; GE Healthcare) for 2 h at 4 °C in a rotating wheel. The purification of the GST-Pak1 RBD and GST control proteins from *Escherichia coli* was done using glutathione-coated beads as above[35]. After the incubation, the beads were washed three times in pulldown buffer and boiled in SDS-PAGE loading buffer. Released proteins were separated electrophoretically, transferred to nitrocellulose filters, and subjected to immunoblot analyses. Aliquots of the total cellular lysates used in the above pulldowns were processed in parallel to determine the amount of Rac1 present in each experimental sample.

**Glut4 translocation assays**. C2C12 cells were transduced with lentivirus produced in 293T cells using the pLenti-myc-Glut4-mCherry plasmid (Cat. #64049; Addgene) as indicated above. mCherry-positive cells were seeded onto coverslips, starved, and stimulated with insulin as indicated for the mCherry-Akt PH translocation experiments. After fixation in nonpermeabilizing conditions with 4% methanol-free formaldehyde (Cat. #28908; Thermo Fisher Scientific) for 15 min, cells were washed thrice with phosphate-buffered solution, blocked with 2% bovine serum albumin in phosphate-buffered saline solution for 1 h, and incubated overnight with an antibody to the Myc epitope (Cat. #M5546, Sigma-Aldrich; 1:1000 dilution). After three washes in phosphate-buffered saline solution, cells were incubated for 30 min with an Alexa Fluor 647-labeled antibody to mouse immunoglobulins (Cat. #A28181, Invitrogen; dilution 1:500), washed thrice, and mounted using Mowiol. Glut4 translocation was measured as the ratio between membrane-exposed (Myc-positive signal; far red) and total (mCherry-positive signal) Glut4 present in cells. To avoid spurious differences due to uneven staining of the samples, control and transfected cells were compared within the same field and values normalized to controls. In the case of knockout clones, the cells were stained for 10 min with the CellTracker Blue CMAC (7-amino-4-chlor-omethylcoumarin) dye (Cat. #C2110, Thermo Fisher Scientific; 1:1000 dilution), washed, and mixed in equal proportions with WT cells (not stained with CMAC) before seeding onto the coverslips.

**Glucose and insulin tolerance tests**. The animals were fasted overnight for 14 h and basal blood glucose levels were measured with an Accu-Chek glucometer using blood samples collected from tail bleeds (Cat. #04680430003, Roche). In the case of glucose and insulin tolerance test, mice fasted as above were injected intraperitoneally with either D-glucose (2 g kg$^{-1}$; Cat. #G8270, Sigma) or insulin (0.5 U kg$^{-1}$; CN #775502, Actrapid NovoNordisk). The levels of glucose in the plasma were determined from tail bleeds taken at the indicated time-points.

**Determination of plasma concentrations of insulin and C-peptide**. Levels of insulin and C-peptide were determined in blood samples using the Rat/Mouse Insulin (Cat. #EZRMI-13K, Merck Millipore) and the mouse C-peptide (Cat. #80-CPTMS-E01, Alpco) ELISA kits, respectively.

**Assessment of glucose homeostasis**. Hyperinsulinemic–euglyclemic clamps were carried in age-matched conscious unrestrained catheterized mice as described[84,85]. To this end, catheters were surgically implanted 7 days prior to the experiment in the right jugular vein and exteriorized above the neck using a vascular access button (Instech Laboratories Inc., Plymouth Meeting, PA). Mice were fasted for 3 h, followed by an infusion for 2 h of [3-$^3$H] glucose (0.05 µCi min$^{-1}$; Cat. #NET331, PerkinElmer). Continuous insulin infusion (1.5 mIU kg$^{-1}$ body weight min$^{-1}$, Umuline, Lilly France) was used for the induction of hyperinsulinemia. Upon reaching steady state, the insulin-stimulated glucose uptake in tissues was determined in vivo using a 10 µCi bolus injection of 2-[$^{14}$C] deoxyglucose (Cat. #NEC720A250UC, PerkinElmer). After 30 min, mice were rapidly killed by cervical dislocation and tissues were removed and stored at −80 °C until use.

Glucose concentration was measured using the glucose oxidase method (GLU, Roche Diagnostics) and insulin using an ELISA commercial kit (Cat. #90080, CrystalChem Inc.). Measurements of 2-[$^{14}$C] deoxyglucose-6-phosphate concentration in individual tissues allowed calculation of the glucose utilization index in tissues. After 30 min, tissues were removed and stored at −80 °C as above.

**Determination of triglyceride concentration in liver and muscle**. Tissue fragments (50 mg) were manually homogenized by mincing with a scalpel blade and triglycerides were extracted in 500 µL of an ice-cold chloroform–methanol solution

(2:1, vol/vol) by shaking in a thermoblock at room temperature for 2 h. For phase separation, Milli-Q water (Millipore) was added and samples were centrifuged at room temperature for 30 min at 8500 × *g*. The organic bottom layers resulting from that centrifugation were collected in a new tube, dried up using a vacuum concentrator centrifuge (UNI VAPO 100 ECH; UniEquip), and resuspended in chloroform. After evaporation of the organic solvent in a thermoblock, the solid residue was resuspended in the Triglycerides-LQ reagent (Cat. #1001314, Spinreact) and triglyceride content was determined using a spectrophotometer (Cat. #170-2501, Bio-Rad).

**Determination of serum concentrations of triglycerides and cholesterol**. Blood triglycerides and cholesterol content was quantified by colorimetric methods. To this end, 10 µL of serum per mice were analyzed using the Triglycerides-LQ (Cat. #1001314, Spinreact) and Cholesterol CHOP-POD (Cat. #41020, Spinreact) reagents in a spectrophotometer (Cat. #170-2501, Bio-Rad) following the manufacturer's instructions.

**Metabolic determinations**. Food intake of individually caged mice was measured every other day. Energy expenditure, respiratory quotients, and locomotor activity of animals were analyzed using a calorimetric system (LabMaster, TSE Systems, Bad Homburg, Germany). Rectal temperature was measured using a digital thermometer (VedoBEEP, Artsana, Grandate, Italy). Skin temperature surrounding BAT was recorded with an infrared camera (B335:Compact-Infrared-Thermal-Imaging-Camera, FLIR, West Malling, Kent, UK).

**BAT oxygen consumption rate**. Interscapular BAT was excised and washed in DMEM supplemented with 25 mM glucose (Cat. #103577-100, Agilent) and 25 mM HEPES (Cat. #15630080, Gibco). After a second wash, fragments (9 mg each) from the central region of the brown fat pad were collected with the help of a punch and placed onto Seahorse XF24 Islet Capture Microplates (Cat. #101122-100, Agilent). Prewet capture screens were placed over each well and the explants were washed again with DMEM wash buffer. After a final wash in assay media (25 mM glucose Seahorse XF DMEM Medium; Cat. #103575-100 and #103577-100, Agilent), 450 µL of assay media were added to each well and the oxygen consumption rate measured in the Seahorse XFe24 Analyzer (Agilent).

**Quantification of mitochondrial DNA content**. Whole genomic and mitochondrial DNA was extracted from frozen tissue samples using the PureLink™ Genomic DNA Mini kit (Cat. #K182002; Invitrogen) according to the manufacturer's indications. The mitochondrial cytochrome *C* oxidase-1 (*Co1*) gene and the single-copy nuclear NADH dehydrogenase ubiquinone flavoprotein 1 (*Ndufv1*) genes were amplified by quantitative PCR using the primers 5′-TGC TAG CCG CAG GCA TTA C-3′ (*mtCo1*, forward) and 5′-GGG TGC CCA AAG AAT CAG AAC-3′ (*mtCo1*, reverse) and 5′-CTT CCC CAC TGG CCT CAA CA G-3′ (*Ndufv1*, forward) and 5′-CCA AAA CCC AGT GAT CCA GC-3′ (*Ndufv1*, reverse) and the iQ SYBR Green Supermix (Cat. #1708880; Bio-Rad) reactive in a One Step Plus device. Relative mitochondrial DNA content was calculated normalizing the *mtCo1* vs. the *Ndufv1* expression values obtained.

**Statistical analyses**. The number of replicates (*n*), the type of statistical test performed, and the statistical significance for each experiment is indicated in the appropriate figure legend. Whenever possible, data normality was analyzed using Shapiro–Wilk tests. Nonparametric tests were applied to non-normally distributed data. In the case of Student's *t* tests, homogeneity of variances was verified by Snedecor *F* test. In the cases in which variance equality was rejected, the Welch's correction was used. Statistical analyses were performed using the GraphPad Prism software (versions 6.0 and 8.3.1).

**Reporting summary**. Further information on research design is available in the Nature Research Reporting Summary linked to this article.

## Data availability

All relevant data are available from the corresponding author upon reasonable request. Source data are provided with this paper.

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

## Acknowledgements

We thank C. García-Macías, the personnel of the CIC Microscopy Unit and M. Blázquez for expert histological analyses, microscopy work and lab assistance, respectively. We also thank the input of S. Virtue on our metabolic data. X.R.B. is supported by grants from the Castilla-León Government (CSI252P18, CLC-2017-01), the Spanish Ministry of Science and Innovation (MSI) (RTI2018-096481-B-100), and the Spanish Association against Cancer (GC16173472GARC). X.R.B.'s institution is supported by the Programa de Apoyo a Planes Estratégicos de Investigación de Estructuras de Investigación de Excelencia of the Castilla-León autonomous government (CLC-2017-01). S.R.-F. and L.F.L.-M. contracts have been mostly supported by funding from the MSI (BES-2013-063573) and the Spanish Ministry of Education, Culture and Sports (L.F.L.-M., FPU13/02923), respectively. Subsequently, they both were supported by the CLC-2017-01 grant. Both Spanish and Castilla-León government-associated funding is partially supported by the European Regional Development Fund.

## Author contributions

S.R.-F. participated in all experimental work, analyzed data, and contributed to both artwork design and manuscript writing. L.F.L.-M., B.P., D.B., O.A.-M., A.A., C.V.-D., C.D., R.C., and R.B. helped in metabolic and energetic analyses of knock-in mice. I.F.-P. and L.F.L.-M. carried out signaling and qRT-PCR experiments in tissues and cells. X.R.B. conceived the work, analyzed data, wrote the manuscript, and performed the final editing of figures.

## Competing interests

The authors declare no competing interests.
