## [Peer Review File · Nature Communications]

Reviewers' Comments:

Reviewer #1:

Remarks to the Author:

In this manuscript, Rodriguez-Fdez and colleagues investigated the contribution of the GEF Vav2, a Rho/Rac/Cdc42 activator, to skeletal muscles growth and metabolism. Central to this study, they exploit mice with hypo-active Vav2 (L332A) or hyper-active Vav2 (onco-Vav2). Mice with low Vav2 activity presented with smaller muscles. They also failed to respond like wild-type animals to insulin. In contrast, mice with hyper-active Vav2 showed bigger muscles and were more responsive to insulin. The authors generated Vav2-null C2C12 cells and show that they fail to respond correctly to insulin stimulation (pAKT, pGSK3 and pS6K). Hence, the authors propose that Vav2 controls the signaling output of the insulin pathway in muscle.

This study is composed of a solid body of work. The mice demonstrate interesting metabolic phenotypes that the authors describe in great details. Where the story falls short is to provide mechanistic insights on the contribution of Vav2 to insulin signaling. In particular, what is, or are, the small GTPase target(s) of Vav2?

Major Comments:

1. In figure 3, the authors studied CRISPR-derived Vav2 KO myoblasts (and rescued versions) to study a key question: the role(s) of Vav2 in insulin signaling. Unfortunately, these cells are poorly characterized. First, do they differentiate normally? This is an important question if the authors would want to generate myotubes. A panel of differentiation markers should be analyzed at different time points of differentiation (Myog, MHC, Myomaker etc.). The authors should also directly establish whether to formation of myotubes is unaffected by Vav2 KO or in the rescue conditions.

2. It is unclear why the signaling was studied in myoblasts rather than in myotubes, which would be the correct model to mimic the in vivo data (muscle fibers). Also, why did the authors rescue with onco-Vav2 and not wild-type Vav2? If Vav2 is contributing to insulin signaling, the wt protein should be sufficient to rescue the defects. One worry is that Onco-Vav2 may have dominant effects and not behave like the normal protein. As such, a bona fide contribution of Vav2 to insulin signaling is not clearly demonstrated in this study. Implicating Vav2 in insulin signaling goes beyond showing functions for Onco-Vav2.

3. The author make a gross assumption that Vav2 is mainly a Rac GEF in vivo in muscle tissue (on p4, this statement is supported by a reference to a review paper – the primary data should have been cited here?). There is absolutely no evidence in this study to support that Vav2 regulates preferentially Rac1 over Cdc42 and RhoA. The GTP-loading status of RhoA, Rac1 and Cdc42 should be determined at a minimum in the Vav2 KO myoblasts. Additional experiments should also be performed to directly determine the contribution of Vav2 GEF activity to insulin signaling. Can the authors test, as an example, if the rescue of insulin signaling with Onco-Vav2 in the KO C2C12 cells can be blunted by depleting Rac1 (and what is the impact of depleting Cdc42 or RhoA?). Finally, how does activation of the small GTPase(s) contribute to insulin signaling?

Additional Comments:

1. The introduction is incomplete. The Rho GTPases (in particular Rac1) and Rho GEFs have been described to control various aspects of myogenesis. This should be mentioned. Rac1 is important for early myogenesis (myoblast fusion step). Could depleting Vav2 affect this process in mice?

2. The authors state that there is hypertrophy in the Onco-Vav2 muscle. This has not been formally demonstrated. An alternative explanation, based on the observation that Onco-Vav2 myoblasts are more proliferative and differentiate better, that more cells are added to fibers. They

authors should determine whether they have increased proliferation/fusion *in vivo* or if they observe hypertrophy.

Reviewer #2:

Remarks to the Author:

This study investigated the metabolic phenotypes of catalytically inactive (Vav2L232) and gain-of-function strain (Vav2onc) of Vav2 knockin mice. Vav2L232 mice exhibit reduced muscle mass, decreased insulin responses, and are much prone to high fat diet (HFD)-induced obesity, whereas Vav2onc mice display these phenotypic changes opposite to Vav2L232 mice. The authors conclude that Vav2 regulate insulin/IGFI signaling in skeletal muscle, thereby explaining the metabolic phenotypes of these two strains of mice. The identification of Vav2 as a potential regulator of insulin signaling and muscle growth is potentially important. However, as neither Vav2L232 mice nor Vav2onc mice are muscle-specific, the data presented is not sufficient to support the claim that the altered insulin responsiveness, adiposity and other metabolic parameters in the mice are due to the specific actions of Vav2 in muscle. Furthermore, the authors failed to explain how altered insulin signaling in skeletal muscle lead to changes in adiposity and adaptive thermogenesis of brown adipose tissues.

Major points:

1. To firmly draw the conclusion that altered insulin responsiveness of the two strains of the mice are attributed to altered insulin sensitivity in muscle, but not in other metabolic organs (adipose tissues, liver), glucose clamp studies should be performed to determine insulin sensitivity in each tissue in the mice.
2. Lipid profiles in the circulation and ectopic lipid accumulation in skeletal muscle should also be measured in both the chow-diet and HFD mice.
3. In Fig. 4 legend, the authors claimed that Reduction of Vav2 catalytic activity impairs systemic response to insulin. However, except ITT data in panel C (which is not convincing) and panel D, there is insufficient evidence to support this conclusion.
4. In fig. 5, how to explain why Vav2L232 mice are more prone to develop HFD-induced obesity? Although insulin signaling in Vav2L232 adipocytes is not affected, it does not necessarily mean that Vav2 has no direct effect on adipocytes.
5. In fig. 6 legend, it states that "Increase of Vav2 catalytic activity improves short-term systemic response". However, none of the data presented in this figure is related to short-term systemic responses. Again, how to explain the lean phenotype of the mice.
6. In Fig. 7, the brown adipocyte function is poorly characterized. The changes in cell morphology and mild increase in mRNA expression of UCP1 do not mean increased brown adipocyte activity. UCP1 protein expression, mitochondrial density and complex activity, *ex vivo* oxygen consumption of brown adipose tissues, expression of a panel of genes involved in adaptive thermogenesis (PGC1alpha, CIEDA, DIO2...) should be analyzed. Furthermore, the mice should be challenged with cold environment to better assess the thermogenic functions of brown adipose tissues.
7. If Vav2 catalytic activity indeed modulates brown adipocyte activity, it may explain the changes in adiposity in the two strains of mice. Then, the authors need to perform additional studies to investigate how Vav2 modulate brown adipocyte activity or formation.

Reviewer #3:

Remarks to the Author:

Vav2, a GEF of Rac1, has been known to regulate insulin-elicited GLUT4 translocation by organizing cortical actin in the skeletal muscle and glucose-activated insulin secretion in the pancreas. Thus, Rac1 disruption in the skeletal muscle prevents insulin-dependent glucose uptake. This manuscript shows that Vav2 is involved in metabolic syndrome using Vav2L332A/L332A and Vav2Onc/Onc knock-in mice. Authors found the reduction of skeletal muscle mass and the increase of adipose mass in Vav2L332A/L332A mice, and vice versa in Vav2Onc/Onc mice. Because insulin-

elicited Akt activation was affected specifically in the skeletal muscle of both knock-in mice, they concluded that Vav2-Rac1-PI3K-Akt axis is necessary for insulin signaling in the skeletal muscle. Because of previous finding of Vav2-Rac1-PI3K-Akt axis (Campa et al., *Small GTPase*, 6, 71-80, 2015), this manuscript lacks novelty. The followings are critical comments for this manuscript.

1. According to Human Protein Atlas, the expression level of Vav2 is high in the liver and adipose tissue, pancreas but low in the skeletal muscle. Add the data for mRNA and protein levels of Vav2 and Rac1 in the various organs including the liver, adipose tissue, skeletal muscle and pancreas, using RT-PCR, immunoblotting and immunofluorescence in WT, Vav2L332A/L332A and Vav2Onc/Onc mice.

2. For analysis of insulin and IGF signaling in the liver, adipose tissue and skeletal muscle from WT and Vav2L332A/L332A and Vav2Onc/Onc mice, add the phosphorylation data of IR/IGFR, IRS-1, and Erk1/2. PIP2 and IP3 levels should be measured in the liver, adipose tissue and skeletal muscle from WT, Vav2L332A/L332A and Vav2Onc/Onc mice. In addition, after insulin and IGF treatment, the phosphorylation of IR/IGFR, IRS-1 and Erk1/2 and the level of PIP2 and IP3 should be measured from C2C12 myoblasts and satellite cells obtained from WT, Vav2L332A/L332A and Vav2Onc/Onc mice

3. No alteration of insulin and IGF signaling in the liver and adipose tissue in Vav2L332A/L332A and Vav2Onc/Onc mice suggests that there might be no change of Rac1 activity. Thus, Rac1 activity should be measured from the liver, adipose and skeletal muscle of WT, Vav2L332A/L332A and Vav2Onc/Onc mice after insulin and IGF administration. According to Human Protein Atlas, Rac1 protein level is very low in the liver and adipose tissue! Thus, authors could explain why insulin and IGF signaling is not altered in the liver and adipose obtained from WT, Vav2L332A/L332A and Vav2Onc/Onc mice.

4. According to previous paper (Chiu et al., *Cell Signal.*, 23, 1546, 2011), there is an Akt-Vav2-Rac1-actin-GLUT4 signaling pathway in the skeletal muscle, Thus, the skeletal muscle of WT, Vav2L332A/L332A and Vav2Onc/Onc mice might have different GLUT4 translocation and glucose uptake. Observe insulin-elicited GLUT4 translocation to the plasma membrane and measure glucose uptake from the gastrocnemius and soleus of WT, Vav2L332A/L332A and Vav2Onc/Onc mice. In addition, adipose tissue might be a good negative control for the insulin-elicited GLUT4 translocation and glucose uptake in WT, Vav2L332A/L332A and Vav2Onc/Onc mice.

5. If there is an insulin resistance, the serum levels of insulin and leptin are elevated. Paradoxically, a Vav2-Rac1 axis is necessary for glucose-induced insulin secretion in the pancreas (Veluthakal et al., *Diabetologia*, 58, 2573, 2015). However, there is no different in the serum levels of insulin among high-fat diet-induced obese WT, Vav2L332A/L332A and Vav2Onc/Onc mice (Fig. S4). Explain why.

6. Because Vav2L332A/L332A show muscular hypotrophy whereas Vav2Onc/Onc mice do muscular hypertrophy, there might be difference in endurance exercise performance among WT, Vav2L332A/L332A and Vav2Onc/Onc mice. Examine endurance performance in WT, Vav2L332A/L332A and Vav2Onc/Onc mice.

7. In Fig. 3D and E, PH-Akt is recruited to the plasma membrane by EGFP-Onc and RacQ61L, suggesting that Vav2-Rac1 pathway is essential for the recruitment of Akt to the plasma membrane. Explain the precise molecular mechanism how the activated Rac1 phosphorylates Akt. I guess that Vav2-Rac1 might reorganize cortical actin and facilitate Akt to be recruited to the IP3-containing plasma membrane. Authors have to test the issue using actin depolymerizers such as latrunculin or cytochalasin. In the presence of actin depolymerizer, insulin/IGF-induced phosphorylation, GLUT4 translocation to the plasma membrane and glucose uptake should be observed in skeletal muscle obtained from WT and Vav2Onc/Onc mice.

Minor points

1. Fig. 1E represents the muscle fiber size. Vav2L332A/L332A mice might have bigger fiber size of gastrocnemius, compared to WT mice. Change the figure showing smaller muscle fiber of Vav2L332A/L332A mice, compared to WT mice.
2. WT shows phosphorylation of Akt after IGF-1 stimulation in Fig. 2I but not in Fig. 2L. The WT gastrocnemius must show IGF-induced phosphorylation of Akt. Change the Fig. 2L.
3. Discuss Vav2-Rac1-actin remodeling-Akt activation in the discussion section.
4. Which molecules are upstream signaling molecules of Vav2 in the skeletal muscle? Grb2? Src? Discuss the issue in the discussion section.

COMMENTS TO REFEREES MANUSCRIPT NCOMMS-19-14366

REVIEWER #1:

General comment. *In this manuscript, Rodriguez-Fdez and colleagues investigated the contribution of the GEF Vav2, a Rho/Rac/Cdc42 activator, to skeletal muscles growth and metabolism. Central to this study, they exploit mice with hypo-active Vav2 (L332A) or hyper-active Vav2 (onco-Vav2). Mice with low Vav2 activity presented with smaller muscles. They also failed to respond like wild-type animals to insulin. In contrast, mice with hyper-active Vav2 showed bigger muscles and were more responsive to insulin. The authors generated Vav2-null C2C12 cells and show that that they fail to respond correctly to insulin stimulation (pAKT, pGSK3 and pS6K). Hence, the authors propose that Vav2 controls the signaling output of the insulin pathway in muscle.*

This study is composed of a solid body of work. The mice demonstrate interesting metabolic phenotypes that the authors describe in great details. Where the story falls short is to provide mechanistic insights on the contribution of Vav2 to insulin signaling. In particular, what is, or are, the small GTPase target(s) of Vav2?

Authors' response: We thank the Referee for his/her kind comments. Regarding the mechanistic issues, we hope that this problem has been solved in the new version of the manuscript, as it will be explained in our comments to the specific Referees' queries. In particular, we believe that we now provide a more solid basis for the implication of the Vav2–Rac1 axis in this new pathway. We also provide evidence indicating that: **(i)** F-actin plays an upstream role in that pathway. **(ii)** It is Pak family-independent.

Major comment #1. *In figure 3, the authors studied CRISPR-derived Vav2 KO myoblasts (and rescued versions) to study a key question: the role(s) of Vav2 in insulin signaling. Unfortunately, these cells are poorly characterized. First, do they differentiate normally? This is an important question if the authors would want to generate myotubes. A panel of differentiation markers should be analyzed at different time points of differentiation (Myog, MHC, Myomaker etc.). The authors should also directly establish whether to formation of myotubes is unaffected by Vav2 KO or in the rescue conditions.*

Authors' response: Agree. We did not pursue this issue further in our original manuscript given that our data did not indicate significant changes in the differentiation of muscle cells both in vivo and in culture when using our gain- and loss-of-function models. In the new version of the manuscript, we provide the additional information requested by this Referee about this issue (see below). However, before getting into this, let us indicate that C2C12 cells can indeed differentiate into myotubes when using specific culturing conditions (e.g., use of specific differentiation media, long-term treatments with insulin or IGF1; see, for example, the data shown in the new **Figs. S5 and S6**) (Yaffe and Saxel, 1977). In fact, it is a cell model widely used in this type of studies.

Following the Referee's request, we have incorporated new information in the manuscript using our genetically manipulated C2C12 cells. These new data indicate that:

(a) $Vav2^{Onc}$ myoblasts have increased expression of mRNAs encoding some myoblast differentiation-associated transcriptional factors (*Myog*) and type II myosin heavy chain subunits (*Myh1*, *Myh4* and *Myh7*) (new **Fig. S5A**). The effect of $Vav2^{Onc}$ in the levels of some (*Myog*, *Myh7*), but not all (*Myh1*, *Myh4*) of those transcripts is further enhanced by the treatment of cells with insulin (new **Fig. S5C**). In the case of myosin II subunits, we have also found elevated expression using Western blot (old **Fig S3C**. Now, new **Fig. S5B**) and confocal microscopy (new **Fig. S5D,E**) analyses. We also demonstrate that all these $Vav2^{Onc}$ -elicited effects are catalysis-dependent (old **Fig. S3C**. Now, new **Fig. S5B-E**).

(b) $Vav2^{Onc}$ -expressing C2C12 cells, despite the results indicated above, do not exhibit any overt defect in differentiation (new **Fig. S5F,G**). However, they do display a statistically significant increase in the thickness of the differentiated myofibers (new **Fig. S5H**). This effect is abrogated when using the catalytically dead $Vav2^{Onc+E200A}$ mutant (new **Fig. S5H**).

(c) *Vav2* knockdown C2C12 cells only show consistent defects in the expression of the *Myog* transcript (new **Fig. S6A**). They also exhibit normal levels of differentiation (new **Fig. S6B,C**). Likewise, they exhibit no statistically significant change in size upon differentiation (new **Fig. S6D**). These results are consistent with the absence of myoblast differentiation defects found in $Vav2^{L332A/L332A}$ mice (**Figs. 1E-G**, new **S1B** and new **S1C**). Several explanations can justify these results: **(i)** That *Vav2* does not play any significant role in this process. **(ii)** That, as in the case of the *Vav2*-dependent pathways previously found in keratinocytes and smooth muscle cells (Lorenzo-Martin et al, manuscript submitted in *Oncogene* ONC-2020-00713, enclosed with this submission as supplemental file), the residual *Vav2* GEF activity present in the *Vav2* knockdown and $Vav2^{L332A/L332A}$ mice could be sufficient to maintain the differentiation response. **(iii)** The potential presence of other compensatory mechanisms (Rac1-dependent or independent) that can maintain the differentiation of both C2C12 and primary satellite cells in the absence of $Vav2^{WT}$.

We have included this new information in the new version of the manuscript to accommodate the Referee's request. It is worth noting, however, that the interest of exploring further these differentiation-related issues is questionable given the absence of overt differentiation defects in the skeletal muscles of both $Vav2^{Onc/Onc}$ and $Vav2^{L332A/L332A}$ mice. Indeed, our data are more consistent with a model in which the main role of *Vav2* is the regulation of both insulin and IGF1 signaling in the already differentiated skeletal muscle cells.

Major comment #2. *It is unclear why the signaling was studied in myoblasts rather than in myotubes, which would be the correct model to mimic the in vivo data (muscle fibers). Also, why did the authors rescue with onco-Vav2 and not wild-type Vav2? If Vav2 is contributing to insulin signaling, the wt protein should be sufficient to rescue the defects. One worry is that Onco-Vav2 may have dominant effects and not behave like the normal protein. As such, a bona fide contribution of Vav2 to insulin signaling is not clearly demonstrated in this study. Implicating Vav2 in insulin signaling goes beyond showing functions for Onco-Vav2.*

Authors' response: Agree. We have used undifferentiated cells to avoid the extensive manipulation of cells. The information requested is now included in the new version of the manuscript. These new data demonstrate that differentiated C2C12 cells lacking *Vav2* also show defects in the insulin-mediated activation of the PI3K–Akt pathway (new **Figs. 3H** and **S4C**).

Conversely, the opposite scenario is seen in the case of differentiated C2C12 cells expressing Vav2^{Onc} (new **Figs. 3H** and **S4D**). As in the case of the nondifferentiated cells, this Vav2^{Onc}-driven effect is catalysis dependent (new **Figs. 3H** and **S4D**).

Regarding the second issue, it is important to note that the experiment shown in old **Fig 3C** has been done with WT cells overexpressing Vav2^{Onc}, not with Vav2 knockout cells rescued with Vav2^{Onc} as the Referee indicates. We have changed the text to make this issue clearer to both the Referee and the future readers of the paper. We tried to avoid rescue experiments as much as possible given that C2C12 cells suffer a lot when subjected to very long periods of selection (in this case, the time required for the initial generation of the knockout or knockdown cells plus the subsequent transduction and selection processes to generate the rescued cells).

We would also like to add that the purpose of using Vav2^{Onc} was just to see the effects of robust Vav2 signaling in C2C12 cells and, in addition, if it triggered mirror-image signaling and biological effects to those found under conditions of Vav2 protein depletion. In addition, we could demonstrate using this method that the catalytic activity of Vav2^{Onc} is important for the signaling effects observed (the Onc+E200A mutant is not active in them). Please, take also into consideration that the role of Vav2^{WT} in this signaling process is demonstrated by the effects found under conditions of Vav2 protein depletion or catalytic inactivation using the 1A-116 inhibitor. To fully substantiate the significance of our data, it is also worth noting that: **(i)** We have always used independent cell clones to demonstrate that the effects obtained could be exclusively attributed to the loss of endogenous Vav2 in cells. **(ii)** Many results have been validated using different experimental approaches (gene-editing and shRNA interference, see new **Fig. 3G**). **(iii)** We have found a good correlation between the data obtained using genetically manipulated C2C12 cells and mice (both in the case of gain- and loss-of-function experiments). **(iv)** We have made a significant effort to carry out independent experiments to be able to get fully statistically significant data in all the key signaling experiments of the work.

Major comment #3. *The author make a gross assumption that Vav2 is mainly a Rac GEF in vivo in muscle tissue (on p4, this statement is supported by a reference to a review paper – the primary data should have been cited here?). There is absolutely no evidence in this study to support that Vav2 regulates preferentially Rac1 over Cdc42 and RhoA. The GTP-loading status of RhoA, Rac1 and Cdc42 should be determined at a minimum in the Vav2 KO myoblasts. Additional experiments should also be performed to directly determine the contribution of Vav2 GEF activity to insulin signaling. Can the authors test, as an example, if the rescue of insulin signaling with Onco-Vav2 in the KO C2C12 cells can be blunted by depleting Rac1 (and what is the impact of depleting Cdc42 or RhoA?). Finally, how does activation of the small GTPase(s) contribute to insulin signaling?*

Authors' response: Agree. From a biochemical point of view, Vav2 can regulate both Rac (e.g., Rac1, RhoG) and RhoA (e.g., RhoA, RhoB and RhoC) subfamily proteins. In our hands, Cdc42 is not a substrate for Vav family proteins under standard catalytic ratios. The structural reason for this has been addressed by us a long-time ago (Movilla et al., 2001). Using transfected cells and Vav2^{Onc/Onc} mice, we have shown before that this protein leads to increased GTP-bound levels in both Rac1 and RhoA (Fabbiano et al., 2014) (see **Fig. 1** in Lorenzo-Martin et al, submitted *Nat Commun* NCOMMS-1907108B, manuscript enclosed with this submission as supplemental file). In the case of Vav2^{L332A} protein, cell culture experiments indicate that it has lost ~70% and 100%

of exchange activity towards Rac1 and RhoA, respectively. It also shows a highly decreased, although detectable activity in downstream signaling (e.g., cytoskeletal remodeling, activation of c-Jun N-terminal kinase and serum responsive factor (see **Fig. 1** in Lorenzo-Martín et al, submitted *Oncogene* ONC-2020-00713). This residual activity is sufficient to maintain Vav2 physiological-dependent responses (e.g., blood pressure, lack of glaucoma) but not skin tumorigenesis (see **Figs. 3** and **4** in Lorenzo-Martín et al, submitted to *Oncogene* ONC-2020-00713). We have modified the Result section of the new version of the manuscript to better indicate both the catalytic specificities and activities of all the Vav2 proteins used in this study (page 6).

We agree with the Referee that, in our first submission, little information was provided regarding the GTPase substrates involved. In fact, the only information given was the observation that the active version of Rac1 (Q61L mutant) elicited a translocation of the mCherry-Akt PH bioreporter similar to that induced by Vav2^{Onc} when expressed in C2C12 cells (old **Fig. 3D**; now, new **Fig. 4A**). Following the Referee's advice, we have included the following new set of experimental observations in the second version of the manuscript.

(a) The demonstration that, in addition to EGFP-Rac1^{Q61L} (new **Figs. 4A, 5A** and **6D,E**), the Rac-related EGFP-RhoG^{Q61L} (new **Fig. 5A**) can induce the translocation of the mCherry-Akt PH bioreporter to the plasma membrane at levels similar to those observed when ectopically expressing EGFP-Vav2^{Onc} (new **Figs. 4A,B,C,D** and **6A**). Such a translocation is not observed when ectopically expressing either EGFP-RhoA^{Q63L} or EGFP-Cdc42^{Q61L} (new **Fig. 5A**).

(b) The finding that, as in the case of Vav2^{Onc} (new **Figs. 3E,H, S4B,D** and **4D**), the expression of either EGFP-Rac1^{Q61L} (new **Fig. 5B**) or EGFP-RhoG^{Q61L} (new **Fig. 5C**) promotes enhanced activation of the PI3K–Akt pathway in C2C12 cells upon insulin stimulation.

(c) The demonstration that the shRNA-mediated depletion of endogenous Rac1 leads to a reduction of the insulin-mediated activation of the PI3K–Akt pathway in both control and Vav2^{Onc}-expressing C2C12 cells (new **Figs. 5D** and **S7A**). These data were obtained using two independent *Rac1* shRNA-expressing cell clones.

(d) The finding that a chemical inhibitor that blocks the interaction of Vav proteins with Rac1 (but not with RhoA or Cdc42) (Cardama et al., 2014)(González et al, *Front Cell Develop Biol*, in press: [doi: 10.3389/fcell.2020.00240](https://doi.org/10.3389/fcell.2020.00240)) also leads to reduced levels of activation of the PI3K–Akt pathway in insulin-stimulated control and Vav2^{Onc}-expressing C2C12 (new **Fig. 5E**).

(e) Data showing that insulin-infused Vav2^{L332A/L332A} mice display low levels of activation of endogenous Rac1 in skeletal muscle (new **Fig. 5F,G**).

Collectively, these data indicate, in our opinion, that Rac1 must be the main Vav2 substrate involved in this new signaling pathway. At present, we cannot rule out a small involvement of RhoG in that pathway. However, it should be noted that the analysis of *Rhog*^{-/-} mice has not revealed any defect in skeletal muscle mass up to now (Goggs et al., 2013; Martinez-Martin et al., 2011; Vigorito et al., 2004).

Further issues on the mechanistics of the Vav2–Rac1–PI3K connection can be found in our answer to Referee #3 (see **Major comment #7**, see page 18).

Additional comment #1. *The introduction is incomplete. The Rho GTPases (in particular Rac1) and Rho GEFs have been described to control various aspects of myogenesis. This should be mentioned. Rac1 is important for early myogenesis (myoblast fusion step). Could depleting Vav2 affect this process in mice?*

Authors' response: Partially disagree. We have briefly summarized this information in our original Introduction (page 4): “Previous studies have shown that the main members of this family, Rac1, RhoA and Cdc42, play key stepwise roles of skeletal myogenesis (Reference #15)”. This sentence has been maintained in the new version of the manuscript (page 4). We believe that further information on this issue will confuse non-expert readers, given that our experimental data do not clearly support a role of Vav2 in myogenesis. Obviously, we could expand this information if the Referee feels otherwise.

The implication of Vav2^{WT} in myoblast fusion indicated by the Referee is an important issue. However, our data do not support this possibility. For example, the skeletal muscle of Vav2^{L332A/L332A} mice show thinner (old and new **Fig. 1E-G**) but not reduced numbers of cells (new **Fig. S1C**, left panel). Conversely, the skeletal muscle of Vav2^{Onc/Onc} mice show thicker fibers (old and new **Fig. 1H-J**) without statistically significant changes in the numbers of cells (new **Fig. S1C**, right panel). The total number of myofibers do not change during the differentiation of Vav2^{Onc}-expressing (new **Fig. S5F-H**) and Vav2 knockdown C2C12 cells in culture (new **Fig. S6B-D**). Clearly, other Rac1 GEFs must be doing this job either specifically or redundantly with Vav2 in this biological context.

Additional comment #2. *The authors state that there is hypertrophy in the Onco-Vav2 muscle. This has not been formally demonstrated. An alternative explanation, based on the observation that Onco-Vav2 myoblasts are more proliferative and differentiate better, that more cells are added to fibers. They authors should determine whether they have increased proliferation/fusion in vivo or if they observe hypertrophy.*

Authors' response: Agree. We thank the Referee for this important point. To address this question, we have quantified the number of nuclei per fiber in skeletal muscle sections from both Vav2^{L332A/L332A} (new **Fig. S1C**, left panel) and Vav2^{Onc/Onc} (new **Fig. S1C**, right panel) mice. These experiments indicate that there are not statistically significant differences in the number of cells/fiber in those two mouse models when compared to controls. Thus, these data indicate that the changes in muscle mass found in those two mouse strains are primarily caused by alterations in the size of the fibers.

REVIEWER #2:

General comment. *This study investigated the metabolic phenotypes of catalytically inactive ($Vav2^{L232}$) and gain-of-function strain ($Vav2^{onc}$) of $Vav2$ knockin mice. $Vav2^{L232}$ mice exhibit reduced muscle mass, decreased insulin responses, and are much prone to high fat diet (HFD)-induced obesity, whereas $Vav2^{onc}$ mice display these phenotypic changes opposite to $Vav2^{L232}$ mice. The authors conclude that $Vav2$ regulate insulin/IGF1 signaling in skeletal muscle, thereby explaining the metabolic phenotypes of these two strains of mice. The identification of $Vav2$ as a potential regulator of insulin signaling and muscle growth is potentially important. However, as neither $Vav2^{L232}$ mice nor $Vav2^{onc}$ mice are muscle-specific, the data presented is not sufficient to support the claim that the altered insulin responsiveness, adiposity and other metabolic parameters in the mice are due to the specific actions of $Vav2$ in muscle. Furthermore, the authors failed to explain how altered insulin signaling in skeletal muscle lead to changes in adiposity and adaptive thermogenesis of brown adipose tissues.*

Authors' response: We respectfully disagree. We acknowledge that the most definitive and clear-cut answer to the phenotypes observed would be the use of skeletal muscle-specific knock-in mice. However, this was not possible in our case for a number of reasons: **(i)** In the case of the $Vav2^{Onc/Onc}$ mice, and as previously described (Fabbiano et al., 2014), the mutant allele is expressed upon the Cre-mediated removal of an Stop cassette. Therefore, before the recombination step, these mice behave as knock-out and lack expression of the endogenous WT protein. This was the only option at the time of the generation of this strain, since a conventional exon swapping could not be done due to the need of removing the most 5' exon of the $Vav2$ locus. **(ii)** The $Vav2^{L332A/L332A}$ mice were initially designed to be able to carry out tissue-specific Cre-mediated recombination steps (see Fig. 2A in Lorenzo-Martin et al, submitted *Oncogene* ONC-2020-00713 article included as supplemental material for the reviewers). However, we found upon generating the mice that the genetically-modified WT allele could not be expressed well before the recombination step probably due to splicing problems. Due to this, we have been forced to use mice carrying the mutant allele from the germline in this case as well. If not done in that way, the analyses of the muscle-specific effects in these mice would be “contaminated” by having the rest of tissues in a knockout-like condition.

Notwithstanding those problems, it must be indicated that the $Vav2^{L332A}$ and $Vav2^{Onc}$ alleles are expressed at WT allele-like levels (this new piece of information is provided in the new **Table S1**). Furthermore, we could not see any compensation effects mediated by changes in expression of Rac1 and other Rac1 GEFs (this information is also provided in the new **Table S1**).

Despite using mouse models with the expression of the $Vav2^{L332A}$ and $Vav2^{Onc}$ alleles in all the tissues that normally express $Vav2$, we do believe that the defects found in skeletal muscle are intrinsic to this tissue because:

(a) The alterations found in the skeletal muscle are the earliest seen both in $Vav2^{L332A/L332A}$ and $Vav2^{Onc/Onc}$ mice (old Fig. 10. Now, new Fig. S18).

(b) The in vivo data obtained in skeletal muscle have been reproduced using both undifferentiated and differentiated C2C12 cells (new Figs. 3, S4, 5 and S7), a myoblast cell line commonly used in skeletal muscle-related studies.

We agree that, formally speaking, we cannot rule out the implication of Vav2 in the metabolic alterations found in the rest of tissues of our two mouse models. Having said that, we do believe that our proposed model is probably the most adequate to explain these systemic metabolic alterations because:

(a) They take place progressively at later stages than the skeletal muscle alterations found in both $Vav2^{L332A/L332A}$ and $Vav2^{Onc/Onc}$ mice (data summarized in old **Fig. 10**. Now, new **Fig. S18**).

(b) Most of those alterations are the spitting image of phenotypes previously seen in other mouse models associated with reductions (in the case of $Vav2^{L332A/L332A}$ mice) and hypertrophy (in the case of $Vav2^{Onc/Onc}$ mice) of the skeletal muscle mass (Bruning et al., 1998; Camporez et al., 2016; Christoffolete et al., 2015; Guo et al., 2009; Hamrick et al., 2006; Kim et al., 2000; Luo et al., 2006). Quite significantly, many of those models involve the genetic manipulation of receptors and signaling elements that participate in the insulin and IGF1 pathway in myocytes.

(c) The new hyperinsulinemic-euglycemic clamp-based experiments further indicate that there are no major alterations in insulin responses and glucose uptake in other peripheral tissues (new **Fig. S9E-H**).

(d) New data further indicate that there are no major alterations in the liver of $Vav2^{L332A/L332A}$ mice. These data were obtained using the administration of a methionine- and choline-free diet (a classical method to induce non-alcoholic fatty liver disease, new **Fig. S13**), tunicamycin treatments (new **Fig. S15**) and the administration of metformin (new **Fig. S16**).

(e) Using qRT-PCR analyses, we have found no major dysfunctions in the expression of enzymes involved in WAT function in young $Vav2^{L332A/L332A}$ and $Vav2^{Onc/Onc}$ mice (new **Fig. S10**).

Independently of these “collateral” metabolic alterations, let us indicate that the main take-home message of our work is the implication of Vav2 in the optimal stimulation of the insulin- and IGF1-mediated stimulation of the PI3K–Akt pathway in skeletal muscle cells. In fact, the rest of the metabolic alterations present in both $Vav2^{L332A/L332A}$ and $Vav2^{Onc/Onc}$ mice could be eliminated from the manuscript without affecting that message and the interest of the newly reported signaling pathway. However, as indicated above, we believe it is worth keeping them given the high level of similarity with the phenotype obtained in mice having deregulated insulin, IGF1 and PI3K–Akt signaling. In addition, the work done has allowed us to get a good view of the stepwise development of each of those defects during the life-span of mice.

Major comment #1. *To firmly draw the conclusion that altered insulin responsiveness of the two strains of the mice are attributed to altered insulin sensitivity in muscle, but not in other metabolic organs (adipose tissues, liver), glucose clamp studies should be performed to determine insulin sensitivity in each tissue in the mice.*

Authors’ response: Agree. Thank you for this suggestion. We have now performed the requested hyperinsulinemic-euglycemic clamp experiments. This was done in collaboration with R. Coppari and C. Veyrat-Durebex (who are now part of the authorship list of this work). These experiments were performed in 3.5-month-old $Vav2^{L332A/L332A}$ mice to avoid indirect influences from the rest of metabolic alterations that take place in the liver, WAT and BAT at later age time

points.

The data obtained indicate that $Vav2^{L332A/L332A}$ animals are indeed insulin resistant (new **Figs. 7F** and **S9A**). Furthermore, they also show decreased levels of peripheral glucose disposal rate (new **Fig. S9F**). Despite this, all the tissues tested, including a number of different skeletal muscle subtypes, display normal uptake of circulating glucose (new **Fig. S9G,H**). These results suggest that the insulin signaling dysfunctions found in the skeletal muscle of $Vav2^{L332A/L332A}$ mice are compensated by other mechanisms either intrinsic or extrinsic to myocytes. In this regard, it is worth noting that similar results have been found before when using loss-of-function mouse models for a large variety of insulin and IGF1 signaling elements, including the respective receptors, PI3K regulatory subunits, and PI3K α itself (Bruning et al., 1998; Li et al., 2019; Luo et al., 2006; O'Neill et al., 2015). Given the importance of skeletal muscle in glucose homeostasis, both in terms of proportional mass and clearance capacity (10-fold higher than WAT according to our data in new **Fig. S9G,H**), the glucose clearance defect found in $Vav2^{L332A/L332A}$ mice is probably explained by the reduced muscle mass present in those animals (old and new **Fig. 1A**).

Despite the insulin and IGF1 signaling defects present in skeletal muscle, we have found that $Vav2^{L332A/L332A}$ mice can maintain an euglycemic state until they reach the 8th month of age (old **Fig. 4F**, now new **Fig. 7G**). This feature has been previously seen in loss-of-function mouse models for insulin and IGF1 signaling elements (Bruning et al., 1998; Li et al., 2019; Luo et al., 2006; O'Neill et al., 2015). In fact, even the concurrent removal of the insulin and IGF1 receptors in skeletal muscle cannot trigger hyperglycemia in mice (O'Neill et al., 2015). It is possible therefore that, as it has been argued in those cases, the euglycemia could be maintained in $Vav2^{L332A/L332A}$ mice via the engagement of an as yet unidentified membrane receptor that can form complexes with either the insulin or IGF1 receptors. This idea is consistent, for example, with the observation that a dominant negative mutant version of the IGF1 receptor, but not the concurrent elimination of both the insulin and IGF1 receptors, does trigger hyperglycemia and hyperinsulinemia when expressed in skeletal muscle (O'Neill et al., 2015). It is also possible that the highly expanded WAT tissue present in those animals could contribute to increased glucose uptake in older animals (**Fig. 1A** and new **Fig. 8** [old **Fig. 6**]). Further work will be needed to unveil these compensatory mechanisms.

Our hyperinsulinemic-euglycemic clamp analyses do have revealed slightly decreased levels of insulin-mediated shut-off of glucose production in the liver of $Vav2^{L332A/L332A}$ animals (new **Fig. S9E**). This could indicate that, against our signaling results (old **Fig. S2,B**. Now, new **Fig. S3E,F**), there can be a defect in the responsiveness of the liver to insulin that could explain the short-term glucose intolerance exhibited by $Vav2^{L332A/L332A}$ mice. However, it is worth noting that previous studies have shown that this biological readout is not a good indicator of the actual insulin sensitivity of this tissue (Perry et al., 2015; Titchenell et al., 2016). Indeed, this readout is influenced by a number of factors such as, for example, alterations in free fatty acids (Perry et al., 2015; Titchenell et al., 2016). In line with this, a similar problem has been found when analyzing the metabolic alterations present in muscle-specific *Akt1;Akt2* knock-in mice (Jaiswal et al., 2019). Regardless of the cause involved, our signaling experiments clearly indicate that $Vav2^{L332A/L332A}$ mice show no defects in the insulin-mediated activation of the PI3K–Akt pathway in liver (old **Fig. S2,B**. Now, new **Fig. S3E,F**).

We would like to add that we have not found any significant alteration in other liver-associated

biological responses when the $Vav2^{L332A/L332A}$ mice have been submitted to: (i) A classical dietary model of non-alcoholic fatty liver disease, an experimental condition that promotes the rapid development of steatosis and reduction of body weight (new **Figs. S13** and **S14**). (ii) Tunicamycin (new **Fig. S15**), a yeast antibiotic that promotes endoplasmic reticulum stress by blocking N-glycosylation in hepatocytes. (iii) Long-term treatments with Metformin (**Fig. S16**), a drug that primarily acts by reducing glucose production in liver (Foretz et al., 2019).

Taken together, those foregoing results further suggest that the metabolic phenotype found in mice with deregulated catalytic activity of Vav2 is primarily caused by alterations in both insulin and IGF signaling in skeletal muscle.

Major comment #2. *Lipid profiles in the circulation and ectopic lipid accumulation in skeletal muscle should also be measured in both the chow-diet and HFD mice.*

Authors' response: Agree. As requested, we have measured serum cholesterol, serum triglycerides and skeletal muscle triglyceride content in both $Vav2^{L332A/L332A}$ and $Vav2^{Onc/Onc}$ mice.

In the former case, we have found increased cholesterol levels in plasma under high fat diet conditions when compared to controls (new **Fig. S12A**). However, we detected no statistically significant changes in the case of plasma triglycerides under both chow and HFD conditions (new **Fig. S12B**). We did find increased levels of triglycerides in the skeletal muscle of 10-month-old but not of 6-month-old $Vav2^{L332A/L332A}$ mice (new **Fig. S12E**) as well as when animals are maintained under HFD conditions (**Fig. S12F**). This result, together with the increased levels of lipids found in the WAT and BAT of these animals, suggest increased levels of uptake and accumulation of circulating triglycerides in all those tissues.

In the case of $Vav2^{Onc/Onc}$ mice, we have found reduced levels of cholesterol in plasma under HFD conditions (**Fig. S12C**). The rest of experimental conditions were all WT-like (**Fig. S12**).

Major comment #3. In Fig. 4 legend, the authors claimed that Reduction of Vav2 catalytic activity impairs systemic response to insulin. However, except ITT data in panel C (which is not convincing) and panel D, there is insufficient evidence to support this conclusion.

Authors' response: Agree. As indicated in our answer to your **Major Comment #1**, the hyperinsulinemic-euglycemic clamps carried out in 3.5-month-old $Vav2^{L332A/L332A}$ mice indicate that these mice are indeed insulin resistant (new **Figs. 7F** and **S9A**).

Major comment #4. *In fig. 5, how to explain why Vav2L232 mice are more prone to develop HFD-induced obesity? Although insulin signaling in Vav2L232 adipocytes is not affected, it does not necessarily mean that Vav2 has no direct effect on adipocytes.*

Authors' response: Agree. As indicated in our response to your **General comment**, the expansion of the WAT and the development of a HFD-like phenotype in $Vav2^{L332A/L332A}$ mice is fully consistent with previous observations using mouse models associated with reduced skeletal muscle mass. Those include many skeletal muscle-specific loss-of-function mouse models for insulin and PI3K pathway elements (Bruning et al., 1998; Luo et al., 2006; Moller et al., 1996). Conversely, the protection against HFD-induced obesity and associated dysfunctions seen in

$Vav2^{Onc/Onc}$ mice correlate well with observations previously made with mouse models displaying skeletal muscle hypertrophy (Camporez et al., 2016; Christoffolete et al., 2015; Guo et al., 2009; McPherron and Lee, 2002).

In the case of mouse models with reduced muscle mass, the reason for the increased in fat body content is as yet unclear. However, it must be skeletal muscle-intrinsic since it is also observed in mouse models in which the genetic manipulation has been specifically performed in that tissue. Several explanations, not mutually exclusive, are possible: **(i)** Reduced energy expenditure in mice caused by the reduction in muscle mass. The change in this metabolic parameter is seen, for example, in young $Vav2^{L332A/L332A}$ mice (old Fig. 5I,K. Now, new Fig. 8I,K). This change is clearly muscle-specific, since normal energy expenditure is observed when is normalized according to the specific amount of lean mass present in $Vav2^{L332A/L332A}$ and control mice (old Fig. 4J. Now, new Fig. 8J). **(ii)** Redirection of glucose to lipid biosynthesis in liver and WAT, leading to the subsequent storage at the WAT and BAT. We have not information on this possibility, although it is clear that the reduction of glucose uptake caused by the general loss of skeletal muscle mass has to be compensated by its uptake by other tissues to maintain the euglycemic state in young $Vav2^{L332A/L332A}$ mice. **(iii)** The paracrine influence of muscle-derived hormones in other peripheral tissues.

In the case of the models involving gain of muscle mass (such as $Vav2^{Onc/Onc}$ mice), it is possible that the protection against HFD-induced metabolic effects could be the result of the general increase of energy consumption due to the expanded skeletal muscle mass. This possibility, for example, is in good agreement with the metabolic data obtained with $Vav2^{Onc/Onc}$ animals (see old Fig. 6I,J that, now, is the new Fig. 9I,J). The long-distance effect of muscle-derived hormones in the metabolic status of other peripheral tissues cannot be ruled out either.

Further indicating that the problems in WAT are subsequent to the insulin- and IGF1-dependent defects found in the skeletal muscle of $Vav2^{L332A/L332A}$ mice, we have found that:

(a) The alterations in skeletal muscle mass present in those mice develop much earlier (2-month-old mice) than those found in gonadal fat (4-month-old mice) and liver (6-month-old mice) (summarized in old Fig. 10. Now, new Fig. S18).

(b) As indicated in our response to this Referee's **General comment** (point "e", see above), there is no statistically significant change in the abundance of transcripts encoding key metabolic enzymes in the WAT of 4-month-old $Vav2^{L332A/L332A}$ mice (new Fig. S10A). This is an age in which the increase in adiposity has only begun to emerge (old and new Fig. S1A, old Fig. 5 and new Fig. 8). It is also a period well before the development of the hyperglycemic state (8-month-old mice) that develops in those mice (old Fig. 4F. Now, new Fig. 7G). No statistically significant change has been found either in the levels of expression of insulin-regulated transcripts such as *Fasn* and *Pck1* (Claycombe et al., 1998; Granner et al., 1986; O'Brien and Granner, 1991) (new Fig. S10A), further indicating that the WAT has normal responsiveness to this endocrine factor as previously seen in our in vivo signaling experiments (old Fig. S2C,D. Now, new Fig. S3G,H).

Major comment #5. *In fig. 6 legend, it states that "Increase of Vav2 catalytic activity improves short-term systemic response". However, none of the data presented in this figure is related to short-term systemic responses. Again, how to explain the lean phenotype of the mice.*

Authors' response: Thank you for pointing out this mistake. It has been corrected in the new version of the manuscript (see the legend to new **Fig. 9**).

To explain the lean phenotype of $Vav2^{Onc/Onc}$ mice, see our prior comments to your **Major Comment #4** above.

Major comment #6. *In Fig. 7, the brown adipocyte function is poorly characterized. The changes in cell morphology and mild increase in mRNA expression of UCP1 do not mean increased brown adipocyte activity. UCP1 protein expression, mitochondrial density and complex activity, ex vivo oxygen consumption of brown adipose tissues, expression of a panel of genes involved in adaptive thermogenesis (PGC1alpha, CIEDA, DIO2...) should be analyzed. Furthermore, the mice should be challenged with cold environment to better assess the thermogenic functions of brown adipose tissues.*

Authors' response: Agree. Thank you for your suggestion. It is clear that the BAT display clear, mirror-image changes in its histology in $Vav2^{L332A/L332A}$ and $Vav2^{Onc/Onc}$ mice when compared to age-matched controls (old **Fig. 7**. Now, new **Fig. 10**). However, we agree that whether such changes eventually translate in clear metabolic differences is not clear with the data presented. In fact, the only data supporting this possibility is the detection of increased BAT temperature in the case of 3- and 12-month-old $Vav2^{Onc/Onc}$ mice. Following your suggestion, we have carried out the following experiments:

(a) Experiments with $Vav2^{L332A/L332A}$ mice:

As requested, we have investigated the levels of Ucp1 protein in the BAT of these animals using Western blot analyses. We selected for these studies 3-month-old mice, since at that age we can already see changes in skeletal muscle mass, defects in both insulin and IGF1 responsiveness in muscle, and incipient changes in BAT histology (old **Fig. 10**. Now, new **Fig. S18**). These analyses have revealed no statistically significant changes in the levels of Ucp1 protein when compared to control mice (new **Fig. S17A**, left panel). We have also found WT-like oxygen consumption rates (**Fig. S17B**, left panel) and mitochondrial content (**Fig. S17C**, left panel) in the BAT of those mice. Finally, we have also observed that the thermogenic response of these animals to low environmental temperatures is also WT-like (new **Fig. S17D**, left panel). We would like to point out, however, that this latter readout could not be adequate to get information about the metabolic status of the BAT under normal environmental conditions. This is because the thermogenic response to cold temperature is mediated by brain-located regulatory centers different from those activated by diet or other metabolic conditions (Morrison et al., 2008).

Taken together, these results suggest that the increased fat content detected in the BAT of $Vav2^{L332A/L332A}$ mice is, as in the case of the WAT, probably the consequence of increased lipid uptake rather than reduced metabolism by brown adipocytes. In line with this, we have shown before that the BAT temperature of $Vav2^{L332A/L332A}$ mice is similar to that found in control animals (old **Table S1**. Now, new **Table S2**).

We thank the reviewer very much for this interesting comment. Given that the new measurements indicate the lack of changes in Ucp1 protein levels and in BAT activity, we have now deleted the original data present in old **Fig. 7D** regarding the levels of expression of both the *Ucp1* and *Hsl*

mRNAs expression in the BAT of 4-month-old $Vav2^{L332A/L332A}$ mice to give a more straightforward message to the reader. In line with this, it is also important to note that the value of $Ucp1$ mRNA levels is rather limited per se given that its elevation does not always correlated with enhanced heat production (Nedergaard and Cannon, 2013).

(b) Experiments with $Vav2^{Onc/Onc}$ mice:

As described in our original manuscript (old **Table S1**. Now, new **Table S2**), we have found in this case increased BAT temperature in both 3- and 12-month-old mice when compared to age-matched controls. This suggests that some increased thermogenic activity is found in the BAT of these animals. In this case, we have also carried out all the assays requested by the Referee. These experiments indicate that there are WT-like levels of $Ucp1$ protein (new **Fig. S17A**, right panel), oxygen consumption rates (new **Fig. S17B**, right panel), and mitochondrial content (new **Fig. S17C**, right panel) in the BAT of 3-month-old $Vav2^{Onc/Onc}$ mice. We have also found WT-like thermogenic responses under low environmental temperature conditions in those mice (new **Fig. S17D**, right panel). In line with this, we have found minor changes in the expression of mRNAs involved in BAT metabolism around this age period (new **Fig. S17E-K**). As a result, and following the same rationale used for the data obtained with $Vav2^{L332A/L333A}$ mice (see point “a” above), we have removed the data on Hsl and $Ucp1$ mRNA levels originally contained in old **Figure 7L**. Collectively, these data suggest that the low lipid content found in those cells in young mice is, as in the case of the WAT, probably the result of reduced lipid availability rather than increased burning capability by brown adipocytes. However, it is worth noting that analyses performed following the recommendation of the Reviewer in older animals do seem to indicate a progressive upregulation in the BAT of mRNAs encoding thermogenesis factors such as $Ucp1$, Hsl , $Pgc1a$, $Prdm16$ in older $Vav2^{Onc/Onc}$ (new **Fig. S17E-K**). Whether this translates into better thermogenic responses in that tissue at older ages is as yet unknown. We have kept this information given that it was requested by the Referee (new **Fig. S17**), although it can be removed if the he/she considers that is not needed for the final take-home message of our paper.

These results, together with previous evidence from other mouse models (Bruning et al., 1998; Camporez et al., 2016; Christoffolete et al., 2015; Guo et al., 2009; Luo et al., 2006; McPherron and Lee, 2002; Moller et al., 1996), further suggest that the systemic alterations found in the liver, WAT and BAT are probably an indirect consequence of the dysfunctions found in the skeletal muscle of those of these two $Vav2$ mouse strains.

Major comment #7. *If $Vav2$ catalytic activity indeed modulates brown adipocyte activity, it may explain the changes in adiposity in the two strains of mice. Then, the authors need to perform additional studies to investigate how $Vav2$ modulate brown adipocyte activity or formation.*

Authors' response: Agree. However, based on the new data discussed in our reply to your **Major comment #6**, it is unlikely that this will occur in very early stages in the case of the two mouse models used. If so, the more likely scenario is that they will contribute to the metabolic phenotype of $Vav2^{Onc/Onc}$ mice only in animals older than 6 months (new **Fig. S17E-K**).

REVIEWER #3:

General comment. *Vav2, a GEF of Rac1, has been known to regulate insulin-elicited GLUT4 translocation by organizing cortical actin in the skeletal muscle and glucose-activated insulin secretion in the pancreas. Thus, Rac1 disruption in the skeletal muscle prevents insulin-dependent glucose uptake. This manuscript shows that Vav2 is involved in metabolic syndrome using Vav2L332A/L332A and Vav2Onc/Onc knock-in mice. Authors found the reduction of skeletal muscle mass and the increase of adipose mass in Vav2L332A/L332A mice, and vice versa in Vav2Onc/Onc mice. Because insulin-elicited Akt activation was affected specifically in the skeletal muscle of both knock-in mice, they concluded that Vav2-Rac1-PI3K-Akt axis is necessary for insulin signaling in the skeletal muscle. Because of previous finding of Vav2-Rac1-PI3K-Akt axis (Campa et al., Small GTPase, 6, 71-80, 2015), this manuscript lacks novelty.*

Authors' response: We respectfully disagree. The article cited by the referee (Campa et al., 2015) is a review, not an original piece of work. In addition, it does not refer in any part of the text to the implication of Vav2 in either skeletal muscle biology or in the regulation of the Rac1-PI3K-Akt pathway in that tissue. To the best of our knowledge, our work is totally original and novel. In addition, we believe that our work identifies a hitherto unknown signaling process that has important roles in the regulation of the optimal output of a critical pathway for human disease: the response of skeletal muscle to insulin and related factors.

Major comment #1. *According to Human Protein Atlas, the expression level of Vav2 is high in the liver and adipose tissue, pancreas but low in the skeletal muscle. Add the data for mRNA and protein levels of Vav2 and Rac1 in the various organs including the liver, adipose tissue, skeletal muscle and pancreas, using RT-PCR, immunoblotting and immunofluorescence in WT, Vav2L332A/L332A and Vav2Onc/Onc mice.*

Authors' response: Agree. We carried out qRT-PCR experiments to recheck the Human Protein Atlas data at the mRNA level in mice. In agreement with that database, we found high levels of Vav2 mRNA expression in the liver, BAT and, to a lower extent, in the WAT. Lower levels of expression were found both in pancreas and skeletal muscle (new **Table S1**). Importantly, these experiments also indicate that the expression of the Vav2 mRNA is similar in WT, Vav2^{L332A/L332A} and Vav2^{Onc/Onc} mice (new **Table S1**). Similar levels of expression of the different Vav2 versions present in those mice have been also found at the protein level (see **Fig. 2B** in Lorenzo-Martin et al, submitted to *Oncogene* [ONC-2020-00713] and included as supplemental material for the reviewers with this resubmission).

We would like to indicate that the essential function of a given protein does not necessarily have to correlate with its expression levels in the tissue of interest (providing, of course, that the protein is actually expressed there). Rac1 is, in fact, a good illustration of this point: it is expressed, like Vav2, at low levels in skeletal muscle when compared to liver, WAT and BAT (see new **Table S1** and the Human Protein Atlas). Yet, it does play crucial roles in skeletal muscle (Raun et al., 2018; Sylow et al., 2013; Sylow et al., 2014; Takenaka et al., 2019; Ueda et al., 2010). The reasons for the lack of correlation between expression and functional relevance can be several-fold: **(i)** The levels of expression of the protein might merely reflect tissue-specific rates of either transcriptional or biosynthetic activity that can diverge widely among different organs and cell types. **(ii)** That the protein involved can perform crucial functions that are not the ones

interrogated in a particular study (in this case, a specific signaling branch of the insulin and IGF1 pathways). (iii) That the function of the investigated protein can be compensated by other redundant molecules in the tissues in which it is highly expressed. Regardless of the reason involved, our experimental data clearly support the implication of Vav2 in the regulation of the PI3K–Akt in skeletal muscle downstream of both insulin and IGF1.

In line with the above, we would also like to add that, despite the high expression levels of Vav2 in liver, new sets of experiments indicate that there are no major alterations in the liver of *Vav2*^{L332A/L332A} mice (see our reply to Referee #2's **General Comment** (point d) and **Major comment #1**). Please, also check the data associated with those comments (new **Figs. S13, S14, S15** and **S16**).

Given that the readers could raise similar questions when reading the manuscript, we have included a paragraph in the Discussion of the new manuscript version to go over this issue (page 27): “The Vav2-mediated regulation of the PI3K–Akt axis is skeletal muscle-specific, because we could not detect any overt signaling alteration in other insulin-responsive tissues such as the liver and WAT in the case of *Vav2*^{L332A/L332A} mice. This is to some extent unexpected, given that Vav2 is expressed at high levels in liver, WAT and BAT (**Table S1**). It is likely that the role of Vav2 in these tissues could be redundant with other GEFs that are also expressed in those tissues such as P-Rex1 and P-Rex2 (liver), Vav3, Tiam1 and P-Rex family members (WAT) or Vav3 and P-Rex2 (BAT) (*Table S1*). Consistent with this idea, it has been shown before that P-Rex2, a guanine nucleotide exchange factor for Rac1, is involved in the stimulation of the PI3K–Akt axis in white adipocytes (Reference #60). However, given that *Vav2*^{Onc/Onc} mice do enhance the insulin-mediated activation of Akt in WAT, we cannot exclude at present whether the remaining catalytic activity of *Vav2*^{L332A} could suffice to ensure efficient insulin signaling in that tissue. It also remains to be explored whether Vav2 mediates the signaling output of the PI3K–Akt axis in other insulin-responsive tissues not tested in this work. In any case, the lack of hyperglycemia in young *Vav2*^{L332A/L332A} mice argues against this possibility”.

Major comment #2. *For analysis of insulin and IGF signaling in the liver, adipose tissue and skeletal muscle from WT and Vav2L332A/L332A and Vav2Onc/Onc mice, add the phosphorylation data of IR/IGFR, IRS-1, and Erk1/2. PIP2 and IP3 levels should be measured in the liver, adipose tissue and skeletal muscle from WT, Vav2L332A/L332A and Vav2Onc/Onc mice. In addition, after insulin and IGF treatment, the phosphorylation of IR/IGFR, IRS-1 and Erk1/2 and the level of PIP2 and IP3 should be measured from C2C12 myoblasts and satellite cells obtained from WT, Vav2L332A/L332A and Vav2Onc/Onc mice.*

Authors' response: The analyses indicated by the Referee are quite sound. However, we are afraid that he/she might not be fully aware of the number of animals and time required to carry out all these requested analyses. In addition, it is difficult that we can get permission to carry out such experiments by our Animal Experimentation Committee given that, in most cases, they are redundant with the data obtained using widely accepted readouts for the activation status of the PI3K pathway (e.g., Akt phosphorylation). We would also like to indicate that we have always aimed at repeating all the experiments to get solid statistically significant data and, in addition, that we have aimed at corroborating all our results in different experimental systems and readouts.

Having said that, we have attempted to satisfy the Referee's request whenever feasible. Due to this, we now have included the following data in the new version of the manuscript:

(a) Phospho-Erk levels in the Western blot analyses of old **Fig. 2** (now, new panels A, E, H and K). These analyses use extracts of skeletal muscle extracted from animals infused with insulin. Depending on the time-point involved in each experiment, we have found: **(i)** No stimulation of Erk phosphorylation when carrying out experiments involving either short stimulation time points or suboptimal stimulation conditions. **(ii)** No statistically significant variations in the levels of phospho-Erk obtained in the rest of experiments.

(b) Levels of phospho-Erk in insulin-stimulated *Vav2* KO and *Vav2^{Onc}*-expressing C2C12 cells (old **Fig. 2C**. Now, new **Fig. 3E**). No statistically significant variations were observed in this case either.

(c) Phosphorylation levels of immunoprecipitated IRS1 from nonstimulated and insulin-stimulated *Vav2* KO and *Vav2^{Onc}*-expressing C2C12 cells. No statistically significant changes were observed in this case (new **Fig. 3F**). This indicates that there are no major signaling defects at the level of either the insulin receptor or the IRS1 signaling layers.

(d) We have measured PIP₃ production in C2C12 cells. As expected, we found lower and higher levels of PIP₃ production in insulin-stimulated *Vav2* KO and *Vav2^{Onc}*-expressing C2C12 cells, respectively (new **Fig. 3I**).

(e) We have also tried to measure PIP₃ production in the skeletal muscle of our two mouse strains upon the *in vivo* infusion of insulin. However, these experiments were quite unsuccessful in our hands for technical reasons. We do not know at this point whether this problem was due to a contaminant carried over during the purification steps (e.g. organic compounds used in the extraction) or to other reasons. In any case, as indicated above, we believe that the deficient phosphorylation of Akt, a readout that was confirmed in the C2C12 models as well, gives a good indicator of the deficient activation of the PIK3 pathway in this tissue. Indeed, this is the standard readout used in works similar to ours when using mouse models.

Major comment #3. *No alteration of insulin and IGF signaling in the liver and adipose tissue in *Vav2^{L332A/L332A}* and *Vav2^{Onc/Onc}* mice suggests that there might be no change of *Rac1* activity. Thus, *Rac1* activity should be measured from the liver, adipose and skeletal muscle of *WT*, *Vav2^{L332A/L332A}* and *Vav2^{Onc/Onc}* mice after insulin and IGF administration. According to Human Protein Atlas, *Rac1* protein level is very low in the liver and adipose tissue! Thus, authors could explain why insulin and IGF signaling is not altered in the liver and adipose obtained from *WT*, *Vav2^{L332A/L332A}* and *Vav2^{Onc/Onc}* mice.*

Authors' response: Agree. The data on *Rac1* activation in skeletal muscle (new **Fig. 5F,G**), WAT (new **Fig. S7D**) and liver (new **Fig. S7E**) before and upon insulin infusion in *Vav2^{L332A/L332A}* mice has been included in the new version of the manuscript. The results indicate defective *Rac1* activation in the skeletal muscle of insulin-infused mice.

The Referee is again correct when indicating that the levels of expression of *Rac1* are low in skeletal muscle according to data from the Human Protein Atlas. We have confirmed such data

using qRT-PCR experiments (new **Table S1**). Please, see our reply to your **Major comment #1** for further comments on this issue.

Major comment #4. *According to previous paper (Chiu et al., Cell Signal., 23, 1546, 2011), there is an Akt-Vav2-Rac1-actin-GLUT4 signaling pathway in the skeletal muscle. Thus, the skeletal muscle of WT, Vav2L332A/L332A and Vav2Onc/Onc mice might have different GLUT4 translocation and glucose uptake. Observe insulin-elicited GLUT4 translocation to the plasma membrane and measure glucose uptake from the gastrocnemius and soleus of WT, Vav2L332A/L332A and Vav2Onc/Onc mice. In addition, adipose tissue might be a good negative control for the insulin-elicited GLUT4 translocation and glucose uptake in WT, Vav2L332A/L332A and Vav2Onc/Onc mice.*

Authors' response: We respectfully disagree. The publication cited by the Referee is a review, not an original article. Furthermore, its topic is related to the role of Rac1 signaling in the translocation of the Glut4 transporter in skeletal muscle (a biological process related to, but not identical to the one we are addressing in our study). In fact, the authors state in that excellent review that: **(i)** Akt and Rac1 (plus F-actin) work in parallel, not in a conventional lineal pathway in this specific biological response. **(ii)** “The Dbl-family GEF FLJ00068 (aka. Puratrophin-1, PLEKHG4 or ArhGEF44), but not of other Dbl-family GEFs including Dbl-I, α -PIX, β -PIX, Vav2 or SWAP-70, potentiated insulin-stimulated GLUT4 translocation and recruitment of GTP-loaded Rac1 to sites of remodeled actin in L6 muscle cells” (taken verbatim from the review article). They also propose that the most likely GEF candidate for that job is the GEF FLJ00068.

The Referee indicates that we must investigate the involvement of Vav2 in the translocation of Glut4. Although available evidence suggest that the Rac1 GEF in charge of this process in vivo is FLJ00068 (Takenaka et al., 2016; Takenaka et al., 2014; Ueda et al., 2008), we have carried out the experiments requested by the Referee. Specifically, we have included in the new version of the manuscript the following set of data:

(a) The demonstration that Vav2^{Onc}, but not the catalytically dead Vav2^{Onc+E200A} mutant, can translocate the Glut4 receptor in C2C12 cells (new **Fig. S8A,B**). This effect is probably due to the use of an active version of Vav2, given that the ectopic expression of Vav2^{WT} does not lead to Glut4 translocation as reported by Satoh's group (Ueda et al., 2008). We have also demonstrated that the translocation of Glut4 is delayed in the case of insulin-stimulated C2C12 cells lacking expression of endogenous Vav2 (new **Fig. S8C,D**).

(b) As requested, we have also carried out hyperinsulinemic-euglycemic clamp analyses in 3.5-month-old Vav2^{L332A/L332A} mice to measure glucose uptake in a number of tissues of those animals. The results obtained have been already described in our replay to Referee #2's **Major Comment 1** (see above in this rebuttal letter, page 7).

We have included these data in the new version of the manuscript, although it is worth noting that the regulation of the Glut4 transporter is only indirectly related to the pathway that has been studied in our work.

Major comment #5. *If there is an insulin resistance, the serum levels of insulin and leptin are elevated. Paradoxically, a Vav2-Rac1 axis is necessary for glucose-induced insulin secretion in*

the pancreas (Veluthakal et al., Diabetologia, 58, 2573, 2015). However, there is no difference in the serum levels of insulin among high-fat diet-induced obese WT, Vav2^{L332A/L332A} and Vav2^{Onc/Onc} mice (Fig. S4). Explain why.

Authors' response: Agree. However, please bear in mind that, as already indicated in our original manuscript (old Fig. 4A-E. Now, new Fig. 7A-D and E), the defects in insulin responsiveness found in the skeletal muscle of Vav2^{L332A/L332A} mice only lead to short-term glucose intolerance problems when the animals are injected with either glucose or insulin (this initial finding has now been corroborated using the hyperinsulinemic-euglycemic clamp analyses requested by Referees #2 and #3, see new Figs. 7F and S9A).

Despite those defects, these animals can maintain normal levels of glucose and insulin for a significant period of time (until they become older than 8 months; see old Fig. 4F. Now, new Fig. 7G). This paradoxical observation has been found in many skeletal muscle-specific loss-of-function mouse models for signaling elements of the insulin, IGF1 and PI3K pathway (e.g., insulin and IGF1 receptors, regulatory subunits of PI3K, PI3K α) (Bruning et al., 1998; Li et al., 2019; Luo et al., 2006; O'Neill et al., 2015). This suggests that defects in insulin, IGF1, PI3K α and Vav2 signaling in skeletal muscle are compensated by alternative mechanisms. We have discussed a number of possibilities for this compensatory mechanism in our reply to Referee #2's **Major comment #1** (see above). In addition, we have devoted part of the Discussion section of the new manuscript version to address this issue (page 27): "Despite the insulin and IGF1 signaling defects present in skeletal muscle, the Vav2^{L332A/L332A} mice can maintain an euglycemic state until they reach the 8th month of age. This feature has been previously seen in other loss-of-function models for insulin and IGF1 signaling elements, including PI3K α itself (References #9,35,36,50). It is possible that euglycemia could be maintained in those animals through the uptake of glucose in the highly expanded WAT tissue. Alternatively, the skeletal muscle of those animals could engage an as yet unidentified membrane receptor that could contribute to the regulation of glucose metabolism by forming complexes with either the insulin or IGF1 receptors. This idea is consistent, for example, with the observation that a dominant negative mutant version of the IGF1 receptor, but not the concurrent elimination of both the insulin and IGF1 receptors, does trigger hyperglycemia and hyperinsulinemia when expressed in skeletal muscle (Reference #9). Further work will be needed to unveil these compensatory mechanisms present in skeletal muscle and, perhaps, other gluco-regulatory tissues".

Regarding the issue that "there is no difference in the serum levels of insulin among high-fat diet-induced obese WT, Vav2^{L332A/L332A} and Vav2^{Onc/Onc} mice (Fig. S4). Explain why", we would like to remind the Referee that all these data have been gathered in animals maintained under chow diet, not high fat diet. The reason for the lack of hyperinsulinemia is the same that the lack of hyperglycemia already discussed above in this point.

Regarding the issue that "Paradoxically, a Vav2-Rac1 axis is necessary for glucose-induced insulin secretion in the pancreas (Veluthakal et al Diabetologia, 58: 2573, 2015)", we would like to indicate that this paper is based on the use of a single rat insulinoma cell line (INS-1 832/13 cells) and lacks any validation with animal models. Without questioning the validity of those results, the data obtained using Vav2^{L332A/L332A} mice is clear cut: the catalytic activity of endogenous Vav2 is not essential for insulin release from pancreas in mice (see old Fig. S4B,C. Now, new Fig. S9C,D). Whether this is due to lack of functional involvement in this pathway or to the redundant action

of other Rac1 GEF remains to be determined.

Elaborating more on this, we would like to indicate that the article by Veluthakal et al. is clearly very far from the biological responses that we have dissected in our current manuscript. In fact, no demonstration is given in that work regarding a potential connection of the Vav2-Rac1 axis with PI3K signaling (the authors link in fact the Vav2-Rac1 response to F-actin remodeling processes in pancreatic cells). In addition, the authors place glucose upstream of Vav2 in that work. In our case, Vav2 is clearly upstream of glucose responses (and downstream of both insulin and IGF1) in skeletal muscle cells.

Major comment #6. *Because Vav2^{L332A/L332A} show muscular hypotrophy whereas Vav2^{Onc/Onc} mice do muscular hypertrophy, there might be difference in endurance exercise performance among WT, Vav2^{L332A/L332A} and Vav2^{Onc/Onc} mice. Examine endurance performance in WT, Vav2^{L332A/L332A} and Vav2^{Onc/Onc} mice.*

Authors' response: Partially agree. The Referee is correct in the case that the changes in the structure of skeletal muscle are severe. However, this might not be the case when the defects are not dire enough to affect the mechanical functions of skeletal muscle. For example, Kahn's group has shown using knock-in mice that the specific loss of PI3K α in skeletal muscle does not change affect the endurance exercise of young animals. Yet, these mice show a loss in skeletal muscle mass even more severe than that found in the case of Vav2^{L332A/L332A} animals (Li et al., 2019). Same results were found in other studies using mice with reduced muscle mass (Luo et al., 2006; Wojtaszewski et al., 1999).

Having said this, we did conduct the sound experiments requested by the Referee. To this end, we subjected 3-month-old Vav2^{L332A/L332A} and Vav2^{Onc/Onc} to a treadmill challenge. The age of the animals was chosen to avoid the possibility that the changes in locomotor endurance could be caused by the alterations in fat content exhibited by these mice at later age points. In agreement with the results found by Kahn and coworkers, we have found no difference in the distance and time ran by these animals when compared to control animals (new Fig. S1D,E and S1G,H).

Major comment #7. *In Fig. 3D and E, PH-Akt is recruited to the plasma membrane by EGFP-Onc and RacQ61L, suggesting that Vav2-Rac1 pathway is essential for the recruitment of Akt to the plasma membrane. Explain the precise molecular mechanism how the activated Rac1 phosphorylates Akt. I guess that Vav2-Rac1 might reorganize cortical actin and facilitate Akt to be recruited to the IP3-containing plasma membrane. Authors have to test the issue using actin depolymerizers such as latrunculin or cytochalasin. In the presence of actin depolymerizer, insulin/IGF-induced phosphorylation, GLUT4 translocation to the plasma membrane and glucose uptake should be observed in skeletal muscle obtained from WT and Vav2^{Onc/Onc} mice.*

Authors' response: Agree. Recent results have shown that Rac1 can favor the stimulation of the PI3K α -Akt axis via three, not mutually exclusive mechanisms: **(i)** The formation of multiprotein complexes nucleated by Pak kinases (Higuchi et al., 2008; Ijuin and Takenawa, 2012). **(ii)** The polymerization of actin in juxtamembrane areas of cells (Asahara et al., 2013; Campa et al., 2015; Murga et al., 2002). **(iii)** Direct interactions with PI3K signaling complexes (Campa et al., 2015; Murga et al., 2002).

The first of those models is not compatible with our data, given that we have observed that

RhoG^{Q61L} (new **Fig. 5A**) and Rac1^{Q61L+Y40C} (new **Fig. 6C-E**) can promote the translocation of the mCherry-Akt PH bioreporter as efficiently as Rac1^{Q61L} itself despite lacking the ability to bind Pak family members (Lamarque et al., 1996; Prieto-Sanchez and Bustelo, 2003).

Our observations do indicate that the cytoskeleton does favor this signaling step, although from an upstream rather than downstream position relative to Vav2. Consistent with this, we have found that latrunculin A and cytochalasin D abrogate the plasma membrane localization of Vav2^{Onc} and, as a consequence, the translocation of the mCherry-Akt PH bioreporter (new **Fig. 6A and E**). By contrast, the ability of Rac1^{Q61L} to promote such a translocation is much less dependent on the F-actin cytoskeleton as shown by: **(i)** Treatments of the transfected cells with latrunculin A and cytochalasin D (**Fig. 6B**). **(ii)** The use of Rac1 switch mutants incapable of promoting F-actin reorganization in cells (**Fig. 6C,D**).

These results indicate that this pathway is not identical to the Rac1- and F-actin-mediated pathway involved in the translocation of Glut4 (Khayat et al., 2000; Moller et al., 2019; Sylow et al., 2014). Based on these new data, it is likely that Rac1, upon activation by Vav2, will directly form complexes with PI3K α as previously described in other cell types (Campa et al., 2015; Murga et al., 2002) (new **Fig. 6E**). Whether such complexes entail direct physical contacts or the participation of bridge molecules such as the PI3K noncatalytic subunits remains to be determined (Campa et al., 2015).

Regardless of the mechanism involved, our protein depletion experiments support the idea that Rac1 is the main Vav2 catalytic substrate involved in this new pathway. In line with this, it is worth noting that no skeletal muscle defects have been found so far in *Rhog*^{-/-} mice (Goggs et al., 2013; Martinez-Martin et al., 2011; Vigorito et al., 2004).

We have not pursued the same investigation on the effect of Vav2 and Rac1 in Glut4 because, as indicated above (see our reply to this Referee's **Major Comment 4**), this pathway is unrelated with the one dissected in the current manuscript. In addition, it is worth noting that:

(a) According to our data, we have not found any major defects in glucose uptake in the case of both *Vav2*^{L332A/L332A} and *Vav2*^{Onc/Onc} mice (new **Fig. S9G,H**).

(b) The role of Rac1 in Glut4 translation has been extensively described before by others (Raun et al., 2018; Sylow et al., 2013; Sylow et al., 2014; Takenaka et al., 2019; Ueda et al., 2010)

(c) The most important Rac1 GEF involved in this process seems to be FLJ00068 according to excellent previous work by Satoh's group (Ueda et al., 2008).

Additional comment #1. *Fig. 1E represents the muscle fiber size. Vav2L332A/L332A mice might have bigger fiber size of gastrocnemius, compared to WT mice. Change the figure showing smaller muscle fiber of Vav2L332A/L332A mice, compared to WT mice.*

Authors' response: Agree. We have changed this panel as requested (see new **Fig. 1E**).

Additional comment #2. *WT shows phosphorylation of Akt after IGF-1 stimulation in Fig. 2I but not in Fig. 2L. The WT gastrocnemius must show IGF-induced phosphorylation of Akt. Change the Fig. 2L.*

Authors' response: We respectfully disagree. As indicated in the figure legend, this experiment has been done with suboptimal concentrations of IGF that cannot trigger the stimulation of the PI3K–Akt pathway in WT mice. This strategy allowed us to visualize the effect of Vav2^{Onc} protein in this pathway (old Fig. 2L–N. Now, new Fig. 2K–M). By contrast, as in the case of insulin signaling (see old Fig. 2E–H. Now, new Fig. S3A–D), no differences in the activation of that pathway were observed at optimal stimulation conditions of IGF1 (data not shown).

In the new version of the manuscript, we have changed the main text to avoid this confusion. In addition, we have included a new experiment using suboptimal concentrations of insulin in Vav2^{Onc/Onc} mice. Unlike the case of the optimal conditions (old Fig. 2E; new Fig. S3A–D), we did detect enhanced Akt phosphorylation under these new conditions (new Fig. 2E,F). As already shown in the first version of the manuscript, we have found that Vav2^{L332A/L332A} mice show reduced levels of activation of the PI3K–Akt route in skeletal muscle under optimal insulin and IGF1 stimulation conditions (old and new Fig. 2A–D and 2I–K). These results are fully consistent with the mechanistic model proposed in our study.

Additional comment #3. *Discuss Vav2-Rac1-actin remodeling-Akt activation in the discussion section.*

Authors' response: Agree. According to our new experimental data, F-actin seems to be upstream rather than downstream of Vav2 (new Figs. 4E,F; 6A and 6E). By contrast, our experiments show that PIK3 α is located downstream of Vav2 in this pathway (old Fig. S3F,G. Now, new Fig. 4E,F). Using specific Rac1^{Q61L} switch mutants (see scheme in new Fig. 6C), we have also demonstrated that this GTPase can trigger the translocation of the mCherry-Akt PH bioreporter in an F-actin- and Pak-family-independent manner (new Fig. 6D,E). This information has been included both in the Results and Discussion section of the new manuscript version).

Additional comment #4. *Which molecules are upstream signaling molecules of Vav2 in the skeletal muscle? Grb2? Src? Discuss the issue in the discussion section.*

Authors' response: We do have no information on this. Grb2 is a possibility, given that is a well-known interactor of Vav family proteins. Alternatively, it is possible that Vav2 binds directly to phospho-IRS proteins via its phosphotyrosine-binding SH2 domain. Given that we have already a lengthy Discussion to integrate all the data obtained in our work, we prefer not to get into further elaborations on this issue unless the Referee believes otherwise.

REFERENCES USED IN REBUTTAL LETTER

Asahara, S., Shibutani, Y., Teruyama, K., Inoue, H.Y., Kawada, Y., Etoh, H., Matsuda, T., Kimura-Koyanagi, M., Hashimoto, N., Sakahara, M., *et al.* (2013). Ras-related C3 botulinum toxin substrate 1 (RAC1) regulates glucose-stimulated insulin secretion via modulation of F-actin. *Diabetologia* 56, 1088-1097.

Bruning, J.C., Michael, M.D., Winnay, J.N., Hayashi, T., Horsch, D., Accili, D., Goodyear, L.J., and Kahn, C.R. (1998). A muscle-specific insulin receptor knockout exhibits features of the metabolic syndrome of NIDDM without altering glucose tolerance. *Mol Cell* 2, 559-569.

Campa, C.C., Ciraolo, E., Ghigo, A., Germena, G., and Hirsch, E. (2015). Crossroads of PI3K and Rac pathways. *Small GTPases* 6, 71-80.

Camporez, J.P., Petersen, M.C., Abudukadier, A., Moreira, G.V., Jurczak, M.J., Friedman, G., Haqq, C.M., Petersen, K.F., and Shulman, G.I. (2016). Anti-myostatin antibody increases muscle mass and strength and improves insulin sensitivity in old mice. *Proc Natl Acad Sci U S A* 113, 2212-2217.

Cardama, G.A., Comin, M.J., Hornos, L., Gonzalez, N., Defelipe, L., Turjanski, A.G., Alonso, D.F., Gomez, D.E., and Menna, P.L. (2014). Preclinical development of novel Rac1-GEF signaling inhibitors using a rational design approach in highly aggressive breast cancer cell lines. *Anticancer Agents Med Chem* 14, 840-851.

Christoffolete, M.A., Silva, W.J., Ramos, G.V., Bento, M.R., Costa, M.O., Ribeiro, M.O., Okamoto, M.M., Lohmann, T.H., Machado, U.F., Musaro, A., *et al.* (2015). Muscle IGF-1-induced skeletal muscle hypertrophy evokes higher insulin sensitivity and carbohydrate use as preferential energy substrate. *Biomed Res Int* 2015, 282984.

Claycombe, K.J., Jones, B.H., Standridge, M.K., Guo, Y., Chun, J.T., Taylor, J.W., and Moustaid-Moussa, N. (1998). Insulin increases fatty acid synthase gene transcription in human adipocytes. *Am J Physiol* 274, R1253-1259.

Fabbiano, S., Menacho-Marquez, M., Sevilla, M.A., Albarran-Juarez, J., Zheng, Y., Offermanns, S., Montero, M.J., and Bustelo, X.R. (2014). Genetic dissection of the Vav2-Rac1 signaling axis in vascular smooth muscle cells. *Mol Cell Biol* 34, 4404-4419.

Foretz, M., Guigas, B., and Viollet, B. (2019). Understanding the glucoregulatory mechanisms of metformin in type 2 diabetes mellitus. *Nat Rev Endocrinol* 15, 569-589.

Goggs, R., Harper, M.T., Pope, R.J., Savage, J.S., Williams, C.M., Mundell, S.J., Heesom, K.J., Bass, M., Mellor, H., and Poole, A.W. (2013). RhoG protein regulates platelet granule secretion and thrombus formation in mice. *J Biol Chem* 288, 34217-34229.

Granner, D.K., Sasaki, K., Andreone, T., and Beale, E. (1986). Insulin regulates expression of the phosphoenolpyruvate carboxykinase gene. *Recent Prog Horm Res* 42, 111-141.

Guo, T., Jou, W., Chanturiya, T., Portas, J., Gavrilova, O., and McPherron, A.C. (2009). Myostatin inhibition in muscle, but not adipose tissue, decreases fat mass and improves insulin sensitivity. *PLoS ONE* 4, e4937.

Hamrick, M.W., Pennington, C., Webb, C.N., and Isales, C.M. (2006). Resistance to body fat gain in 'double-muscled' mice fed a high-fat diet. *Int J Obes* 30, 868-870.

Higuchi, M., Onishi, K., Kikuchi, C., and Gotoh, Y. (2008). Scaffolding function of PAK in the PDK1-Akt pathway. *Nat Cell Biol* 10, 1356-1364.

Ijuin, T., and Takenawa, T. (2012). Regulation of insulin signaling by the phosphatidylinositol 3,4,5-triphosphate phosphatase SKIP through the scaffolding function of Pak1. *Mol Cell Biol* 32, 3570-3584.

Jaiswal, N., Gavin, M.G., Quinn, W.J., 3rd, Luongo, T.S., Gelfer, R.G., Baur, J.A., and Titchenell, P.M. (2019). The role of skeletal muscle Akt in the regulation of muscle mass and glucose homeostasis. *Mol Metab* 28, 1-13.

Khayat, Z.A., Tong, P., Yaworsky, K., Bloch, R.J., and Klip, A. (2000). Insulin-induced actin filament remodeling colocalizes actin with phosphatidylinositol 3-kinase and GLUT4 in L6 myotubes. *J Cell Sci* 113 Pt 2, 279-290.

Kim, J.K., Michael, M.D., Previs, S.F., Peroni, O.D., Mauvais-Jarvis, F., Neschen, S., Kahn, B.B., Kahn, C.R., and Shulman, G.I. (2000). Redistribution of substrates to adipose tissue promotes obesity in mice with selective insulin resistance in muscle. *J Clin Invest* 105, 1791-1797.

Lamarche, N., Tapon, N., Stowers, L., Burbelo, P.D., Aspenstrom, P., Bridges, T., Chant, J., and Hall, A. (1996). Rac and Cdc42 induce actin polymerization and G1 cell cycle progression independently of p65PAK and the JNK/SAPK MAP kinase cascade. *Cell* 87, 519-529.

Li, M.E., Lauritzen, H., O'Neill, B.T., Wang, C.H., Cai, W., Brandao, B.B., Sakaguchi, M., Tao, R., Hirshman, M.F., Softic, S., *et al.* (2019). Role of p110a subunit of PI3-kinase in skeletal muscle mitochondrial homeostasis and metabolism. *Nat Commun* 10, 3412.

Luo, J., Sobkiw, C.L., Hirshman, M.F., Logsdon, M.N., Li, T.Q., Goodyear, L.J., and Cantley, L.C. (2006). Loss of class IA PI3K signaling in muscle leads to impaired muscle growth, insulin response, and hyperlipidemia. *Cell Metab* 3, 355-366.

Martinez-Martin, N., Fernandez-Arenas, E., Cemerski, S., Delgado, P., Turner, M., Heuser, J., Irvine, D.J., Huang, B., Bustelo, X.R., Shaw, A., *et al.* (2011). T cell receptor internalization from the immunological synapse is mediated by TC21 and RhoG GTPase-dependent phagocytosis. *Immunity* 35, 208-222.

McPherron, A.C., and Lee, S.J. (2002). Suppression of body fat accumulation in myostatin-deficient mice. *J Clin Invest* 109, 595-601.

Moller, D.E., Chang, P.Y., Yaspelkis, B.B., 3rd, Flier, J.S., Wallberg-Henriksson, H., and Ivy, J.L. (1996). Transgenic mice with muscle-specific insulin resistance develop increased adiposity, impaired glucose tolerance, and dyslipidemia. *Endocrinology* 137, 2397-2405.

Moller, L.L.V., Klip, A., and Sylow, L. (2019). Rho GTPases-Emerging Regulators of Glucose Homeostasis and Metabolic Health. *Cells* 8.

Morrison, S.F., Nakamura, K., and Madden, C.J. (2008). Central control of thermogenesis in mammals. *Exp Physiol* 93, 773-797.

Movilla, N., Dosil, M., Zheng, Y., and Bustelo, X.R. (2001). How Vav proteins discriminate the GTPases Rac1 and RhoA from Cdc42. *Oncogene* 20, 8057-8065.

Murga, C., Zohar, M., Teramoto, H., and Gutkind, J.S. (2002). Rac1 and RhoG promote cell survival by the activation of PI3K and Akt, independently of their ability to stimulate JNK and NF-kappaB. *Oncogene* 21, 207-216.

Nedergaard, J., and Cannon, B. (2013). UCP1 mRNA does not produce heat. *Biochim Biophys Acta* 1831, 943-949.

O'Brien, R.M., and Granner, D.K. (1991). Regulation of gene expression by insulin. *Biochem J* 278, 609-619.

O'Neill, B.T., Lauritzen, H.P., Hirshman, M.F., Smyth, G., Goodyear, L.J., and Kahn, C.R. (2015). Differential Role of Insulin/IGF-1 Receptor Signaling in Muscle Growth and Glucose Homeostasis. *Cell Rep* 11, 1220-1235.

Perry, R.J., Camporez, J.G., Kursawe, R., Titchenell, P.M., Zhang, D., Perry, C.J., Jurczak, M.J., Abudukadier, A., Han, M.S., Zhang, X.M., *et al.* (2015). Hepatic acetyl CoA links adipose tissue inflammation to hepatic insulin resistance and type 2 diabetes. *Cell* 160, 745-758.

Prieto-Sanchez, R.M., and Bustelo, X.R. (2003). Structural basis for the signaling specificity of RhoG and Rac1 GTPases. *J Biol Chem* 278, 37916-37925.

Raun, S.H., Ali, M., Kjobsted, R., Moller, L.L.V., Federspiel, M.A., Richter, E.A., Jensen, T.E., and Sylow, L. (2018). Rac1 muscle knockout exacerbates the detrimental effect of high-fat diet on insulin-stimulated muscle glucose uptake independently of Akt. *J Physiol* 596, 2283-2299.

Sylow, L., Jensen, T.E., Kleinert, M., Hojlund, K., Kiens, B., Wojtaszewski, J., Prats, C., Schjerling, P., and Richter, E.A. (2013). Rac1 signaling is required for insulin-stimulated glucose uptake and is dysregulated in insulin-resistant murine and human skeletal muscle. *Diabetes* 62, 1865-1875.

Sylow, L., Kleinert, M., Pehmoller, C., Prats, C., Chiu, T.T., Klip, A., Richter, E.A., and Jensen, T.E. (2014). Akt and Rac1 signaling are jointly required for insulin-stimulated glucose uptake in skeletal muscle and downregulated in insulin resistance. *Cell Signal* 26, 323-331.

Takenaka, N., Araki, N., and Satoh, T. (2019). Involvement of the protein kinase Akt2 in insulin-stimulated Rac1 activation leading to glucose uptake in mouse skeletal muscle. *PLoS ONE* 14, e0212219.

Takenaka, N., Nihata, Y., and Satoh, T. (2016). Rac1 Activation Caused by Membrane Translocation of a Guanine Nucleotide Exchange Factor in Akt2-Mediated Insulin Signaling in Mouse Skeletal Muscle. *PLoS ONE* 11, e0155292.

Takenaka, N., Yasuda, N., Nihata, Y., Hosooka, T., Noguchi, T., Aiba, A., and Satoh, T. (2014). Role of the guanine nucleotide exchange factor in Akt2-mediated plasma membrane translocation of GLUT4 in insulin-stimulated skeletal muscle. *Cell Signal* 26, 2460-2469.

Titchenell, P.M., Quinn, W.J., Lu, M., Chu, Q., Lu, W., Li, C., Chen, H., Monks, B.R., Chen, J., Rabinowitz, J.D., *et al.* (2016). Direct Hepatocyte Insulin Signaling Is Required for Lipogenesis but Is Dispensable for the Suppression of Glucose Production. *Cell Metab* 23, 1154-1166.

Ueda, S., Kataoka, T., and Satoh, T. (2008). Activation of the small GTPase Rac1 by a specific guanine-nucleotide-exchange factor suffices to induce glucose uptake into skeletal-muscle cells. *Biol Cell* 100, 645-657.

Ueda, S., Kitazawa, S., Ishida, K., Nishikawa, Y., Matsui, M., Matsumoto, H., Aoki, T., Nozaki, S., Takeda, T., Tamori, Y., *et al.* (2010). Crucial role of the small GTPase Rac1 in insulin-stimulated translocation of glucose transporter 4 to the mouse skeletal muscle sarcolemma. *FASEB J* 24, 2254-2261.

Vigorito, E., Bell, S., Hebeis, B.J., Reynolds, H., McAdam, S., Emson, P.C., McKenzie, A., and Turner, M. (2004). Immunological function in mice lacking the Rac-related GTPase RhoG. *Mol Cell Biol* 24, 719-729.

Wojtaszewski, J.F., Higaki, Y., Hirshman, M.F., Michael, M.D., Dufresne, S.D., Kahn, C.R., and Goodyear, L.J. (1999). Exercise modulates postreceptor insulin signaling and glucose transport in muscle-specific insulin receptor knockout mice. *J Clin Invest* 104, 1257-1264.

Yaffe, D., and Saxel, O. (1977). Serial passaging and differentiation of myogenic cells isolated from dystrophic mouse muscle. *Nature* 270, 725-727.

CHANGES MADE IN NEW VERSION NCOMMS-19-14366A

(A) Main text:

Title page. We have included four new authors (M.I. Fernández-Pisonero, C. Veyrat-Durebex, D. Beiroa and R. Coppari) that have been actively involved in some of the new experiments carried out for this resubmission.

Abstract. We have made some changes in the abstract (highlighted in red). Total final words: 120.

Introduction. We have made minor changes (highlighted in red).

Results. They have been modified to include all the new experiments and controls suggested by Referees. Changes in the text have been highlighted in red.

Discussion. It has been modified to accommodate the new results and to discuss some of the issues raised by the referees (changes made have been highlighted in red).

Experimental Procedures. They were extensively modified to incorporate all the new techniques used in the new version of the manuscript (changes made have been highlighted in red).

References. We have incorporated new references (not highlighted).

Figure legends. They have been modified to incorporate the new experimental data requested by the Referees.

(B) Figures:

Old/New Figure 1. We have changed panel E, following the request indicated by Referee #1 (Additional Comment #1).

Old/New Figure 2. The original panels E to F (infusion of $Vav2^{Onc/Onc}$ mice with optimal amounts of insulin) have been transferred to the new Fig. S3A-D). These panels have been replaced by the new panels E to G (infusion of $Vav2^{Onc/Onc}$ mice with suboptimal amounts of insulin). The rest of panels have been maintained (A, D and I to N, which are now the new panels H to M).

Old/New Figure 3. We have included new panels for $Vav2$ knockdown cells (B, D and G). As a result, the old panel C is now the new panel E. We have also included panels to show: **(i)** The phosphorylation of IRS1 in control and $Vav2$ KO cells (new panel F; requested by Referee #3, see Major Comment #2). **(ii)** The effect of the shRNA-mediated depletion of endogenous $Vav2$ protein in the insulin pathway (new panel G). **(iii)** The effect of deregulated $Vav2$ signaling in the insulin-mediated stimulation of the PI3K–Akt pathway in differentiated C2C12 cells (new panel H; requested by Referee #2, see Major Comment #2). **(iv)** The levels of PIP_3 production in basal and insulin-stimulated C2C12 cells in the presence of $Vav2^{Onc}$ and $Vav2^{Onc+E200A}$ as well as in the absence of endogenous $Vav2$ (new panel I; requested by Referee #3, see Major Comment #2).

Due to the inclusion of all these data, the panels D to F present in the old version of this figure have been transferred to the new **Fig. 4** (see below).

New Figure 4. We have included here the panels D (now A), E (now B) and F (now C) originally present in old **Fig. 3**. We have also included in this figure the panels originally present in the old **Fig. S3** to consolidate all the information in a single figure.

New Figure 5. We have included new data to demonstrate the implication of Rac1 in this Vav2-dependent pathway. Requested by Referee # 1 (General Comment and Major Comment #3).

New Figure 6. We have included new data to establish the role of the F-actin cytoskeleton and Pak family proteins in the Vav2- and Rac1-mediated regulation of insulin signaling in C2C12 cells. Requested by Referee #3 (see General Comment and Majors Comments #3, #4 and #7).

New Figure 7/old Figure 4. We have incorporated a new panel (F) showing results from the new euglycemic-hyperglycemic clamp analyses (requested by Referee #2, see Major Comment #1). The rest of panels have been maintained. However, due to the inclusion of the new panel F, the old panels F to J are now the new panels G to K in the modified figure.

New Figure 8. This is the old **Fig. 5**. No changes made.

New Figure 9. This is the old **Fig. 6**. No changes made.

New Figure 10. This is the old **Fig. 7**. We have removed the data showing *Ucp1* mRNA levels (original panels D and L), according to our comments made to Referee #2 (see Major Comment #6). As a result, some of the positions of other panels have also been changed.

Old Figure 8. This figure is now the new **Fig. S18**.

(C) Supplemental information:

(C.1) Supplemental text:

Front page: We have included the names of the new authors included in the new version of the manuscript.

Figure legends. We have extensively modified them to accommodate all changes made in the new version of the manuscript.

(C.2) Supplemental figures:

Old/New Figure S1. We have modified panel A to specify more clearly the structure of the $Vav2^{Onc}$ mutant protein expressed in the knock-in mice. We have also included three new panels with information regarding the number of cells/muscle fiber (C) and the performance of skeletal muscle in $Vav2^{L332A/L332A}$ (D and E) and $Vav2^{Onc/Onc}$ mice (G and H). These data have been requested by Referee #1 (Additional Comment #2) and Referee #3 (Major Comment #6). As a result of these incorporations, the old panel C is now the new panel F in the reformatted version

of this figure.

Finally, due to space constraints, we have transferred the original panels D to K of the old version of this figure to the new **Fig. S2**.

New Figure S2. It contains all the information originally included in old **Fig. S1D-K** as indicated above.

New Figure S3. This is the old **Fig. S2**. We have included here as panels A to D those originally present in old **Fig. 2E to H**, respectively. The content of the rest of panels originally present in old **Fig. S2** (now panels B to I) have not been modified.

New Figure S4. It includes densitometric data from the signaling experiments shown in the main text.

New Figure S5. In addition to a new panel (B) originally present in old **Fig. S3C**, we have included 7 new panels (A and C to H) with information on the effect of ectopically expressed $Vav2^{Onc}$ on the differentiation of C2C12 cells (experiments requested by Referee #1, Major Comment #1).

New Figure S6. We included here new information on the effect of the depletion of endogenous $Vav2$ on the differentiation of C2C12 cells (experiments requested by Referee #1, Major Comment #1).

New Figure S7. It contains: **(i)** Densitometry data of signaling experiments included in the main text (panel A). **(ii)** The effect elicited by the total depletion of endogenous $Rac1$ on the insulin-mediated stimulation of the PI3K–Akt pathway in C2C12 cells (panels B and C. This is associated with the experiments requested by Referee # 1 in his/her General Comment and Major Comment #3). **(iii)** Data on the levels of $Rac1$ activity present in the liver and WAT from insulin-stimulated control and $Vav2^{L332A/L332A}$ mice (panels D and E). These data have been requested by Referee #3 (Major Comment #3).

New Figure S8. New data using gain- and loss-of-function models regarding the effect of $Vav2$ in the translocation of the Glut4 transporter (data requested by Referee # 3, Major Comment #4).

New Figure S9. New data obtained from the hyperinsulinemic-euglycemic clamp analyses requested by requested by Referee #2 (Major Comment #1).

New Figure S10. New data showing the data of mRNAs for key metabolic enzymes in the WAT from $Vav2^{L332A/L332A}$ (panel A) and $Vav2^{Onc/Onc}$ (panel B) mice. These data were requested by Referee #2 (Major Comment #4).

New Figure S11. This is the old **Fig. S5**. No changes have been made.

New Figure S12. New data on cholesterol and triglyceride levels requested by Referee #2 (Major Comment #2).

New Figure S13-S16. New data with assays to test liver responses in our mouse models.

New Figure S17. New data on metabolic status of BAT requested by Referee #2 (Major

Comment #6).

New Figure S18. This is old **Fig. 8**. We have made minor modifications to incorporate some of the data generated in this submission. In addition, we have made minor stylistic changes in the overall design of the figure.

(C.3) Supplemental tables:

New Table S1. It contains data regarding the expression of mRNAs encoding Vav2, other Rac1 GEFs and Rac1 in tissues from both *Vav2*^{L332A/L332A} and *Vav2*^{Onc/Onc} mice. These data have been requested by both Referee #2 (General Comment) and Referee #3 (Major Comments #1 and #3).

New Table S2. This is the old **Table S1**. Only stylistic changes have been made in the new design of the table.

Reviewers' Comments:

Reviewer #1:

Remarks to the Author:

The authors provided new experiments to answer my previous questions and comments. In addition, when not possible, they provided clear and reasonable explanations. Overall, the manuscript now provides stronger mechanistic insights explaining the potential roles of Vav2 in insulin signalling.

Jean-Francois Cote

Reviewer #2:

Remarks to the Author:

The revised manuscript has addressed most of my concerns. However, there are still a couple of points which need further clarification:

1. In response to my general comments "However, as neither Vav2^{L232} mice nor Vav2^{onc} mice are muscle-specific, the data presented is not sufficient to support the claim that the altered insulin responsiveness, adiposity and other metabolic parameters in the mice are due to the specific actions of Vav2 in muscle. Furthermore, the authors failed to explain how altered insulin signaling in skeletal muscle lead to changes in adiposity and adaptive thermogenesis of brown adipose tissues". The authors disagree. However, the authors only emphasized the difficulties in generating skeletal muscle-specific KO mice. This weakness remains an issue and should not be discussed

2. The new hyperinsulinemic-euglycemic clamp data is complicated (as agreed by the authors) and does not support the notion that deregulated catalytic activity of Vav2 is primarily caused by alterations in both insulin and IGF signaling in skeletal muscle.

Reviewer #3:

Remarks to the Author:

The authors responded satisfactorily to the questions I asked. However, there is still one question. Hyperactive Vav2^{ONC} has a gain of function in C2C12 cells, showing Vav2^{ONC}-induced GLUT4 trans-localization to the plasma membrane even without insulin treatment as shown in Fig. S8A and B. The Vav2^{ONC}-induced GLUT4 trans-localization to the plasma membrane is similar that after insulin stimulation in control cells. However, Fig. 2 and 3 did not show the effect of Vav2^{ONC} as a gain of function in Akt, GSK, and G6K phosphorylation in insulin-untreated samples. Resolve this discrepancy.

COMMENTS TO REFEREES
MANUSCRIPT NCOMMS-19-14366A

REVIEWER #1:

Remarks to the Author. *The authors provided new experiments to answer my previous questions and comments. In addition, when not possible, they provided clear and reasonable explanations. Overall, the manuscript now provides stronger mechanistic insights explaining the potential roles of Vav2 in insulin signalling.*

Authors' response: Thank you for your decision and useful help during all this reviewing process.

REVIEWER #2:

Remarks to the Author. *The revised manuscript has addressed most of my concerns. However, there are still a couple of points which need further clarification:*

1. In response to my general comments “However, as neither Vav2^{L232} mice nor Vav2^{onc} mice are muscle-specific, the data presented is not sufficient to support the claim that the altered insulin responsiveness, adiposity and other metabolic parameters in the mice are due to the specific actions of Vav2 in muscle. Furthermore, the authors failed to explain how altered insulin signaling in skeletal muscle lead to changes in adiposity and adaptive thermogenesis of brown adipose tissues”. The authors disagree. However, the authors only emphasized the difficulties in generating skeletal muscle-specific KO mice. This weakness remains an issue and should not discussed

2. The new hyperinsulinemic-euglycemic clamp data is complicated (as agreed by the authors) and does not support the notion that deregulated catalytic activity of Vav2 is primarily caused by alterations in both insulin and IGF signaling in skeletal muscle.

Authors' response: Agree. It is clear that our phenotypes are consistent and correlate well with previous observations made with gain- and loss-of-function mutations for many signaling elements of the insulin/IGF1 pathway in skeletal muscle. Although they are not clarified in any of the foregoing skeletal muscle-specific models, it is generally assumed that the collateral effects seen in WAT, liver and other tissues are the consequence of the metabolic changes that take place in the skeletal muscle when the output of those pathways becomes dysregulated. In this context, it is also important that the caveats indicated by the referee in his/her remark #2 have been found in some of those alternative skeletal muscle-specific mouse models.

However, although our explanation is perhaps the most plausible according to the data available, **we do agree** with the referee that we cannot formally demonstrate this unequivocally since we could not generate skeletal muscle-specific alterations using our two mouse models. Due to this, we have included a specific sentence in the **new Discussion section** to point out this problem. The new text now indicates (page 30, new modifications in brown):

“We have observed that *Vav2*^{L332A/L332A} and *Vav2*^{Onc/Onc} mice progressively develop mirror-image histological and functional alterations in the BAT, WAT, and liver (**Fig. S18a,b**). These physiological alterations, which take place at older ages than the skeletal muscle dysfunctions, recapitulate well those extensively seen in other mouse models with either reduced (in the case of *Vav2*^{L332A/L332A} mice) or increased (in the case of *Vav2*^{Onc/Onc} mice) skeletal muscle mass^{12,13,15,17,18,67,74}. Many dysfunctions found in *Vav2*^{L332A/L332A} mice also resemble those found in HFD-fed WT mice (this work), further suggesting that they are caused by the progressive accumulation of body fat indirectly caused by the early alterations in muscle mass and function detected in those mice. According to our data and previous publications^{112,13,15,17,18,68,75}, we believe that the most plausible explanation for all those late dysfunctions is that they are side effects of the signaling alterations present in the skeletal muscle of both *Vav2*^{Onc/Onc} and *Vav2*^{L332A/L332A} mice. However, we caution the readers that we cannot formally exclude the possibility that they could be the consequence of the alteration of some intrinsic functions of *Vav2* in other peripheral tissues given that we have not utilized skeletal muscle-specific knock-in animals in our work.”

We hope that with these changes the concern raised by the referee is fully conveyed to the readers of the article.

Thank you for your comments on the revised version of our manuscript.

REVIEWER #3:

Remarks to the Author. *The authors responded satisfactorily to the questions I asked. However, there is still one question. Hyperactive Vav2ONC has a gain of function in C2C12 cells, showing Vav2ONC-induced GLUT4 trans-localization to the plasma membrane even without insulin treatment as shown in Fig. S8A and B. The Vav2ONC-induced GLUT4 trans-localization to the plasma membrane is similar that after insulin stimulation in control cells. However, Fig. 2 and 3 did not show the effect of Vav2ONC as a gain of function in Akt, GSK, and G6K phosphorylation in insulin-untreated samples. Resolve this discrepancy.*

Authors' response: Thank you for your comments. Regarding the issue of the effects found with *Vav2*^{Onc} in the translocation of the Glut4 transporter, there is in fact no discrepancy. In agreement with our observations, it has been shown before that constitutively active versions of Rac1 and upstream GEFs (e.g., Tiam1 and Plekhg4 [also known as FLJ00068 and puratrophin-1]) promote a similar effect (Ueda et al, *Biol Cell* 2008, PMID: 18482007; Chiu et al, *J Biol Chem* 2013, PMID: 23640896. For reviews, see Chiu et al, *Cell Signal* 2011, PMID: 21683139; Satoh, *Small GTPases* 2014, PMID: 24613967; Klip et al, *Am J Physiol Cell Physiol* 2014, PMID: 24598362 & Jaldin-Fincati et al., *Trends Endocrinol Metab* 2017, PMID: 28602209) . Constitutively active Rac1 does so even when ectopically expressed *in vivo* (Ueda et al, *FASEB J* 2010, PMID: 20203090).

It is also important to note that, according to current knowledge, Akt and constitutively active Rac1 impinge on the translocation of the Glut4 transporter using converging but independent pathways. In the former case, it seems that Akt mediates the translocation of Glut4 via AS160 and Rab proteins (and, possibly, other targets as well). By contrast, constitutively active Rac1 does so using both F-actin- and RalA-dependent mechanisms (see reviews mentioned above) .

Based on this model, it is clear that constitutively active Rac1 and Rac1 GEFs (Vav2, Tiam1, Plekhg4) can promote this effect without the need of engaging the PI3K-Akt pathway.

As we mentioned in the previous rebuttal letter, we do not believe that this biological response is very relevant for the take-home message of our work. Of course, it is clear these data provide further support for the involvement of Rac1 in the signaling pathway of Vav2 in skeletal muscle. However, our *in vivo* work suggests that Vav2 must not regulate this pathway in primary cells (or, if it does it, it has to do it in a concerted manner with other redundant GEFs). In line with this, previous publications from Satoh's group suggest that the most likely candidate for the Rac1-mediated regulation of Glut4 translocation is the GEF Plekhg4 (Ueda et al, *Biol Cell* 2008; PMID: 18482007; Takenaka et al, *FASEB J* 2014, PMID: 24438685 & Takenaka et al, *Cell Signal* 2016, PMID: 25025572).

We consider that the point raised by the referee is important to make this issue clearer to the readers. Due to this, we have incorporated some of the above information in several parts of our manuscript. Thus, in the **new Results section**, we now say (page 16): “The lack of insulin-dependency of Vav2^{Onc} in this regulatory step is consistent with previous results using constitutively active versions of either Rac1 or other upstream GEFs such as Tiam1 and Plekhg4 (also known as FLJ00068 and puratrophin-1)^{22,24,44,51,53,54}.”

In the **new Discussion section**, we also indicate that (page 27): “Unlike the case of the activation of the PI3K α -Akt axis, we have observed that ectopically-expressed Vav2^{Onc} can promote the translocation of the Glut4 transporter to the plasma membrane in the absence of insulin stimulation. These results mimic the observations previously made with chronically activated versions of both Rac1 and other upstream GEFs^{22,24,44,51,53,54}. They are also consistent with previous findings indicating that Rac1 and the PI3K α -Akt axis contribute to the translocation of Glut4 using mechanistically independent pathways^{22,51,52,70}. We have also found that the knockdown of endogenous Vav2 delays the translocation of Glut4 in C2C12 cells. Despite those data, the relevance of this regulatory step *in vivo* is unclear given that Vav2^{L332A/L332A} and Vav2^{Onc/Onc} mice do not exhibit overt alterations in glucose uptake. It is possible, however, that this function of Vav2 could be redundantly performed by other Rac1 GEFs present in skeletal muscle. Based on current data, the most likely candidate for this is Plekhg4⁵³.”

We thank again the referee for helping us clarifying specific aspects of our work.

Reviewers' Comments:

Reviewer #2:

Remarks to the Author:

The revised version has fully addressed my concerns. thank you

Reviewer #3:

Remarks to the Author:

The authors resolved the discrepancy in a gain of function of VAV2ONC

COMMENTS TO REFEREES
MANUSCRIPT NCOMMS-19-14366B

REVIEWER #2:

Remarks to the Author. *The revised version has fully addressed my concerns. Thank you.*

Author's response: Thank to you for both the input and suggestions made throughout all this editorial process.

REVIEWER #3

Remarks to the Author. *The authors resolved the discrepancy in a gain of function of Vav2^{Onc}.*

Author's response: Thank to you for both the input and suggestions made throughout all this editorial process.